# Vertical profiles of aerosol and black carbon in the Arctic: a seasonal phenomenology along two years (2011-2012) of field campaigns

Luca Ferrero[1], David Cappelletti[2,3], Maurizio Busetto[3], Mauro Mazzola[3], Angelo Lupi[3], Christian Lanconelli[4], Silvia Becagli[5], Rita Traversi[5], Laura Caiazzo[5], Fabio Giardi[5], Beatrice Moroni[2], Stefano Crocchianti[2], Martin Fierz[6], Griša Močnik[7,8], Giorgia Sangiorgi[1], Maria G. Perrone[1], Marion Maturilli[9] and Vito Vitale[3], Roberto Udisti[5], Ezio Bolzacchini[1]

[1]Department of Earth and Environmental Sciences, University of Milano-Bicocca, Piazza della Scienza 1, 20126, Milano, Italy

[2]Department of Chemistry, Biology and Biotechnology, University of Perugia, 06123 Perugia, Italy

[3]Institute of Atmospheric Sciences and Climate, CNR-ISAC, Via Gobetti 101, 40129, Bologna, Italy

[4]European Commission, Joint Research Centre (JRC), Institute for Environment & Sustainability, via E. Fermi, 2749 - 21027 Ispra (VA), Italy

[5]University of Florence, Via della Lastruccia 3, 50019, Sesto Fiorentino, Florence Italy

[6]University of Applied Sciences Northwestern, Switzerland, Windisch, Switzerland

[7]Aerosol d.o.o., Kamniska 41, SI-1000 Ljubljana, Slovenia

[8]Condensed Matter Physics Department, Jozef Stefan Institute, Jamova 39, SI-1000 Ljubljana, Slovenia

[9]Alfred Wegener Institute, Helmholtz Centre for Polar and Marine Research, Telegraphenberg 43A, 14473 Potsdam, Germany.

*Correspondence to*: Luca Ferrero (luca.ferrero@unimib.it)

**Abstract.** We present results from a systematic study of vertical profiles of aerosol number size distribution and black carbon (BC) concentrations conducted in the Arctic, over Ny-Ålesund (Svalbard). The campaign lasted 2 years (2011-2012) and resulted in 200 vertical profiles measured by means of a tethered balloon (up to 1200 m AGL) during the spring and summer seasons. In addition, chemical analysis of filter samples, aerosol size distribution and a full set of meteorological parameters were determined at ground. The collected experimental data allowed a classification of the vertical profiles into different typologies, which allowed us to describe the seasonal phenomenology of vertical aerosol properties in the Arctic.

During spring, four main types of profiles were found and their behaviour was related to the main aerosol and atmospheric dynamics occurring at the measuring site. Background conditions generated homogenous profiles. Transport events caused an increase of aerosol concentration with altitude. High Arctic haze pollution trapped below thermal inversions promoted a decrease of aerosol concentration with altitude. Finally, ground-based plumes of locally formed secondary aerosol determined profiles with decreasing aerosol concentration located at different altitude as a function of size.

During the summer season, the impact from shipping caused aerosol and BC pollution plumes constrained close to the ground, indicating that increasing shipping emissions in the Arctic could bring anthropogenic aerosol and BC in the summer Arctic affecting the climate.

## 1    Introduction

The Arctic is subject to an amplification of the global warming, as the observed temperature increase has been almost twice than the global average (IPCC, 2013; Serreze and Barry, 2011; Shindell and Faluvegi, 2009). This resulted in the first complete opening of the Northwest Passage in 2007 (Serreze et al., 2007) together with a greening of the coastal tundra (Bhatt et al., 2010) and altered wind patterns (Overland and Wang, 2010). The "Arctic amplification" is the result of complex global feedbacks (acting at different spatial and temporal scales): the impact of sea ice changes on the heat fluxes between the ocean and the atmosphere (Screen and Simmonds, 2010a and 2010b), the effect of changes in the cloud cover and water vapor on the longwave radiation fluxes (Francis and Hunter, 2006), the changes in atmospheric and oceanic heat transports (Yang et al., 2010), the black carbon (BC) deposition on the snow (Hansen and Nazarenko, 2004), and the changes in the atmospheric BC and aerosol concentrations themselves (Flanner, 2013; Serreze and Barry, 2011; Shindell and Faluvegi, 2009). Many of these processes  depend on aerosol absorption and scattering of the solar radiation (direct effect). Additionally, indirect effects play an important role as the aerosols seed and modify the cloud properties. Lastly, light absorption by BC can alter the atmospheric thermal structure within, below, or above clouds consequently affecting cloud distributions (IPCC, 2013; Bond et al., 2013; Ramanathan and Feng, 2009; Koren et al. 2008; Koren et al., 2004; Kaufman et al., 2002). Shindell and Faluvegi (2009) estimated that globally the decreasing concentrations of sulfate aerosols and the increasing concentrations of BC contributed (during 1976–2007) with 1.09±0.81 °C to the Arctic surface temperature increase of 1.48±0.28 °C.

Aerosol particles are short-lived pollutants (~one-few weeks of residence time) and act as short-lived climate forcers; thus, their effect could be employed in short-term climate strategies (Ødemark et al., 2012; Shindell et al., 2012; Jacobson, 2010; Quinn et al., 2008). To adopt the right mitigation strategies, key scientific issues in the study of Arctic aerosols has to be solved. They include the identification of the relative importance of  long-range advection with respect to local emissions (Flanner, 2013; Sand et al., 2013; Shindell and Faluvegi, 2009). Most important, the seasonal characterization of the aerosol vertical structure, a very poorly determined piece of information, is required.

Indeed, the aerosol properties (size distribution, chemical composition, optical properties) in the Arctic exhibit a pronounced seasonal variation due to an interplay of dominating sources (outside or inside the Arctic region) with meteorological conditions that allow or inhibit the transport from source regions (Quinn et al., 2008; Eckhardt et al., 2003). The spring period is characterized by the presence of the Arctic Haze dominated by the accumulation mode aerosol (enriched in BC).  During the Arctic Haze, an inflow of pollution (aerosol and gases) from northern mid-latitudes (during winter-spring) results in a reduction in visibility (Jacob et al., 2010; Sthol et al., 2006; Radke et al., 1984; Barrie and Hoff, 1985; Brock et al., 1989; Shaw, 1995). The Arctic Haze occurs under meteorological conditions with stable stratifications and the frequent and persistent occurrences of surface-based inversions. According to Stohl et al. (2006), within these conditions, the air pollution can be transported into the Arctic at low-level (followed by ascent in the Arctic or low-level alone) or with an uplift outside the Arctic, followed by descent

in the Arctic itself. The summer period is dominated by fresh Aitken particles, locally formed, with a negligible BC content (Tunved et al., 2013; Spackman et al., 2010;Eleftheriadis et al., 2009; Ström et al., 2003 and 2009; Udisti et al., 2013; Viola et al., 2013).

In addition to the aforementioned seasonality, the same type of aerosol can produce different climatic effects (warming or cooling) and local feedbacks (snow/ice-albedo, clouds) depending on its vertical location (Flanner, 2013; Sand et al., 2013; Shindell and Faluvegi, 2009). For example, it is well known that BC aerosol absorbs solar radiation and heats the surrounding air (Ferrero et al., 2014 and 2011a; Samset et al., 2014; Samset et al., 2013; Ramana et al., 2007). The surface temperature response varies considerably with the altitude of the induced heating. BC may potentially warm the Arctic if it is located immediately above snow and ice while it has a cooling effect, if it is located in the free troposphere. In the latter case, BC may reduce the surface air temperature and promote the increase in the sea-ice fraction (Flanner, 2013; Brock et al., 2011; Seinfeld and Pandis, 2006; Hansen and Nazarenko, 2004). The latter phenomenon results from a combination of the weakening of the northward heat transport (due to a reduction in the meridional temperature gradient) and the increasing of atmospheric stability (caused by the contemporary dimming of the surface and heating aloft) which turns into a reduction of the downward sensible heat flux (Flanner, 2013; Sand et al., 2013; Shindell and Faluvegi, 2009).

In addition to the vertical distribution of BC, that of the total aerosol particles is important; it can influence the indirect effect and the related feedbacks. Changes in the cloud cover (especially low-level Arctic stratus) increase the downward longwave flux to the surface in function of the cloud base temperature and cloud phase (liquid, mixed or ice) (Serreze and Barry, 2011; Francis and Hunter, 2006). Low clouds mainly warm the surface in the Arctic (with the exception of a brief period in summer) (Vavrus et al., 2009; Intrieri et al., 2002) due to the stable stratified conditions that often prevail in the Arctic (Manabe and Wetherald, 1975). Because the highest number density of aerosol particles observed in the Arctic is due to a locally formed aerosol (mainly in summer as stated above) (Tunved et al., 2013; Engvall et al., 2008; Ström et al., 2003) it is important to assess the vertical behavior of the aerosol concentration in function of its size and the season.

It is therefore necessary to measure the vertical profiles in the Arctic.

At this purpose, several field campaigns have been performed in the Arctic in recent years with the aim to characterize aerosol properties along the vertical direction.

The ARCTAS mission (Jacob et al., 2010 and reference therein) showed highly-layered air pollution transport from North America and East Asia in spring, characterized by anthropogenic aerosol below 2 km and by biomass burning in the 2–4 km layer. The ARCPAC campaign (Brock et al. 2011) grouped the aerosol affecting the Arctic in spring in four categories: background troposphere (relatively diffuse, sulfate-rich aerosol); depleted aerosol within the surface inversion layer over sea-ice; layers of organic-rich biomass burning aerosol (above the top of the inversion layer) (see also Warneke et al., 2010) and layers dominated by fossil fuel combustion. The ASTAR campaign (Engvall et al., 2008), focussed on the spring to summer transition period in Svalbard, found Aitken and accumulation mode particles more concentrated in the free troposphere compared to the boundary layer. Kupiszewski et al. (2013), reported new particle formation events in the near-surface layer (possibly related to biological processes) during the summer ASCOS campaign.

Considering the BC, the springtime, PAM-ARCMIP (Stone et al., 2010) and HIPPO (Schwarz et al., 2010) campaigns showed high BC concentrations close to the ground, below the thermal inversion, but also dense pollution and BC at high altitudes over the Arctic (Wofsy et al., 2011). Interestingly, the PAM-ARCMIP results

show a decrease of BC compared to past measurements (i.e. AGASP, Hansen and Novakov, 1989). In addition, the HIPPO campaign revealed that in the lower troposphere the BC vertical gradient can change seasonally from positive to negative (Schwarz et al., 2013). In this respect, Spackman et al. (2010) and Koch et al. (2009) reported BC located mainly in the Arctic free troposphere with a positive gradient in the lower troposphere.

The aforementioned campaigns were conducted mainly using aircraft (or helicopters) that for their inner nature are limited to intensive observational periods (Kupiszewski et al., 2013;Bates et al., 2013; Spackman et al., 2010;Schwarz et al., 2010;Koch et al., 2009). Thus, aerosol vertical profiles in the Arctic appear scarse if compared with the number of available data collected at ground level (Samset et al., 2013; Koch et al., 2009). There is the need for regular vertical aerosol profiling campaigns to improve the description of a seasonally resolved aerosol

and BC vertical behavior.

In addition to this, aerosol vertical distribution could be affected in the future by changes in the aerosol emissions within the Arctic itself. The increasing of shipping emission in the Arctic is a good example. Shipping emissions inject the BC directly into the Arctic planetary boundary layer (probably warming the surface and depositing on snow and ice). The importance of the increasing shipping emission in the Arctic has been recently underlined

(Eckhardt et al., 2013; Corbett et al., 2010; Granier et al., 2006). Although the final impact is debated (Browse et al., 2013), the effective vertical distribution of these emissions has not yet been investigated.

Thus, there is a clear need to also improve the knowledge about aerosol vertical profiles in the Arctic during week-long campaigns along years to find common rules of behavior.

The Arctic site of Ny-Ålesund (Svalbard Islands) is particularly suitable for such measurements,  featuring long

term data series of ground based aerosol properties, lidar profiles, radiometric and meteorological data (Maturilli et al. 2015; Tunved et al., 2013; Di Liberto et al., 2012; Vihma et al., 2011; Hoffmann et al., 2009;Eleftheriadis et al., 2009; Eleftheriadis et al., 2004; Stock et al., 2012; Ström et al., 2003, 2009). Long-term upper-air observations by daily radiosondes provided an overview of the atmospheric vertical structure above Ny-Ålesund, inlcuding the Planetary Boundary Layer (PBL) altitude range (Maturilli and Kayser, 2016; Vihma et al., 2011). In this

climatological approach, stable atmospheric conditions, radiative surface-based inversion were frequently found during polar night conditions, indicating stable atmospheric conditions with suppressed vertical exchange. Once the snowmelt leads to considerable sensible and latent heat fluxes at the surface, atmospheric stratification becomes neutral or instable, allowing convection and vertical mixing.

These observations point towards the need to understand how the aerosol is vertically layered in function of the

meteorological changes along seasons. Despite this, as stated above, aerosol and BC measurements along vertical profiles are reported to be sparse, even recent UAVs applications could improve the available datasets (Bates et al., 2013).

Thus, this paper reports new data of aerosol and BC vertical profiles measured over Ny-Ålesund (Svalbard Islands) in two successive years (2011-2012) during an extensive field campaign (200 vertical profiles). Vertical profiles

measurements were conducted in the framework of the PRIN2009 "ARCTICA" project. The main part of the scientific activities at Ny-Ålesund was aimed at studying the chemical and physical properties of the aerosols and the long-range transport processes relevant for the measurements of organic and inorganic species at the site and along vertical profiles (Moroni et al., 2015; Udisti et al., 2013).

We describe first the sampling sites and the vertical profile measurements (section 2). Results and discussion

follow in section 3, with the conclusions in the final section 4.

## 2    Methodology

Tethered balloon soundings were carried out during spring 2011 and two summers 2011 and 2012 over Ny-Ålesund. The site is located at the Kongsfjorden, a fjord that develops in the north-west south-east direction. Northwards, Ny-Ålesund faces the sea, while a small chain of 400-500 m high mountains is located to the South (Figure1a).

Vertical profiles were measured from two sampling sites: during spring, the vertical profiles were taken at the Italian CNR Gruvebadet sampling site (78°55'03"N 11°53'40"E; Figure 1b) to assure a large distance to the Ny-Ålesund village. During summer, the tethered balloon measurements were operated at the German-French AWIPEV research base (78°55'24" N 11°55'15"E) to lie in the proximity of the Ny-Ålesund harbour (600 m) allowing the measurement of ship plume diffusion (section 3.3). Table 1 lists the dates of the campaign (25 measurement days), the number of flights (197 measured profiles), the maximum altitudes (~700-1300 m), and the cloud base height (clouds present for 48% of campaign). The aerosol and meteorological measurements were carried out both at ground and along the profiles as described in the following sections.

### 2.1    Ground-based measurements

Ground-based measurements were carried out at the Gruvenbadet laboratory (Figure 1b) where the distance (1.2 km southern Ny-Ålesund) and the limitations established for snow mobile traffic and other potentially contaminant activities limits the impact from local emissions.

The Gruvebadet laboratory is equipped with a series of instruments aimed at measuring aerosol physical and optical properties, and to collect samples for chemical analysis (section 2.1.1). The aerosol size distribution was measured using a Scanning Mobility Particle Sizer (TSI-SMPS 3034, 54 size classes, 10–487 nm) coupled with an Aerodynamic Particle Sizer (TSI-APS 3321, 52 classes, 0.5– 20 $\mu$m). The two coupled systems measure one size spectrum every 10 minutes (Giardi et al, 2016). PM samples were collected by high-volume and low-volume samplers. For the purpose of the present paper, $PM_{10}$ samples collected using two TECORA SkyPost low-volume sampler (EN 12341; $PM_{10}$ sampling head, flow 2.3 $m^3$ $h^{-1}$; PTFE and and quartz microfiber filters, Ø=47 mm) were considered. Sampling was carried out in ambient conditions: pressure and temperature were continuously monitored in order to maintain the constant flow rate of 2.3 $m^3$ $h^{-1}$. The first sampler collected $PM_{10}$ for 24 h on Teflon filters (Pall R2PJ047) to determine the ionic fraction, while the second one collected $PM_{10}$ for 96 h on pre-fired Quartz microfiber filters (chm QF1 grade) to determine organic and elemental carbon (section 2.1.1). The Teflon filters were conditioned for 48 hours (25°C and 50% relative humidity) before and after the sampling, then weighted by a 5-digit microbalance (Sartorius ME235P). The reproducibility error for filter weighing was lower than 5% (experimentally evaluated). After sampling, filters were individually sealed in pre-washed (with Milli-Q water, 18.3 MΩ cm) polystyrene filter containers and stored at – 20°C until analysis.

### 2.1.1    Aerosol chemistry measurements at ground level

$PM_{10}$ samples collected at ground-level at Gruvebadet were analyzed to determine first the water-soluble ionic fraction. Half of each $PM_{10}$ Teflon filter was extracted in 10 ml of ultrapure water (Milli-Q, 18.3 MΩ cm resistivity) by ultrasonic bath for 20 min. Filters manipulation was carried out under a class-100 laminar-flow hood, in order to minimize contamination risks. Inorganic cations and anions together with organic anions, were simultaneously

measured by a triple Dionex ion-chromatography system, equipped with electrochemical-suppressed conductivity detectors. Cations ($Na^+$, $NH_4^+$, $K^+$, $Mg^{2+}$ and $Ca^{2+}$) have been determined by a Dionex CS12A-4 mm analytical column with 20 mM $H_2SO_4$ eluent.Anions ($Cl^-$, $NO_3^-$, $SO_4^{2-}$ and $C_2O_4^{2-}$) were measured by a Dionex AS4A-4 mm analytical column with a 1.8 mM $Na_2CO_3$ / 1.7 mM $NaHCO_3$ eluent, while $F^-$ and some organic anions (acetate, glycolate, formate and methanesulfonate) were determined by a Dionex AS11 separation column by a gradient elution (0.075 mM to 2.5 mM $Na_2B_4O_7$ eluent) (Udisti et al., 2004; Becagli et al., 2011).

The detection limit (ng $m^{-3}$) of each analyzed chemical component is reported in the supplemental material (Table S1) together with the measured ambient ion concentrations. All the analyzed chemical components were largely above the detection limit.

The contribution of sea salt and crustal components in Ny-Ålesund is not-negligible (Udisti et al., 2016; Giardi et al., 2016; Moroni et al., 2015). Thus, $Na^+$, $Ca^{2+}$ and $SO_4^{2-}$(which originate from both these sources) were apportioned between sea-salt (ss-) and non-sea-salt (nss-) fractions on the basis of known w/w (weight/weight) ratios in sea water and Earth crust (Udisti et al., 2016; Giardi et al., 2016; Becagli et al., 2012; Udisti et al., 2012):

$$\text{tot-}Na^+ = \text{ss-}Na^+ + \text{nss-}Na^+ \tag{1}$$

$$\text{tot-}Ca^{2+} = \text{ss-}Ca^{2+} + \text{nss-}Ca^{2+} \tag{2}$$

$$\text{ss-}Na^+ = \text{tot-}Na^+ - 0.562 \text{ nss-}Ca^{2+} \tag{3}$$

$$\text{nss-}Ca^{2+} = \text{tot-}Ca^{2+} - 0.038 \text{ ss-}Na^+ \tag{4}$$

where 0.562 represents the w/w $Na^+/Ca^{2+}$ratio in the crust (Bowen, 1979) and 0.038 is the $Ca^{2+}/Na^+$ w/w ratio in seawater (Nozaki, 1997). Similarly, the ss-$SO_4^{2-}$ fraction was calculated from the ss-$Na^+$ using the 0.253 $SO_4^{2-}/Na^+$ w/w ratio in seawater (Bowen, 1979). The crustal fraction of sulfate (cr-$SO_4^{2-}$) was determined from the nss-$Ca^{2+}$ using the 0.59 $SO_4^{2-}/Ca^{2+}$ w/w ratio in the uppermost Earth crust (Wagenbach et al., 1996). Finally, the nss-nc-$SO_4^{2-}$ fraction, which can be due to anthropogenic or secondary formed aerosol, was calculated by subtracting the ss- $SO_4^{2-}$and cr-$SO_4^{2-}$contributions from the total $SO_4^{2-}$ concentrations.

The Organic Carbon (OC) and Elemental Carbon (EC) fractions were determined in $PM_{10}$ samples using a Thermo-Optical Transmission (TOT) method following the NIOSH protocol. The organic matter (OM) was calculated by multiplying the OC fraction by 2.1 (Turpin and Lim, 2001) typical for remote sites with a large fraction of secondary aerosols. Table S1 in supplemental material also reports the detection limit for EC and OC.

### 2.1.2    Meteorological Context

Meteorological parameters are currently measured at different sites in Ny-Alesund. The German-French AWIPEV research base operates surface meteorology measurements with 1-minute time resolution, including temperature and relative humidity at 2 m height, wind speed and direction at 10 m height, and pressure at station level close to the summer campaign balloon launch site (Maturilli et al., 2013). The cloud base height above the station is retrieved using a Vaisala LD-40 ceilometer. Daily radiosoundings (1100 UTC) by the AWIPEV observatory provide auxiliary data for the aerosol profile analysis.

The Italian National Research Council (CNR) operates since 2009 the Amundsen-Nobile Climate Change Tower (CCT), providing meteorological, micro-meteorological, radiation and snow measurements continuously all year-long (Mazzola et al. 2016). Conventional and micro-meteorological parameters are measured at different heights (4 and 3 levels, respectively) in order to investigate their vertical variations in different conditions. Dai et al. (2011) and Mazzola et al. (2015) found that both in the Adventfjorden and in Kongsfjorden, where Ny-Ålesund is located,

the atmosphere is stable for about 50% of the time along the year, by analyzing micro-meteorological data. The term stability refers to the propensity of air masses to move vertically: stable air resists any vertical motion, while unstable air masses are prone to vertical movements. A parcel of air results to be stable/unstable if the temperature lapse rate is lower/higher than the adiabatic one, i.e. if the potential temperature is increasing/decreasing with height, respectively. In stable stratification, turbulence and vertical mixing is suppressed, leading to trapping of pollutants near ground level.

On the above grounds, the spring 2011, summer 2011, and summer 2012 campaign periods can be put into a climatological context.

## 2.2    Vertical profile measurements

Vertical profile measurements have been carried out by means of a kytoon-shape helium-filled tethered balloon (lenght 8 m, $\varnothing$=3 m, volume 55.0 $m^3$, payload 25 kg, Figure 1c). The tethered balloon was designed to fly in severe wind conditions. However, the presence of the payload limits the balloon flights from low to moderate wind conditions (< 10 m $s^{-1}$).

The tethered balloon was equipped with an instrumental package consisting of:

1) an Optical Particle Counter (OPC GRIMM *1.107*; 31 size classes between 0.25 to 32 μm, 6 sec sampling time) for the particle number size distribution determination;

2) a miniaturized electrical particle detector (miniDiSC, Matter Aerosol) to measure the total particle number concentration (1 sec sampling time);

3) two micro-Aethalometers: the microAeth[®] AE51 and a prototype (1-60 sec sampling time);

4) a meteorological station (LSI-Lastem: pressure, temperature and relative humidity, 6 sec sampling time).

During the period 11/04/2011-30/04/2011, the Vaisala tethersonde TTS111 (pressure, temperature, relative humidity, wind speed and wind direction; 1 sec sampling time) was also used.

The maximum height reached during each flight depended on atmospheric conditions and, for the majority of the profiles, was between 0.7 and 1.3 km. An electric winch controlled the ascent/descent rates that were set at 40.0 ± 0.1 m/min.

A deeper description of each instrument is reported here below.

### 2.2.1    Size distribution data

In this study, the total aerosol concentration and the number size-distribution along height were measured using a coupled miniDiSC –OPC (λ=655 nm) system.

The miniDiSC is a miniature Diffusion Size Classifier, a small and portable instrument (4×9×18 cm, 670 g, 8h of battery supply) (Fierz et al., 2011). The aerosol is first charged in a standard positive unipolar diffusion charger (the average charge is approximately proportional to the particle diameter). The charged particles flow through a diffusion stage (an electrically insulated stack of stainless steel screens connected to a sensitive electrometer that collect the finest particles) and into a second stage (equipped with a HEPA filter) where the current of larger particles is measured with an electrometer. The miniDiSC has a $d_{50}$ cutoff at 14 nm. Thus, the instrument underestimates particle number concentrations for particles smaller than 20 nm (Nucleation Mode). As a result, the miniDiSC counts only partially the Nucleation Mode, while it allows a whole determination of Aitken and

Accumulation Mode particles. As demonstrated by Fierz et al. (2011), a bimodal lognormal aerosol size distribution with a fixed accumulation mode at 100 nm and a varying nucleation mode at 20 nm introduces an underestimation of about -2% – -10% of the total aerosol concentration in the miniDiSC response. The particle number determination is robust, and the error never exceeds 20%.

5    The OPC used in the campaign was the model the Grimm *1.107* that counts and classifies the aerosol in 31 size classes between 250 nm and 32 μm. As reported in literature (Ferrero et al., 2014; Howell et al., 2006; Heyder and Gebhart, 1979), OPCs size classification of the aerosol particles is a function of their ability to scatter the laser light under the assumption of spherical particles. The aerosol particles are classified in terms of their optical equivalent diameter, which is defined as "the diameter of a sphere of known refractive index (that of polystyrene latex spheres used for of calibration) that scatters light as efficiently as the real particle in question". This effect usually results in an "undersizing" of the size classification, due to the higher refractive index of the polystyrene latex spheres (PSL spheres, $m$=1.58 at 655 nm; Ma et al., 2003) used in the OPC calibration compared to ambient aerosol (Guyon et al., 2003; Liu and Daum, 2008; Schumann, 1990). In order to derive a proper size classification of the aerosol over Svalbard, the "undersizing" issue was solved by correcting the OPC size channels to account for the ambient aerosol refractive index $m$. The OPC response function ($S$: the partial light scattering cross section of the particle related to the specific optical design of the OPC) was computed at 655 nm as follows (Baron and Willeke, 2005; Heyder and Gebhart, 1979):

$$S(\theta_0, \Delta\Omega, x, m) = \frac{\lambda^2}{4\pi^2} \iint_{\Delta\Omega} i(\theta, \Phi, x, m) sin\theta d\theta d\Phi \tag{5}$$

where $\theta_0$ represents the mean scattering angle of the optical arrangement, $\Delta\Omega$ the receiver aperture, $x$ the dimensionless size parameter, $m$ the refractive index and $i(\theta,\phi,x,m)$ the Mie scattering function composed by the perpendicular and parallel components $i_1(\theta,x,m)$ and $i_2(\theta,x,m)$, respectively. The optical arrangement of the OPC 1.107 consists of: 1) a wide angle parabolic mirror (121°, from 29.5° to 150.5°, $\theta_0$=90°)that focuses scattered light on the photodetector located on the opposite side; 2) 18° of direct collected scattered light on the photodetector (from 81° to 99°, $\theta_0$=90°) (Heim et al., 2008).

25    The response function was calculated both for PSL spheres ($S_{PSL}$) and for ambient aerosol ($S_{AMB}$). The refractive indexes of ambient aerosol used in $S_{AMB}$ calculations were obtained from the closest AERONET site (Horsund site, 77°00'04" N 15°33'37" E) for spring 2011 and summer 2011-2012: 1.544+0.013i and 1.535+0.015i, respectively. These refractive indexes were determined at 674 nm (the closest AERONET wavelength to the OPC laser wavelength of 655 nm) and were close to those determined at 530 nm at Gruvebadet site (range 1.4-1.8 during 2010 and 2011; Lanconelli et al., 2013). Table S2 shows the new size corrected channels in comparison with the PSL spheres equivalent ones. The new channels were used to define three broad-size ranges (detailed here below) to evaluate the vertical behavior of aerosol.

The coupled miniDiSC–OPC (λ=655 nm) system measurement range covers the relevant region of the aerosol number size-distribution. In order to study the behavior of different size classes along height, three aerosol number concentration size ranges were selected:

1) the number concentration of aerosol between 14 nm ($d_{50}$ of miniDiSC) and 260 nm (cfr Table S2) obtained as the difference between the total number concentration measured by the miniDiSC and that measured by the OPC, hereinafter indicated as $N_{14-260}$;

2) the number concentration of aerosol between 260 nm (lower limit of OPC) and 1200 nm hereinafter indicated as $N_{260-1200}$;

3) the number concentration of aerosol above 1200 nm, hereinafter indicated as $N_{>1200}$.

The mode $N_{14-260}$ includes a small fraction of the Nucleation mode (from 14 to 20 nm), the totality of the Aitken mode (20-100 nm) and a fraction of the Accumulation mode (from 100to 260 nm). The mode $N_{260-1200}$ includes most of the Accumulation mode particles. Finally, mode $N_{>1200}$ covers the totality of Giant Nuclei mode.

The accuracy of both miniDiSC and OPC measurements was investigated comparing the lowermost portion of their measurements along vertical profile with SMPS+APS data collected at ground-level at Gruvebadet. This comparison was performed during spring 2011 to avoid any contamination from ship plumes arriving from Ny-Ålesund harbor towards Gruvebadet in summer (balloon sounding were conducted from the Koldeway station instead that from Gruvebadet; section 2.1). The comparison of $N_{14-260}$ (miniDiSC vs. SMPS) and of $N_{>260}$ (OPC vs. SMPS+APS) was characterized by an excellent correlation ($R^2>0.9$; linear best fit close to the ideal one) with an average error of 7% and 16% for both $N_{14-260}$ and $N_{>260}$, respectively (supplemental material, Figure S1a-b). These results highlight the reliability of measurements carried out along the vertical profiles, an important feature considering the low aerosol concentration values and their variation, which are present within the Arctic (section 1).

Number concentration data were also used in section 3.2.4 to estimate the contribution of locally formed aerosol. The method is based on the N/BC ratio, developed by Rodríguez and Cuevas (2007) and successfully applied in Europe by Reche et al. (2011). The basic concept of this method is that highest values of N/BC ratios (i.e. the lowest BC fraction values) occur during secondary aerosol formation in the atmosphere (Reche et al., 2011; Dall'Osto et al., 2013 and 2011). The methodology is as follows:

$$N_2=N-N_1 \tag{6}$$

$$N_1=S_1×BC \tag{7}$$

where $N_2$ represents the secondary aerosol concentration locally formed in the atmosphere, N is the measured aerosol number concentration and $N_1$ is the aerosol number concentration already present in the background air. $S_1$ represents a reference value for the N/BC ratio (expressed as particles $cm^{-3}$/ng $m^{-3}$ of BC) in the background air. The parameter $N_1$ is calculated from the parameter $S_1$ multiplied by the measured BC concentration (see next section 2.2.2). $S_1$ can vary from ~2 to ~9, while the N/BC ratio during secondary aerosol formation reaches values higher than ~15-20 and up to ~100-200 (Reche et al., 2011; Dall'Osto et al., 2013 and 2011). The differences in $S_1$ values determined in different sites can be caused by: 1) the use of different particle counters (with different $d_{50}$ cutoff), as lowest $S_1$ values are usually observed when devices with largest $d_{50}$ cutoff are used; 2) the influence of the ambient air conditions on the secondary aerosol formation. Thus, $S_1$ is site-instrument specific and has to be determined on-site depending on the used particle counter. If the Rodríguez and Cuevas (2007) method is applied to ground-based temporal data series, $S_1$ can be obtained as the minimum N vs. BC slope observed during the day (Reche et al., 2011). However, in the case of measured vertical profiles, the values of $S_1$ were taken as that of background aerosol above the aerosol stratifications described in section 3.2.4.

### 2.2.2    Black Carbon

BC have been determined using two micro-Aethalometers: the microAeth® AE51 and a prototype (Magee Scientific; 250 g, 117x66x38 $mm^3$). Adopting the nomenclature recommended by Petzold et al. (2013), by Andreae

and Gelencser (2006) and by other authors (Gilardoni et al., 2010; Hoffman et al., 2011; Sthol et al., 2013; Eckhardt et al., 2013), we refer to the measured parameter as "equivalent black carbon" (eBC) due to the absence of an overall agreed reference material, linking light absorption to the empirically defined BC mass concentration. In agreement with the above cited literature, we also report the absorption coefficient values.

5 AE51 and the prototype were identical with the exception that the prototype measured at 2-$\lambda$ (370 and 880 nm) while AE51 only at 880 nm. At the time of campaign the prototype was just yet developed and was used instead the AE51 on the balloon platform during the spring 2011 campaign only when necessary (i.e. AE51 in charge) to ensure the continuity of measurements during the campaign.

In both the Aethalometers the aerosol containing BC was continuously sampled onto a PTFE-coated borosilicate

10 glass fiber filter (Fiberfilm™ Filters, Pall Corporation) where the light attenuation ($ATN$) was measured at 880 nm relative to a clean part of the filter. $ATN$ was calculated as:

$$ATN=100*\ln(I_0/I) \tag{8}$$

where $I_0$ and $I$ are the light intensities transmitted throughout a reference blank spot and the aerosol-laden 3 mm diameter sample spot of the filter, respectively.

15 The attenuation coefficient of the particles collected on the filters, $b_{ATN}$, was derived from $ATN$ as follows (Weingartner et al., 2003):

$$b_{ATN} = \frac{A}{100Q}\frac{\Delta ATN}{\Delta t} \tag{9}$$

where $\Delta ATN$ indicates the $ATN$ variation during the time period $\Delta t$, $A$ is the sample spot area ($7.1 \cdot 10^{-6}$ m$^2$) and $Q$ is the volumetric flow rate ($2.5 \cdot 10^{-6}$ m$^3$ sec$^{-1}$ for the AE51 and $4.42 \cdot 10^{-6}$ m$^3$ sec$^{-1}$ for the prototype).

20 Finally, to determine the eBC ambient concentration the apparent mass attenuation cross-section ($\sigma_{ATN}$ = 12.5 m$^2$ g$^{-1}$) is needed; it is defined for the eBC collected on the PTFE-coated borosilicate glass fiber filter. The $\sigma_{ATN}$ value (12.5 m$^2$ g$^{-1}$) was obtained by comparing the eBC values measured with the microAeth® Model AE51, with an AE31 Aethalometer (880 nm wavelength) operating in a test chamber with different eBC concentrations at low attenuation values. The comparison was then repeated using ambient air (Ferrero et al., 2011a). This value is not

25 far from the $\sigma_{ATN}$ values of 15.2 m$^2$ g$^{-1}$ and 15.9 m$^2$ g$^{-1}$ reported in Eleftheriadis et al. (2009) when reported ten years of eBC measurements in Ny-Ålesund at the Zeppelin station with the Aethalometers AE9 and AE31. The difference between these values results from the use of different filter materials to collect the sample in the different Aethalometers, which was quantified in Ferrero et al. (2011a) and Drinovec et al (2015).

The eBC concentrations were determined as follows:

30 $$eBC = \frac{b_{ATN}}{\sigma_{ATN}} \tag{10}$$

The accuracy of eBC measurement was investigated. The AE51 and the prototype measurements carried out simultaneously agreed very well ($R^2$=0.852; slope=0.976; Figure S1c supplemental material). This result was important as obtained with two different flowrates ($2.5 \cdot 10^{-6}$ m$^3$ sec$^{-1}$ for the AE51, and $4.42 \cdot 10^{-6}$ m$^3$ sec$^{-1}$ for the prototype).

35 However, a large scatter is present at low eBC concentrations (i.e. 10-20 ng m$^{-3}$; see Figure S2c). Thus, the absolute error (in percentage) of each eBC value (considering the average of the two Aethalometers) was calculated for intervals of 5 ng m$^{-3}$ of concentrations. At low concentrations, the error can reach up to 90% and more (Figure S2a, supplemental material). This error decreases with increasing concentration, dropping below 20 ng m$^{-3}$ for eBC concentrations above 5 ng m$^{-3}$. The relative error lies below 20% for both the average and the 90[th] Percentile

at eBC concentrations above 20 ng m$^{-3}$. Thus, it is possible to consider this value as the limit above which a single eBC measurement point is not affected by instrumental noise. Nevertheless, this limit is close to the BC concentrations that have been previously measured in the Arctic (Eleftheriadis et al., 2009). In this respect, we note that the BC profiles presented in the manuscript are an average of many measurements, hence the effect of the noise on the reported eBC concentrations is further reduced. The aim of this paper is to determine the seasonal phenomenology of the aerosol behavior along vertical profiles classifying the collected experimental data, according to their shape and averaging them for each season. This is very important as, even the error in percentage of each data point can reach high values (especially at low concentrations), the average of the data stabilizes the instrumental fluctuations. This effect is demonstrated by Figure S2b (supplemental material) which reports the correlation between the BC concentrations (AE51 and prototype) averaged on the same intervals of 5 ng m$^{-3}$ used in Figure S2a ($R^2$=0.986; slope=1.017).

The above reported analysis underlines a critical situation for summer because, as reported in Eleftheriadis et al. (2009), the eBC concentration range expected in summer is ~0-10 ng m$^{-3}$. Therefore, summer eBC data were used here only to highlight the impact of shipping emissions on the Arctic background concentrations along the atmospheric column. Due to high ship impact (section 3.4), the performance of the micro-Aethalometers was suitable and reliable for the purpose of this application.

In addition to eBC, the micro-Aethalometers allows also the determination of the aerosol absorption coefficient, $b_{abs}$, that was calculated as follows:

$$b_{abs} = \frac{b_{ATN}}{C \cdot R(ATN)} \tag{11}$$

where $C$ and $R(ATN)$ are the multiple scattering optical enhancement factor and the aerosol loading factor, respectively. Briefly, the constant optical enhancement factor $C$ compensates for the enhanced optical path through the filter caused by multiple scattering induced by the filter fibers themselves (Schmid et al., 2006; Arnott et al., 2005, Weingartner et al., 2003). The parameter $R(ATN)$ compensates for the nonlinearity – the loading effect due to reduction of the measurement sensitivity due to the saturation caused by the collected sample on the filter. The compensation with the parameter $R(ATN)$ is needed only when $ATN$ becomes higher than 20 (Schmid et al., 2006; Arnott et al., 2005; Weingartner et al., 2003). In this study, the experimental design allowed us to neglect the use of $R(ATN)$: all eBC vertical profiles were conducted in the clean Arctic environment and the filter tickets were changed regularly to always keep ATN lower than 20 as recommended by Weingartner et al. (2003). For the AE51, and the prototype, the only parameter C available in the literature is 2.05 ± 0.03 (at λ = 880 nm) (Ferrero et al., 2011a), even though recently Ran et al. (2016) proposed a C value of 2.52 for ground-based measurements in China.

The C value of 2.05 ± 0.03 was determined over Milan in Ferrero et al. (2011a) and thus a brief description is necessary to determine its applicability in the Arctic area. The parameter C was determined using data collected both below the mixing layer and above it, in a cleaner atmosphere, along the vertical profiles (Ferrero et al., 2011a). During the C determination, a new filter ticket was used for each profile. As a result, ATN never reached values higher than 20 (average ATN was 5±1) and the total amount of aerosol collected on each filter during the C determination was negligible. Therefore, the determined C was exclusively dependent on the filter material and the AE51 instrumental geometry. This ensures the negligible influence of the particles in the filter matrix on the

C value. The reliability of the obtained C (2.05±0.03) was demonstrated in Ferrero et al. (2014) below the mixing layer and in free troposphere.

Finally, it should be noted that absorbing non-BC particles may contribute to the signal in Aethalometers (i.e. Brown Carbon, dust). However, BrC is characterized by negligible absorption in the infrared (Andreae and Gelencsér, 2006), the wavelength range of the eBC measurements (micro-Aeth AE51 uses 880 nm). In this respect, Massabò et al. (2013) showed the potential contribution of BrC to the determination of eBC to be below 10%.

To estimate the possible influence of BrC on eBC measurements carried out during the spring 2011 campaign, the data collected with the micro-Aeth prototype at 370 and 880 nm were considered. They highlighted a BrC positive artifact on eBC measurements less than 10% during the campaign. Details are reported in supplemental material.

### 2.2.3  Meteorological data and aerosol stratifications

Meteorological data along height allowed the determination of the absolute height of the balloon using the hypsometric equation; due to change during April 2011 in the measuring system (section 2.2), a comparison of the altitude obtained by the LSI-Lastem and Vaisala tethersonde was conducted during several target flights. The result ($R^2$=0.997; slope=0.999; Figure S1d, supplemental material) demonstrated the accuracy of the height determination. The measurement of altitude is fundamental in the study of the vertical aerosol properties in relationship with meteorological parameters. In fact, vertical aerosol profiles allows the determination of the height of aerosol stratifications by means of a gradient method, applied to aerosol concentration profiles, as suggested by Seibert et al. (2000).

The gradient method is based on the determination of the minimum value of the vertical derivative of the aerosol concentration.  The use of gradient method to determine the aerosol mixing height has been demonstrated at lower latitudes in previous works (Ferrero et al., 2012, 2011a, 2011b and 2007; Sangiorgi et al., 2011; Di Liberto et al., 2012). However, in remote areas, such as the Arctic, several processes other than dispersion can shape the aerosol profiles. The two most important ones are: 1) differential advection (Tunved et al., 2013) and 2) a lack of emission of aerosol from ground. These processes should generate a vertical structure not directly related to the PBL height. Therefore, in the present work the gradient method has been limited to individuate aerosol stratification heights ($AS_h$), even if these related to the behaviour of meteorological variables governing the behaviour of the PBL as will be addressed in section 3.1.

The $AS_h$ will be used in the next sections to calculate averaged aerosol and eBC profiles (sections 3.2 and 3.3). In fact, in order to investigate the variation of aerosol properties with height, vertical profiles were statistically averaged. As reported in previous works (Ferrero et al., 2011a, 2012 and 2014), a way to average vertical profile data by taking their main gradients ($AS_h$) into account, is to consider the relative position of each measured data point in respect to the $AS_h$. Thus, vertical profiles were first normalized, introducing a standardized height ($H_s$) calculated as follows:

$$H_s = \frac{z - AS_h}{AS_h} \tag{12}$$

where z is the height above ground. $H_s$ assumes a value of 0 at the $AS_h$, and values of -1 and 1 at ground-level and at twice the $AS_h$, respectively.

Examples of $AS_h$, accompanied with the corresponding potential temperature ($\theta$) and RH profiles, are presented in Figure 2a-d. The presented data, accurately describe the vertical distribution of the aerosol and its properties in the first kilometer above Ny-Ålesund. Moreover, they allowed to obtain different piece of information.

The absence or the presence of marked aerosol stratifications (AS) is notable. When present, the altitude at which they occur ($AS_h$) was determined by the gradient method, described above. This is the first obtainable information. A second piece of information was the size dependent vertical behavior of aerosol concentrations. Figures 2b and 2c highlight a similar behavior for both $N_{14-260}$ and $N_{260-1200}$, while, Figure 2d shows different behavior for $N_{260-1200}$ due to a concentration change located at a different altitude. Finally, we obtained the magnitude of the observed concentration change at each $AS_h$ for each size range ($N_{14-260}$, $N_{260-1200}$ and $N_{>1200}$) and season.

The analysis and combination of these three types of information allowed to classify, as a function of seasonality, the altitude, magnitude and frequency of aerosol stratifications. Furthermore, it has been possible to shed some light on the dynamics underlying the seasonal phenomenology found during the field campaigns. A detailed discussion on these points is reported in the following section.

## 3    Results and Discussion

Vertical profiles of aerosol number size distribution and eBC concentrations were measured to assess changes in aerosol properties within the vertical column in the Arctic region. The results obtained along vertical profiles are discussed in order to highlight first the vertical behavior of the $AS_h$ in relation to the main atmospheric meteorological parameters (section 3.1). Then, vertical aerosol properties are discussed in details for springtime (section 3.2) and summertime (section 3.3). All averaged data are reported hereinafter as mean ± mean standard deviation.

### 3.1    Aerosol stratifications: seasonal vertical frequency distribution and relationship with meteorology

As reported in Table 1, about 200 profiles were measured during 3 campaigns in spring 2011, summer 2011 and summer 2012. Here below, the ambient conditions under which the vertical profiles were measured are briefly described.

First of all, the observational periods (spring 2011, summer 2011 and summer 2012) were addressed in a climatological context. In this respect, the temperature measured in spring 2011 was within the standard deviation range of the long-term observations, while a 10-day period at the end of April 2011 was slightly warmer than the climatological mean (Figure S3a). The temperatures during the summer seasons 2011 and 2012 were mostly within the range of the long-term observations (Figure S3b). Neither of the campaign periods was conducted under exceptional meteorological conditions, so the vertical profile measurements can be considered to have been obtained under typical meteorological conditions representative for the Ny-Ålesund environment.

We note, however, that the tethered balloon measurements have limitations with respect to its launch conditions (section 2.2). Particularly, balloon profiles were measured in low wind conditions, as it is very difficult to launch the balloons during high winds. This introduces a bias in respect to average meteorological conditions above the launch site. The maximum wind speed measured at the Amundsen-Nobile Climate Change Tower (section 2.1.2) during balloon flights was lower than that during the whole period of the campaign (April 2011, June and July 2011-2012): 4.9 m s$^{-1}$ and 10.7 m s$^{-1}$ (springtime and summertime balloon profiles) compared to 27.9 m s$^{-1}$ and 16.3 m s$^{-1}$ (full spring 2011 and summer 2011-2012). Table 1 resumes the conditions for all the measured profiles.

The majority of vertical profile measurements was conducted under clear sky conditions (no clouds) or with clouds with base height above the balloon payload.

Thus, it is possible to assert that the measured profiles, and the seasonal phenomenology described hereinafter, are representative of typical Arctic springtime and summertime periods mainly for low wind and clear sky conditions.

Figure 2a-d highlights different atmospheric dispersal conditions upon Ny-Ålesund. Although these four case studies are not illustrative and comprehensive of the whole data set, their discussion helps to illustrate the seasonal and size-dependent behavior of the frequency distribution of the $AS_h$ with altitude.

An example for homogeneous dispersion of aerosol (independent from its size) in the lower troposphere is shown in Figure 2a. Decreasing potential temperature with height indicates atmospheric instability, allowing the vertical mixing of air masses by convection. Homogeneous aerosol profiles with convective conditions were found in 15% of the profiles in spring and 37% of the profiles in summer, respectively. Convective conditions generally are observed more frequent during summer in Ny-Alesund related to the different level of radiation energy at disposal and surface properties. In summer homogenous profiles were observed often (37%) than in spring (15%), due to a synergy of the higher solar power density at disposal ($186.4\pm71.2$ W m$^{-2}$ in summer and $109.2\pm35.9$ W m$^{-2}$ in spring) together with a lower albedo ($0.15\pm0.01$ in summer and $0.87\pm0.04$ in spring) induced by the summer snowmelt in Svalbard (Mazzola et al. 2015). The resulting change in surface energy balance affects the atmospheric stability, from the more stable conditions and inversion situations with snow-cover to the unstable conditions favoring mixing within the boundary layer once the snow cover has disappeared.

In spring, the presence of different layers of aerosol, separated by abrupt changes in the aerosol properties (i.e. concentration) along height were observed more frequently than the homogeneous mixing conditions (Figure 2b-d). The presence of aerosol stratification was complementary to the homogenous profiles and thus occurred for 85% and 63% of cases during spring and summer, respectively.

Within stratified conditions it was possible to determine the $AS_h$ for each aerosol size and season together with the altitude of sharp changes observed also for $\theta$ and RH. For example, Figure 2c shows $AS_h$ at 630 m in agreement with the vertical gradients of $\theta$ and RH. On the other hand, Figure 2d shows a first $AS_h$ at 134 m only for $N_{14-260}$, while $N_{260-1200}$ appeared homogenously distributed until a second $AS_h$ located at 674 m. The first $AS_h$ detected for the smallest particles was related to a ground-based $\theta$ inversion, while the second $AS_h$, detected for the accumulation mode particles, was related to an elevated $\theta$ inversion.

The aforementioned case studies helped us to introduce the description of the $AS_h$ frequency distribution with altitude and season. The $AS_h$ were used independently from the sign of the aerosol concentration change (either positive or negative; Figure 2b-c) and were computed separately for each broad size range (i.e. Figure 2d). The resulting frequency distribution with altitude of the first $AS_h$ is reported in Figures 3a-d and Figures 3e-h for spring and summer, respectively. It has to be underlined that sometimes it was possible to detect up to two $AS_h$ for each profile. This situation was observed for ~30% of profiles (characterized by the presence of aerosol gradients) in spring and summer, due to the limited maximum altitude reached by the balloon during the flight (usually between 0.7 and 1.3 km). The behavior of the second $AS_h$ is shown in the supplemental material (Figure S4) to support the description of the behavior of the first aerosol stratification with altitude.

Focusing on each season and considering first the springtime $AS_h$ vertical frequency distributions (Figure 3a-d) a common behavior can be first observed for both the three size ranges ($N_{14-260}$, $N_{260-1200}$ and $N_{>1200}$) and meteorological parameters ($\theta$ and RH): all of them showed a bimodal distribution characterized by a minimum

within the ~400-500 m height range. Maturilli and Kayser (2016) identified a frequent occurrence of a temperature inversion layer in the shear zone above the mountain ridges; this phenomenon is typically present throughout the year, leading to a decoupling of the lowermost kilometer of the atmosphere from the free troposphere above. In between the mountains, the atmosphere is characterized by wind channeling along the fjord axis, disturbed by e.g. glacier outflow or land-sea breeze. Thus, the observed separation of $AS_h$ most likely relates to the separation of the atmospheric flow.

Moving forward, some differences were then found in the behavior of each aerosol size range below and above the minimum at 400-500 m. While $N_{260-1200}$ and $N_{>1200}$ (Figure 3b-c) appeared equally distributed below and above the minimum at 400-500 m, $N_{14-260}$ showed highest frequencies of $AS_h$ below 400 m (Figure 3a). Particularly, the 82% of $AS_h$ for $N_{14-260}$ were located below 400 m and showed a clear maximum peak close to the ground (0-100 m) with a frequency of 38%. Moreover, the average values of $AS_h$ for $N_{14-260}$ below and above the minimum at 400-500 m were 143±13 m and 669±32 m, respectively. Conversely, $AS_h$ for $N_{260-1200}$ and $N_{>1200}$ occurred for 52% and 51% below 400 m peaking in the 100-200 m range (22% and 21%, respectively) with average values of 208±19 m and 177±14 m, respectively. Above 400 m the $AS_h$ for $N_{260-1200}$ and $N_{>1200}$ peaked in the 600-700 m range (22% and 19%, respectively) with average values of 672±16 m and 652±20 m.

The observed lack of symmetry between $N_{14-260}$ and $N_{260-1200} - N_{>1200}$ is explained in Figure 2d, where a decoupled trend for $N_{14-260}$ and $N_{260-1200}$ is shown. As stated above, the behavior of smallest particles was in that case related to a ground-based θ inversion. Considering the whole data set, the behavior of the vertical frequency distribution of the first gradient of both θ and RH was in agreement with that of $AS_h$ for $N_{14-260}$. In this respect, a maximum peak for both θ and RH was found close to the ground (0-100 m), with a frequency of 49% and 38%, respectively (average values below the minimum at 400-500 m for gradients for θ and RH of 117±12 m and 669±32 m). Interestingly, the frequency of ground-based θ inversions (49%) was higher than that of $AS_h$ for $N_{14-260}$ (38%). This feature is due to vertical profile along which, even in the presence of a ground-based θ inversion, $N_{14-260}$ did not show any variation of concentration; an example is reported in Figure 2b. Thus, the presence of a ground-based θ inversion appears as a necessary but not sufficient condition to observe the aforementioned behavior (resumed by Figure 2d, Figure 3a and 3d). The phenomenology and the aerosol dynamic responsible of this behavior (together with that of $N_{260-1200} - N_{>1200}$) will be addressed and discussed in the following section 3.3.

We describe below the summer $AS_h$ behavior. Figure 3e-h (and Figure S4) show that even in summer, the multi-layered structure persisted and was also characterized by a bimodal distribution, as in spring, but with a higher minimum (than in spring) that ranged approximately between 500 m and 600 m. The summer $AS_h$ for all size ranges and the gradient for θ and RH peaked between 100 and 300 m: 78% ($N_{14-260}$; average value 276±19 m), 71% ($N_{260-1200}$; average value 269±18 m), 76% ($N_{>1200}$; average value 272±18 m), 83% (θ; average value 262±18 m) and 79% (RH; average value 268±18 m). Figure 3e-h, and the aforementioned data, highlight that the vertical frequency distribution for $AS_h$ of all sizes and gradients for θ and RH behave similarly. This phenomenon, different from that observed in spring, will be addressed and discussed in section 3.3. As a final conclusion, a multi-layered structure was found over Ny-Ålesund both in spring and summer (see also Figure S4), and the most important atmospheric thermodynamic parameters (θ and RH) indicated the role of meteorology in shaping the aerosol vertical profiles. This result is of great importance as the majority of the aerosol measurements conducted in the Arctic area is ground-based and thus, it is necessary to understand their validity with altitude.

### 3.2 Springtime phenomenology

The previous section introduced the vertical behavior of sized aerosol in terms of frequency distribution of $AS_h$. However, it is also necessary to describe the intensity of the aerosol concentration changes at the $AS_h$, and the possible dynamics underlying these changes. Thus in this section, the springtime vertical aerosol phenomenology will be investigated. All the profiles measured in spring 2011 were classified based on their vertical behavior (i.e. shape) and averaged considering the relative position of each measured data point with respect to the $AS_h$. The obtained averaged vertical profiles were referred to a standardized height ($H_s$, Eq. 12) as described in section 2.2.3. As the size classes can behave differently with height, $H_s=0$ was referred to that observed for the intermediate $N_{260-1200}$ size class. The result of the classification and averaging procedure is reported in Figure 4a-m (all profile data are reported in Figure S5, supplemental material). Four main typologies of vertical profile were found. According to their shape they were named as follows:

1) Type 1, homogeneous profiles (hereinafter addressed as HO), Figure 4a-c

2) Type 2, profiles characterized by a positive gradient at $H_s=0$ (hereinafter addressed as PG), Figure 4d-f

3) Type 3, profiles characterized by a negative gradient at $H_s=0$ (hereinafter addressed as NG), Figure 4g-i

4) Type 4, profiles characterized by negative gradients located ad different altitude in function of size (hereinafter addressed as decoupled negative gradient, DNG), Figure 4l-m.

Average concentrations of aerosol ($N_{14-260}$, $N_{260-1200}$, $N_{>1200}$) and eBC below and above $H_s=0$ for each profile class are summarized in Table 3.

We first report here the columnar averages of both total aerosol number and eBC concentrations obtained by averaging all the aforementioned profile classes: $236.1\pm23.9$ cm$^{-3}$ ($N_{14-260}$), $21.1\pm1.3$ cm$^{-3}$ ($N_{260-1200}$), $0.2\pm4*10^{-2}$ cm$^{-3}$ ($N_{>1200}$) and $52\pm8$ ng m$^{-3}$ (eBC). They perfectly agree with long-term data series collected over Ny-Ålesund at the Zeppelin observatory (Eleftheriadis et al., 2009; Tunved et al., 2013) during Spring (~100-250 cm$^{-3}$ and 50–70 ng m$^{-3}$ of eBC during April). This agreement indicates that all the profile classes discussed below can be considered to be characteristic (with their occurring frequencies and altitudes) for the background Arctic aerosols measured by Arctic observatories within GAW, AMAP and EMEP observation programs. Moreover, the eBC data also agreed with results from PAM-ARCMIP (Stone et al., 2010) which showed a 40-90 ng m$^{-3}$ range of eBC within the surface inversion layer and 30–50 ng m$^{-3}$ above.

All the CCT wind data were used to compute wind rose graphs timely coincident with each profile typology (Figure 5a-d) and will be used in the following sections. The fjord direction into which the wind is often channeled is NW-SE. Here we underline that Figure 5a-d shows the absence of wind from north during the profile measurements, thus any influence from the Ny-Ålesund village is negligible. In addition, Figure 6 shows the ground-based number size distribution measured at Gruvebadet (section 2.1) for HO, PG, NG and DNG profiles, respectively.

Finally, a brief discussion of the air masses origin for the four categories is summarized in the supplemental material (see also Figure S6).

### 3.2.1 Homogeneous Profiles (HO)

HO profiles (Type1, Figure 4a-c) were observed in 15% of cases (during 7 days) and were characterized by a homogenous vertical distribution of aerosol and eBC upon Ny-Ålesund. HO profiles are reported with an absolute

height AGL, because they did not show any $AS_h$ to calculate $H_s$. They appear in some way analogous to the relatively diffuse background aerosol reported in the springtime ARCPAC campaign (Brock et al. 2011). During HO profiles, local wind (Figure 5a) was blowing mainly from the SW direction, from the glaciers behind Ny-Ålesund (Figure 1a) and not along the predominant NW-SE direction. Moreover, HO profiles featured the lowest wind speed (range 0-2 m s$^{-1}$, average of 0.6±0.1 m s$^{-1}$ at 33 m; Table 2).

At the same time, as shown in Figure 4c, a slightly positive θ profiles characterized, on average, the HO profiles. To this average contributed θ profiles with both positive and negative (as shown in Figure 2a) vertical gradients Negative θ gradients allowed vertical mixing. Positive θ gradients instead favored stable conditions. However, they are not in contrast with the absence of an aerosol stratification. In fact, just the presence of an important aerosol source (either local or transported) allow the formation of a distinct aerosol layer. This process is detailed in section 3.2.3.

The number concentrations in HO profiles were 80.2±16.4 cm$^{-3}$ (81.9±16.8% of the total aerosol concentration), 17.5±2.0 cm$^{-3}$ (17.9±2.0% of the total aerosol concentration) and 0.2±0.1 cm$^{-3}$ (0.2±0.1% of the total aerosol concentration) for $N_{14-260}$, $N_{260-1200}$, $N_{>1200}$, respectively. The aerosol number concentration was thus dominated by the $N_{14-260}$ size fraction along the whole profile. eBC and the related $b_{abs}$ (section 2.2.2, eq. 4) reached values of 35±21 ng m$^{-3}$ and 0.22±0.13 Mm$^{-1}$.

Aerosol number concentration in HO profiles lies close to the lower values registered at Zeppelin in April (Tunved et al., 2013) and were characterized by a ground-based size distribution dominated by the accumulation mode particles (Figure 6). eBC was close to the 25°-50° percentiles reported in Eleftheriadis et al. (2009a) for April and to the lower troposphere value of refractory BC (rBC) measured in spring during the HIPPO campaign (Schwarz et al., 2013).

Within HO profiles the aerosol pollution, previously transported from mid-latitudes, affected the first km of the atmosphere. The observed homogenous mixing conditions, allowed us to assume that the aerosol properties measured at ground-level in Ny-Ålesund were representative for the lower troposphere.

### 3.2.2    Positive Gradient Profiles (PG)

PG profiles (Type 2, Figure 4d-f) occurred in 17% of cases (during 6 days) and were characterized by an increase of aerosol number concentrations above $H_s$=0 and moderate eBC concentrations (24±3 ng m$^{-3}$; $b_{abs}$ was 0.15±0.02 Mm$^{-1}$). The average value of $AS_h$ (corresponding to $H_s$=0) was 417±266 m. During PG profiles local wind (Figure 5b) was blowing mainly from the SE along the predominant NW-SE direction. This situation is common in Kongsfjiorden (Vihma et al., 2011). PG profiles featured the highest wind speed (range 0-5 m s$^{-1}$, average of 2.3±0.1 m s$^{-1}$ at 33 m; Table 2). At the same time, as shown in Figure 4f, a positive θ profiles was present with a +1.5±0.4 K increase from $H_s$=0. A stable atmosphere was present and the aerosol was brought to the site by long-range transport in this stable situation. The increment of aerosol number concentrations with altitude was particularly evident for $N_{14-260}$ that increased by +171.5±25.4% (going from 205.4±12.5 cm$^{-3}$ below the $AS_h$ to 557.6±45.9 cm$^{-3}$ above it) while, $N_{260-1200}$ experienced a more modest increase of 11.8±7.0% (going from 19.9±0.2 cm$^{-3}$ below the $AS_h$ to 22.3±1.4 cm$^{-3}$ above it). Conversely, the coarse fraction ($N_{>1200}$) decreased with altitude of -38.1±10.0% (going from 0.26±0.02 cm$^{-3}$ below the $AS_h$ to 0.16±0.02 cm$^{-3}$ above it). The observed increase of Aitken and Accumulation mode fractions ($N_{14-260}$ plus $N_{260-1200}$), and the corresponding decrease of the coarse fraction ($N_{>1200}$), appear to be in agreement with the observation that during transport events wet removal processes

and dry deposition decrease the coarse particle concentration by scavenging and, at the same time, establish conditions that favor secondary aerosol formation due to the lowering of the condensational sink (Tunved et al., 2013).

The aerosol number concentration values above $H_s$=0 were close to those reported in Engvall et al. (2008) for the Arctic free troposphere during ASTAR. The ground-based size distribution, not influenced by the pollution layer at high altitude, was dominated by accumulation mode particles as in HO profiles (Figure 6).

The PG profile data suggested that high altitude transport events could be the origin of this type of profiles during springtime. An example of this process is the interesting case study of 23[th] April 2011 (1200-1330 UTC), when an intense plume of aerosol was transported over Ny-Ålesund. Figure 7a-b shows this event with the associated air mass back trajectories, and the time evolution of the event obtained through the interpolation of 6 vertical profiles, each of which lasted ~15 min and was removed about 1 min from the following profile.

Altitude layers of pollution were documented in Brock et al. (2011) and in Kupiszewski et al. (2013) and Wofsy et al. (2011). Jacob et al. (2010) reported that transport from North America and East Asia takes place mainly at higher altitudes, as also documented in Stohl et al. (2006). In this respect, the back-trajectories reported in Figure 6a described high altitude air from North America and Asia that descended in the Arctic. High aerosol concentrations at high altitude are important because aerosols can act as CCN and thus impact on climate via the indirect aerosol effects.

### 3.2.3    Negative Gradient Profiles (NG)

NG profiles (Figure 4g-i) were observed in 48% of all cases (during 9 days) making the NG the dominant typology of profiles. The average value of $AS_h$ (corresponding to $H_s$=0) was 506±212 m. The predominant wind direction was the same as in PG profiles (SE-E direction) with a component also from SW (Figure 5c). Figure 4i shows that a strong, positive θ profile (+4.1±0.3 K increase from $H_s$=0) characterized NG profiles.

Within this condition, the case study reported in figure 7b showed the origin of NG profiles. Figure 7b shows first a transport event that generated PG profiles (section 3.2.2). Afterwards, the transported aerosol was mixed downward within the PBL until ground. Most important, at the end of the process (1330 UTC) a negative concentration gradient with altitude was established generating a NG profile.

As now shown, NG profiles might be originated from the entrance of Arctic Haze into the PBL after a transport process. In the Arctic, in absence of an important local aerosol source (i.e. nucleation which acts mainly in summer; Tunved et al., 2013), only transported aerosol trapped within a thermal inversion made possible the presence of this typology of profiles. The presence of an intense θ inversion stabilizes the situation maintaining a NG typology of profile since vertical mixing is prevented (see also Figure 2c). It has to be noticed that an intense θ inversion is just a necessary condition to promote the formation of NG profiles, not a sufficient one. This result explains the presence of HO profiles even in a stable atmosphere (section 3.2.1) and is in agreement with the observation (reported in section 3.1.1) that the frequency of ground-based θ inversions (49%, Figure 3d) reached values higher than those for any $AS_h$ for any aerosol size (Figure 3a-c).

NG profiles were characterized by high pollution levels below $H_s$=0 where an intense decrease of both aerosol and eBC was observed. Crossing the $AS_h$, aerosol concentrations decreased by -52.9±8.7% (from 252.3±17.5cm$^{-3}$ to 118.9±9.3cm$^{-3}$) for $N_{14-260}$, by -57.9±2.6% (from 23.1±0.4cm$^{-3}$ to 9.7±0.3 cm$^{-3}$) for $N_{260-1200}$ and by -66.5±11.5% (from 0.53±0.05cm$^{-3}$ to 0.18±0.02cm$^{-3}$) for $N_{>1200}$. eBC behaved similarly, decreasing by -50.4±6.8% (from 71±4

ng m$^{-3}$ to 35±2 ng m$^{-3}$), with the same phenomenology for $b_{abs}$ (from 0.43±0.02Mm$^{-1}$ to 0.21±0.01Mm$^{-1}$). The last finding is very important because the altitude of eBC occurrence in the atmosphere modulates its influence on the climate in the Arctic.

NG profiles exhibited characteristics in agreement with literature data. Focusing on eBC, the vertical behavior and the observed concentrations agreed with those found in the PAM-ARCMIP campaign. Stone et al. (2010) reported rBC concentrations of 40-90 ng m$^{-3}$ within the surface-based temperature inversion layer, decreasing to 30–50 ng m$^{-3}$ above it. Results reported in Schwarz et al. (2013) for the HIPPO campaign in January agree with our measurements. Aerosol number concentration and eBC are also close to the higher values registered at the Zeppelin station in April (Tunved et al., 2013; Eleftheriadis et al., 2009a). In fact, NG profiles represent the most polluted situation affecting the whole boundary layer, quite the opposite to HO profiles.

### 3.2.4    Decoupled Negative Gradient Profiles (DNG)

A particular kind of profiles characterized by a decrease in concentration with altitude is the DNG typology. Within this class, observed in 20% of cases (during 5 days), a lack of symmetry between $N_{14-260}$ and $N_{260-1200}$– $N_{>1200}$ was observed (Figure 4l-n). The average value of $AS_h$ (corresponding to $H_s$=0) was 585±90 m. The main wind direction was the same as in PG profiles (SE-E direction; Figure 5d) but with a lower wind speed (close to that of HO profiles; 0-2 m s$^{-1}$; average of 0.7±0.1 m s$^{-1}$ at 33 m).

Figure 4n shows two strong, positive θ inversions. The first one, ground based, resulted in +1.3±0.4 K increase from ground; the second one characterized by +1.1±0.2 K increase from $H_s$=0.

Within this condition, $N_{14-260}$ showed a concentration peak close to the ground of 601.3±19.9 cm$^{-3}$ that was not present for $N_{260-1200}$ and $N_{>1200}$. $N_{14-260}$ quickly decreased above the ground-based peak (-56.7±4.4%) to a concentration value of 260.6±13.1 cm$^{-3}$ analogous to that observed in standard NG profiles (252.3±17.5 cm$^{-3}$) below the $AS_h$ (before reaching $H_s$=0). $N_{260-1200}$ and $N_{>1200}$ instead remained quite constant (32.4±0.8 cm$^{-3}$ and 0.17±0.01 cm$^{-3}$, respectively) from ground until $H_s$=0 where decreased by -31.1±2.9% (to 22.3±0.8 cm$^{-3}$) and by -54.2±4.7% (to 0.08±0.01 cm$^{-3}$), respectively.

Interestingly, eBC concentrations behave contrary to the $N_{14-260}$ aerosol fraction. Lowest eBC concentrations were found close to the ground (36±11 ng m$^{-3}$; $b_{abs}$ was 0.22±0.06 Mm$^{-1}$) in correspondence of the $N_{14-260}$ concentration peak. Above this peak, eBC concentrations were higher (121±5 ng m$^{-3}$; $b_{abs}$ was 0.74±0.03 Mm$^{-1}$).

All the aforementioned observations suggest that a particular process could have influenced ground-level concentrations for this size class only. In order to shade light on this process several parameters will be here below considered, namely: meteorological parameters, the aerosol chemical composition and the aerosol number size distribution. Starting with meteorological parameters, and recalling first Figure 2d, we see that the behavior of smallest particles in the proximity of the ground can be observed concomitantly to the presence of ground-based θ inversions, a necessary condition (or a concurrent cause) to promote the presence of ground-based concentration peaks for $N_{14-260}$. The crucial point to unravel this phenomenon is to understand the possible origin of this particles. Thus, ground-based aerosol and meteorological measurements, collected at Gruvebadet laboratory and at the CCT (section 2.1) and temporally coincident with the observation of DNG profile, were considered.

Figure 8a-d shows the ground-level PM$_{10}$ chemical composition determined for the four categories (HO, PG, NG, DNG) of profiles.

The nss-nc-$SO_4^{2-}$ in DNG profiles (1349.9±354.7 ng m$^{-3}$) was 3.0±0.7 times higher than that observed in the other profile classes (389.9±113.2 ng m$^{-3}$, 410.5±104.3 ng m$^{-3}$ and 622.1±210.0 ng m$^{-3}$, for HO, PG and NG). At the same time, the ss-$SO_4^{2-}$ was 0.6±0.2 times lower compared to the other profile classes while the cr-$SO_4^{2-}$ remained quite constant (ratio 1.1±0.4). The same pattern can be observed considering the aforementioned sulfate fractions in the PM$_{10}$ samples (Figure S7, supplemental material).

These observations, coupled with the lowering of eBC fraction in proximity of the ground (Figure 4l) point towards the hypothesis that the ground-based N$_{14-260}$ concentration peak was secondary in origin. The nss-nc-$SO_4^{2-}$ fraction during DNG profiles appeared in acidic form, as it was just poorly neutralized by the ammonium. Particularly, the w/w (weight/weight) nss-nc-$SO_4^{2-}$/$NH_4^+$ ratio was 1.6±0.4 times higher for DNG profiles (10.3±1.5) than that observed in the other profile classes: 6.1±1.9,5.8±2.1 and 7.0±2.2 for HO, PG and NG, respectively. As reported in literature (Udisti et al., 2016; Becagli et al., 2012; Udisti et al., 2012) these values for DNG profiles feature the presence of sulfate in acidic form ($H_2SO_4$). This is in agreement with the finding that "springtime submicron aerosol in the Arctic surface sites is composed predominantly of partially neutralized sulfate and sea-salt, with lesser contributions from nitrate, BC, soil and trace elements" as reported in Quinn et al. (2002).

This information is very important when coupled with meteorological data measured at the CCT (Table 2). Focusing first on the air temperature, it can be observed that, during DNG profiles, the temperature close to the ground (-17.2±0.3 °C) was lower than that observed in the other profile classes: -9.7±0.5 °C, -5.9±0.3 °C and -9.9±0.2 for HO, PG and NG, respectively. In addition, the RH was 18.0±2.0% higher (73.6±0.3%) for DNG profiles, compared to that observed in the other profile classes. Finally, also the wind speed during DNG profiles was half than during the other profile classes and was not affected by north direction, avoiding the influence of Ny-Ålesund (Figure 5d). All the aforementioned conditions, featured during DNG profiles, can be resumed in: higher acidic sulfate fraction, lower eBC fraction, lower temperature, higher relative humidity and lower wind speed during DNG profiles.

As reported in literature (Kirkby et al., 2011; Reddington et al., 2011;Lovejoy et al., 2004) these conditions decrease the height of the barrier for new particle formation just considering the very simplified binary $H_2SO_4$–$H_2O$ system. Under these conditions, the secondary aerosol formation can proceed at ambient acid concentrations in the cooler mid-troposphere and at lower altitude in polar regions. However, it has to be underlined that, as recently reported (Riccobono et al., 2014), organics plays a fundamental role for secondary aerosol formation. They were found in Ny-Ålesund even in spring (Zangrando et al., 2013). These figures, coupled with the aforementioned data indicated the N$_{14-260}$ concentration peaks at ground as locally formed secondary aerosol.

The ground-based aerosol number size distribution (Figure 6) shows a huge Aitken mode for DNG profiles, while it is negligible for the other profile classes. This mode was characterized by a geometric mean diameter $D_g$ of 0.032±0.001 μm and by a geometric standard deviation $\sigma_g$ of 1.790±0.006 that were in agreement with ten-year average values reported in Tunved et al. (2013) at the Zeppelin observatory during the month of April. The presence of a clearly visible Aitken mode in DNG profiles supports the aforementioned hypothesis of the presence of a ground-based plume of locally newly formed aerosol particles.

In order to estimate (meaning the order of magnitude) the contribution of locally formed aerosol, the method based on the N/eBC ratio (section 2.2.1), developed by Rodríguez and Cuevas (2007) was used. The value of $S_1$ (2.4±0.2) was taken as that of background aerosol above the the ground N$_{14-260}$ plume in DNG profiles. This value was very similar to that measured during homogeneous profiles (2.5±0.1) when a pure background aerosol was measured.

These S1 values, obtained over Ny-Ålesund, are close to the lowest values reported in literature (Reche et al., 2011; Dall'Osto et al., 2013), a fact resulting from the $d_{50}$ cutoff size of the miniDiSC (14 nm; section 2.2.1), which is higher than $d_{50}$ cutoff sizes (~2-7 nm) usually present in the widely used condensation particle counters. Using this reference $S_1$ value, $N_1$ (background number concentration) and $N_2$ (locally formed secondary aerosol) were computed as reported in section 2.2.1. Their vertical behavior is reported in Figure 9. The total amount of secondary aerosol close to the ground is clearly visible and accounted on average for 63.7±5.6% (up to 95% at ground) of the total $N_{14-260}$ plume. In fact, within these plumes, the $N_{14-260}$/eBC ratio reached an average value of 22.5±5.4 (maximum value of 54.8 at ground) clearly indicating the presence of a secondary formed aerosol (Reche et al., 2011; Dall'Osto et al., 2013). In addition to this, Figure 10 shows the temporal behavior of SMPS+APS data collected at Gruvebadet during April 2011 together with the percentiles (25°, 50°, 75°, 90°) of the measured number size distribution. It is clearly evident the presence of nanoparticles (below 100 nm) even in Spring in the Arctic.

Interestingly, all the aforementioned results are analogous to data reported in ARCPAC by Brock et al. (2011). Within the surface inversion layer over sea-ice, they found a region of depleted BC and organic mass concentrations (lower than in the background case), while sulfate concentrations were similar or higher. However, under this condition, Brock et al. (2011), did not feature an increase in particle number concentration. Surprisingly, DNG profiles are more similar to vertical aerosol profiles discussed by Kupiszewski et al. (2013) during the summertime ASCOS campaign. In fact, they found a plume of nanoparticles within near-surface layer (not related with the behavior of accumulation mode particles) during new particle formation events. They hypothesize that the origin of ultrafine particles was related to biological processes. This observation becomes important when considering again the number size distribution reported in Figure 10. The 75° and 90° percentile exhibited a summer-like behavior when compared with Zeppelin data reported in Engvall et al. (2008) and in Tunved et al. (2013). These findings point towards the importance of measuring the frequencies of these episodes, present in the surface layer of Ny-Ålesund, together with their vertical development (i.e. vertical mixing), to understand their importance of CCN influencing the Arctic climate.

### 3.3    Summer phenomenology

Vertical profiles measured during summers 2011-2012 were also classified according to their vertical behavior (i.e. shape). They were averaged considering the relative position of each measured data point with respect to the $AS_h$. The obtained averaged vertical profiles were referred to the standardized height $H_s$. The result of the classification and averaging procedure is reported in Figure 11a-f (all profile data are reported in Figure S8, supplemental material). In summer, two main categories were observed:

      1) Type 1, homogeneous profiles (HO), Figure 11a-c

      2) Type 2, profiles characterized by the presence of shipping emissions (hereinafter addressed as SP), Figure 11d-f.

Average concentrations of aerosol ($N_{14-260}$, $N_{260-1200}$, $N_{>1200}$) and eBC below and above $H_s$=0 for each profile class are summarized in Table 3.

### 3.3.1 HO profiles

HO profiles (Type1, Figure 11a-c) were observed in 37% of cases due to the summer higher solar power density at disposal together with the low albedo as discussed in section 3.2. As already reported for springtime results, these are the only averaged profiles referred to an absolute height AGL because they did not show any $AS_h$ to calculate $H_s$. During HO profiles, local wind (Figure 5e) was interestingly blowing from the same direction as in the case of HO springtime profiles: SW direction, from the glaciers behind Ny-Ålesund (Figure 1a) and not along the predominant NW-SE direction of the Kongsfjord. However, summertime HO profiles featured higher wind speed than in spring (range 0-11 m s$^{-1}$, average of 4.1±0.1 m s$^{-1}$ at 33 m; Table 2). At the same time, as shown in Figure 11c, a slightly positive θ profiles characterized, on average, the HO profiles.

The aerosol number concentrations were found to be 435.9±5.8 cm$^{-3}$ for $N_{14-260}$, 2.1±0.1cm$^{-3}$ for $N_{260-1200}$, and $4*10^{-2}±4*10^{-3}$ cm$^{-3}$ for $N_{>1200}$, respectively. Thus, $N_{14-260}$ accounted for 99.5±1.9% of the total aerosol number concentration, which is considerably higher than 81.9±16.8% observed in springtime HO profiles. This is in agreement with the observations, reported in literature, that the sunlit summer period is dominated by small locally formed Aitken particles (Giardi et al., 2016; Tunved et al., 2013; Ström et al., 2009 and 2003; Udisti et al., 2013; Viola et al., 2013). eBC and the related $b_{abs}$ were negligible, as also reported by Eleftheriadis et al. (2009). HO profiles were observed in the absence of ships anchoring in the Ny-Ålesund harbor.

### 3.3.2 Ship impact along vertical profiles

Summer vertical profiles showed a considerable impact of ship emissions. The number of ships and the number of passengers (a useful proxy of the ship dimension) was registered by the Kings-Bay Kull Company and it is reported in Figure S9 for summer 2011 and 2012, respectively. Particularly, 57 days with a total of 103 ship arrivals were registered during JJA of 2011 (62% of days; Figure S9a) while 78 days (85% of days) with 138 ships (Figure S9b) were registered during JJA of 2012.

Figure 12a-e reports the case study of 6$^{th}$ July 2011, when four ships anchored (not simultaneously) in the harbor of Ny-Ålesund from 0700 UTC to 1900 UTC. The largest ship arrived in the morning with approx. 1000 passengers. Figure 12a-d shows four profiles (0740 UTC, 0901 UTC, 0932 UTC and 1340 UTC) together with ground SMPS data collected at Gruvebadet (Figure 12e). The ship impact in the Kongsfjord, distant about 1.200 m as the crow flies, is clearly evident. $N_{14-260}$ concentrations reached values up to 2-3*10$^4$ cm$^{-3}$ and the eBC concentrations reached the maximum value of 2000 ng m$^{-3}$ (at 0932 UTC).

To highlight the impact of ship emission along the two years, average vertical profiles were calculated. The result is shown in Figure 11d-f. During SP, wind blew mainly from the N-NE direction (where the harbor is located; Figure 5f). The average wind speed was 2.6±0.1 m s$^{-1}$. Once a ship plume arrived, SP profiles were characterized by high pollution levels below $H_s$=0. Figure 11f also shows a positive gradient of θ (+1.4±0.2 K increase from $H_s$=0). This gradient constrains the ship plume above ground to an altitude variable with time as shown in Figure 12a-d (from 103 m to 592 m). Particularly, $N_{14-260}$ showed a concentration peak close to the ground of $9.0*10^3±2.5*10^2$ cm$^{-3}$, $N_{260-1200}$ of 7.0±0.7 cm$^{-3}$ and $N_{>1200}$ of $6*10^{-2}±4*10^{-3}$ cm$^{-3}$. eBC concentrations behave similarly reaching concentrations of 319±14 ng m$^{-3}$. These concentration values were higher by a factor of 13.9±0.7 ($N_{14-260}$), 5.1±0.5 ($N_{260-1200}$), 4.8±0.4 ($N_{>1200}$) and 13.4±1.0 (eBC) than those observed above $H_s$=0 where an intense decrease of both aerosol and eBC is observed. Crossing the $AS_h$, aerosol concentrations decreased to 648.1±27.3 cm$^{-3}$ for $N_{14-260}$, to 1.4±0.1 cm$^{-3}$ for $N_{260-1200}$ and to $1*10^{-2}±1*10^{-3}$ cm$^{-3}$ for $N_{>1200}$. These values were close to the

background values observed during summer HO profiles. eBC behave similarly decreasing to 24±1 ng m$^{-3}$. As the ship plume of eBC is located close to the ground, it may exert  positive forcing (Flanner, 2013; Brock et al., 2011; Seinfeld and Pandis, 2006; Hansen and Nazarenko, 2004). In fact, the $b_{abs}$ reached values of 1.95±0.09 Mm$^{-1}$ below $H_s$=0.

It is important to note that SP profiles were observed in summer. In summer the long-range transport of aerosol from mid-latitudes is minor (Browse et al., 2012; Quinn et al., 2008; Stohl et al., 2006) and the locally formed aerosol becomes dominant (Giardi et al., 2015; Tunved et al., 2013; Ström et al., 2009 and 2003). Within this context, the SP profiles show that the rising shipping emissions in the Arctic (Corbett et al., 2010; Granier et al., 2006) could affect the concentrations and the vertical distribution of aerosol, resulting in a positive forcing, induced

by a positive feedback through the local anthropogenic impact on climate.

## 4    Conclusions

Vertical profiles of in situ aerosol number size distribution and black carbon measurements were conducted by tethered balloon in the atmosphere over Ny-Ålesund. The balloon payload was equipped with an Optical Particle

Counter (31 size classes, 0.25 to 32 μm), an electrical particle detector (d$_{50}$=14 nm), two micro-Aethalometers and meteorological sensors. Moreover, chemical analysis of filter samples, aerosol size distribution and a full set of meteorological parameters at ground were available. A systematic study of vertical profiles of aerosol number size distribution (14 nm – 32 μm) and equivalent black carbon concentrations was conducted. 200 vertical profiles were measured during spring and summer along 2 years (2011-2012). Vertical aerosol profiles were classified for

each season according to their shape allowing to obtain a description of the seasonal phenomenology of vertical aerosol properties in the Arctic.

Focusing on spring, four main types of profiles were found.

The first one was the homogeneous profiles class (HO), characterized by constant aerosol and eBC concentration with altitude, and representative of Arctic background conditions.

The second class was that of positive gradient profiles (PG) characterized by an increase of aerosol concentration with altitude. The importance of this class is related to the fact that aerosols can act as CCN influencing the cloud cover and thus the longwave fluxes.

The third class was characterized by negative gradient profiles (NG) with a decrease of aerosol concentration with altitude and thus high pollution level close to the ground. This finding is very important because a eBC layer

located immediately above snow and ice may induce a positive forcing.

The fourth class of profiles was characterized by negative gradients located at different altitude in function of size (DNG). These profiles were observed during ground-based events of locally formed secondary aerosol. It is important as locally formed aerosol can act as CCN. As low clouds play a particular role in the sensitive Arctic climate system, the aerosol-cloud interactions will be one focus of future research activities within the Ny-Ålesund

research community, manifested in the Ny-Ålesund Atmospheric Flagship program.

The four categories described above are important when considering the large amount of ground-based data available for comparisons with modelling results. Particularly, for HO profiles, ground measurements were fully representative of the vertical column (up until ~1 km, vertical limit of experimental activity). During NG and PG profiles, the ground-based measurements were representative of the air column up to the planetary boundary layer.

Finally, DNG profiles showed that ground-based measurements differ from those conducted aloft. However, the

last case was influenced by secondary aerosol formation that can be easily detected by an SMPS (or similar experimental devices). Thus, ground-based measurements (coupled with a proper PBL determination) are fundamental and very useful for model comparison. In addition to these, vertical profiles shed light on the phenomenology and dynamics of the vertical distribution of aerosols in the Arctic.

During summer, two main types of profiles were observed. The first class was characterized by homogeneous background condition profiles while the second class reflected the impact of shipping emissions. The ship impact resulted in a plume of aerosol and eBC pollution constrained close to the ground. In summer, atmospheric transport from mid-latitudes is minor. Increasing shipping emissions in the Arctic could significantly increase anthropogenic aerosol and eBC concentrations in the summer Arctic, enhancing the climate change that this region is already experiencing.

**Acknowledgements**

The scientific activity in Ny Alesund was carried out in the framework of the CNR Polar Program. We thank the PRIN2009 "ARCTICA" project for financial support. We thank the Alfred Wegener Institute and the CICCI project for logistical support. We thank both Brent Holben and Piotr Sobolewski for their effort in establishing and maintaining the AERONET Hornsund site. Finally, we thank LSI-Lastem and Federico Pasquini for the cooperation concerning meteorological instruments.

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

| Date | UTC Time | N° profiles | max altitude (m) | Cloud base (m) |
|---|---|---|---|---|
| Spring 2011 | | | | |
| 30-03-2011 | 1240-1518 | 6 | 741 | No clouds |
| 01-04-2011 | 0630-1725 | 10 | 788 | No clouds |
| 04-04-2011 | 1817-2018 | 6 | 748 | 1152 |
| 06-04-2011 | 1642-1922 | 8 | 716 | No clouds |
| 07-04-2011 | 1251-1923 | 10 | 712 | No clouds |
| 08-04-2011 | 0825-1944 | 14 | 740 | No clouds/1534[1] |
| 10-04-2011 | 1245-1438 | 6 | 300 | No clouds/984[2] |
| 14-04-2011 | 1502-1558 | 5 | 738 | No clouds |
| 22-04-2011 | 1917-2009 | 4 | 846 | No clouds |
| 23-04-2011 | 1210-1334 | 6 | 1008 | 2414 |
| 26-04-2011 | 1607-2200 | 8 | 1152 | No clouds |
| 30-04-2011 | 0946-1048 | 6 | 855 | 4018 |
| Summer 2011 | | | | |
| 06-07-2011 | 0740-1755 | 10 | 1143 | No clouds/2813[3] |
| 08-07-2011 | 1643-2053 | 2 | 1208 | 1787 |
| 12-07-2011 | 0819-1001 | 6 | 724 | 506 |
| Summer 2012 | | | | |
| 21-06-2012 | 1512-1611 | 2 | 980 | No clouds |
| 23-06-2012 | 0555-1107 | 12 | 1024 | 622 |
| 24-06-2012 | 1102-1516 | 8 | 1076 | No clouds |
| 26-06-2012 | 0743-1300 | 10 | 948 | No clouds |
| 29-06-2012 | 0758-1319 | 10 | 1144 | No clouds |
| 30-06-2012 | 0920-2030 | 8 | 1100 | No clouds/821[4] |
| 01-07-2012 | 0835-2140 | 10 | 1212 | No clouds |
| 04-07-2012 | 1330-1805 | 8 | 1192 | 654 |
| 10-07-2012 | 0844-2107 | 8 | 1268 | No clouds |
| 11-07-2012 | 0837-2320 | 14 | 1196 | No clouds/722[5] |

Table 1. Dates, UTC time, number of profiles, maximum altitude and sky conditions reached during the 2011-2012 Spring-Summer campaign in Ny-Ålesund; [1]from 1600 UTC (last 6 profiles), [2]variable for half of the time, [3]variable for half of the time, [4]clouds until 1130 UTC, [5]from 2219 UTC (last 2 profiles).

| Season | Profile Type | | T (°C) | | | | RH (%) | | | | WS (m/s) | | | | P (hPa) |
|---|---|---|---|---|---|---|---|---|---|---|---|---|---|---|---|
| | | | 33 m | 10 m | 5 m | 2 m | 33 m | 10 m | 5 m | 2 m | 33 m | 10 m | 5 m | 2 m | Ground |
| Spring | HO | mean | -8.5 | -8.9 | -9.7 | -9.7 | 61.0 | 61.2 | 61.8 | 62.7 | 0.6 | 0.7 | 0.6 | 0.6 | 997.4 |
| | | $\sigma_m$ | 0.5 | 0.5 | 0.5 | 0.5 | 1.0 | 1.0 | 0.9 | 0.9 | 0.1 | 0.1 | 0.1 | 0.1 | 0.4 |
| | PG | mean | -4.7 | -5.2 | -5.8 | -5.9 | 64.8 | 65.0 | 65.9 | 67.3 | 2.3 | 2.1 | 1.8 | 1.7 | 996.2 |
| | | $\sigma_m$ | 0.2 | 0.3 | 0.3 | 0.3 | 0.4 | 0.4 | 0.4 | 0.4 | 0.1 | 0.1 | 0.1 | 0.1 | 0.5 |
| | NG | mean | -8.9 | -9.3 | -9.9 | -9.9 | 57.2 | 58.5 | 59.1 | 60.1 | 1.3 | 1.1 | 1.0 | 0.9 | 992.8 |
| | | $\sigma_m$ | 0.2 | 0.2 | 0.2 | 0.2 | 0.4 | 0.3 | 0.3 | 0.3 | 0.1 | 0.1 | 0.1 | 0.1 | 0.2 |
| | DNG | mean | -15.8 | -16.5 | -17.2 | -17.2 | 72.6 | 73.2 | 72.9 | 73.6 | 0.7 | 0.9 | 1.0 | 0.9 | 994.8 |
| | | $\sigma_m$ | 0.3 | 0.3 | 0.3 | 0.3 | 0.5 | 0.3 | 0.3 | 0.3 | 0.1 | 0.1 | 0.1 | 0.1 | 0.4 |
| Summer | HO | mean | 5.7 | 5.8 | 5.5 | 6.0 | 76.5 | 75.3 | 74.7 | 74.9 | 4.1 | 3.8 | 3.6 | 3.4 | 1004.1 |
| | | $\sigma_m$ | 0.1 | 0.1 | 0.1 | 0.1 | 0.2 | 0.2 | 0.2 | 0.2 | 0.1 | 0.1 | 0.1 | 0.1 | 0.2 |
| | SP | mean | 5.8 | 5.9 | 5.6 | 6.1 | 78.0 | 76.4 | 75.6 | 75.8 | 2.6 | 2.6 | 2.5 | 2.4 | 1001.9 |
| | | $\sigma_m$ | 0.1 | 0.1 | 0.1 | 0.1 | 0.2 | 0.2 | 0.2 | 0.2 | 0.1 | 0.1 | 0.1 | 0.1 | 0.1 |

Table 2. Meteorological parameters (temperature, relative humidity, wind speed, pressure) measured at the CCT at different levels (33, 20, 5 and 2 m) and averaged (timely coincident) for each profile class.

| Season | Profile Type | | $N_{14-260}$ (cm$^{-3}$) | $N_{260-1200}$ (cm$^{-3}$) | $N_{>1200}$ (cm$^{-3}$) | eBC (ng m$^{-3}$) | $b_{abs}$ (Mm$^{-1}$) |
|---|---|---|---|---|---|---|---|
| Spring | HO (Column) | mean | 80.2 | 17.5 | 0.20 | 35 | 0.22 |
| | | $\sigma_m$ | 16.4 | 2 | 0.10 | 21 | 0.13 |
| | PG (Hs<0) | mean | 205.4 | 19.9 | 0.26 | 24 | 0.14 |
| | | $\sigma_m$ | 12.5 | 0.2 | 0.02 | 3 | 0.02 |
| | PG (Hs>0) | mean | 557.6 | 22.3 | 0.16 | 26 | 0.16 |
| | | $\sigma_m$ | 45.9 | 1.4 | 0.02 | 4 | 0.02 |
| | NG (Hs<0) | mean | 252.3 | 23.1 | 0.53 | 71 | 0.43 |
| | | $\sigma_m$ | 17.5 | 0.4 | 0.05 | 4 | 0.02 |
| | NG (Hs>0) | mean | 118.9 | 9.7 | 0.18 | 39 | 0.24 |
| | | $\sigma_m$ | 9.3 | 0.3 | 0.02 | 2 | 0.01 |
| | DNG (Hs<0, Ground Aitken Plume) | mean | 601.3 | 32.4 | 0.17 | 36 | 0.22 |
| | | $\sigma_m$ | 19.9 | 0.8 | 0.01 | 11 | 0.06 |
| | DNG (Hs<0, Above Ground Plume) | mean | 260.6 | 32.4 | 0.17 | 121 | 0.74 |
| | | $\sigma_m$ | 13.1 | 0.8 | 0.01 | 5 | 0.03 |
| | DNG (Hs>0) | mean | 187.8 | 22.3 | 0.08 | 102 | 0.62 |
| | | $\sigma_m$ | 17.8 | 0.5 | 0.01 | 11 | 0.07 |
| Summer | HO (Column) | mean | 435.9 | 2.1 | 0.04 | -- | -- |
| | | $\sigma_m$ | 5.8 | 0.1 | 0.004 | -- | -- |
| | SP (Hs<0) | mean | 9000 | 7.0 | 0.06 | 319 | 1.95 |
| | | $\sigma_m$ | 250 | 0.7 | 0.004 | 14 | 0.09 |
| | SP (Hs>0) | mean | 648.1 | 1.4 | 0.01 | 24 | 0.15 |
| | | $\sigma_m$ | 27.3 | 0.1 | 0.001 | 1 | 0.01 |

Table 3. Average concentrations of $N_{14-260}$, $N_{260-1200}$, $N_{>1200}$, eBC and $b_{abs}$ along height over Ny-Ålesund for the springtime and summertime typologies of vertical profiles.

.

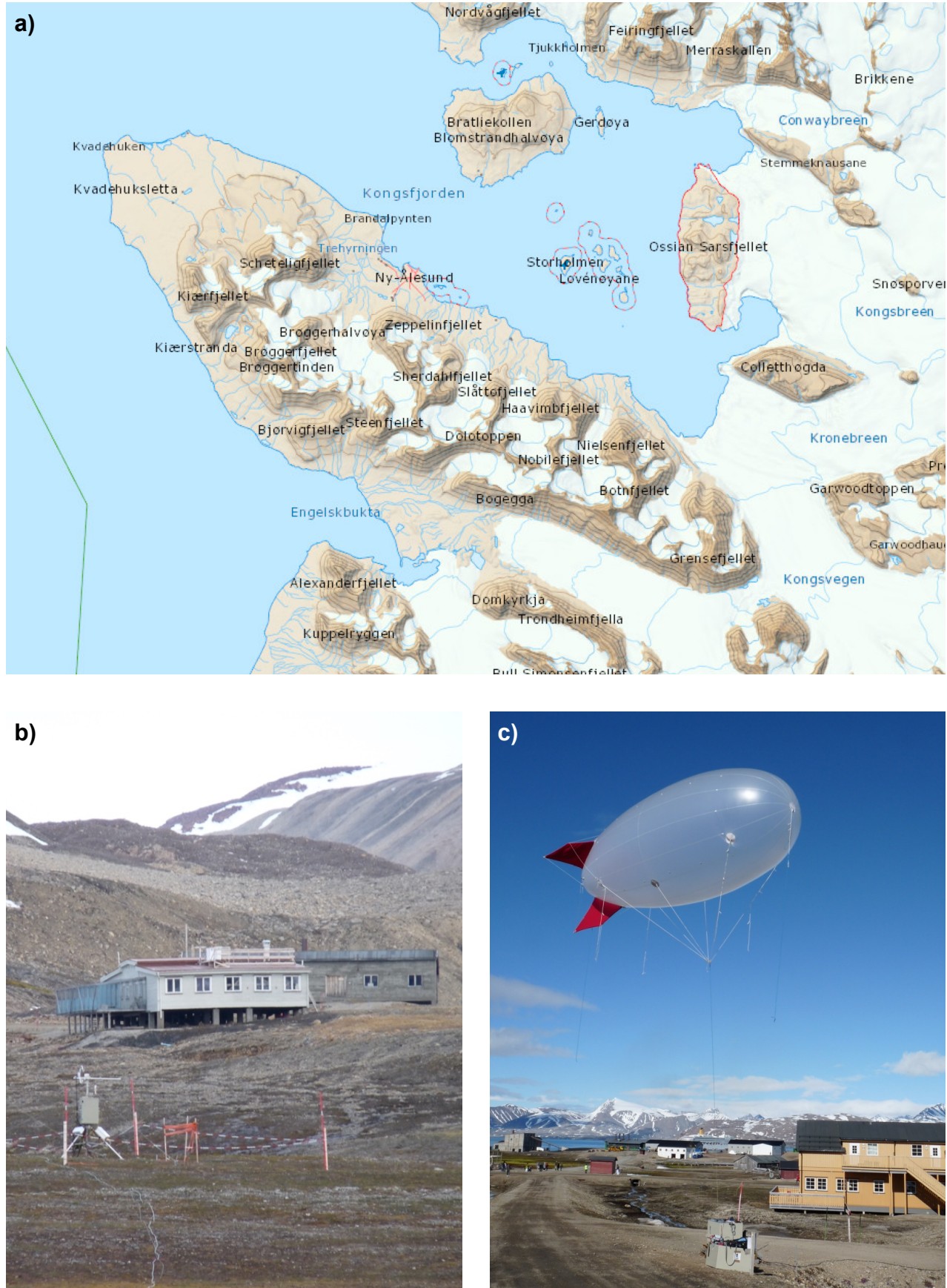

**Figure 1. a) Ny-Ålesund, the Kongsfjord and the surrounding orography; b) Gruvebadet sampling site; c) the tethered balloon in Ny-Ålesund.**

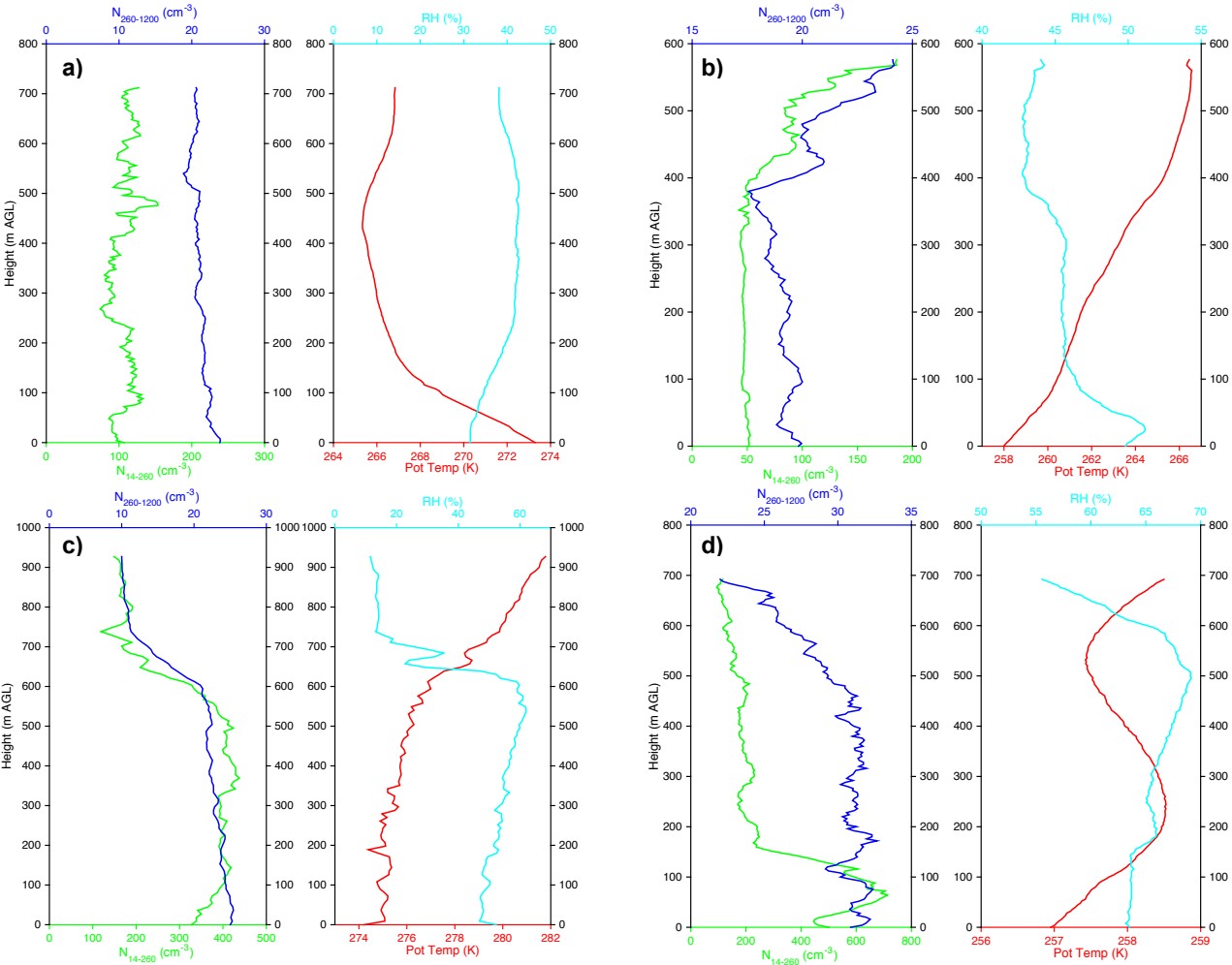

**Figure 2. Vertical profiles of $N_{14-260}$ (green line), $N_{260-1200}$ (blue line), potential temperature (red line) and relative humidity (light blue line) measured over Ny-Ålesund on: a) 7th April 2011 (1251-1310 UTC); b) 1st April 2011 (0630-0657 UTC); c) 23th April 2011 (1321-1334 UTC); d) 6th April 2011 (1803-1822 UTC).**

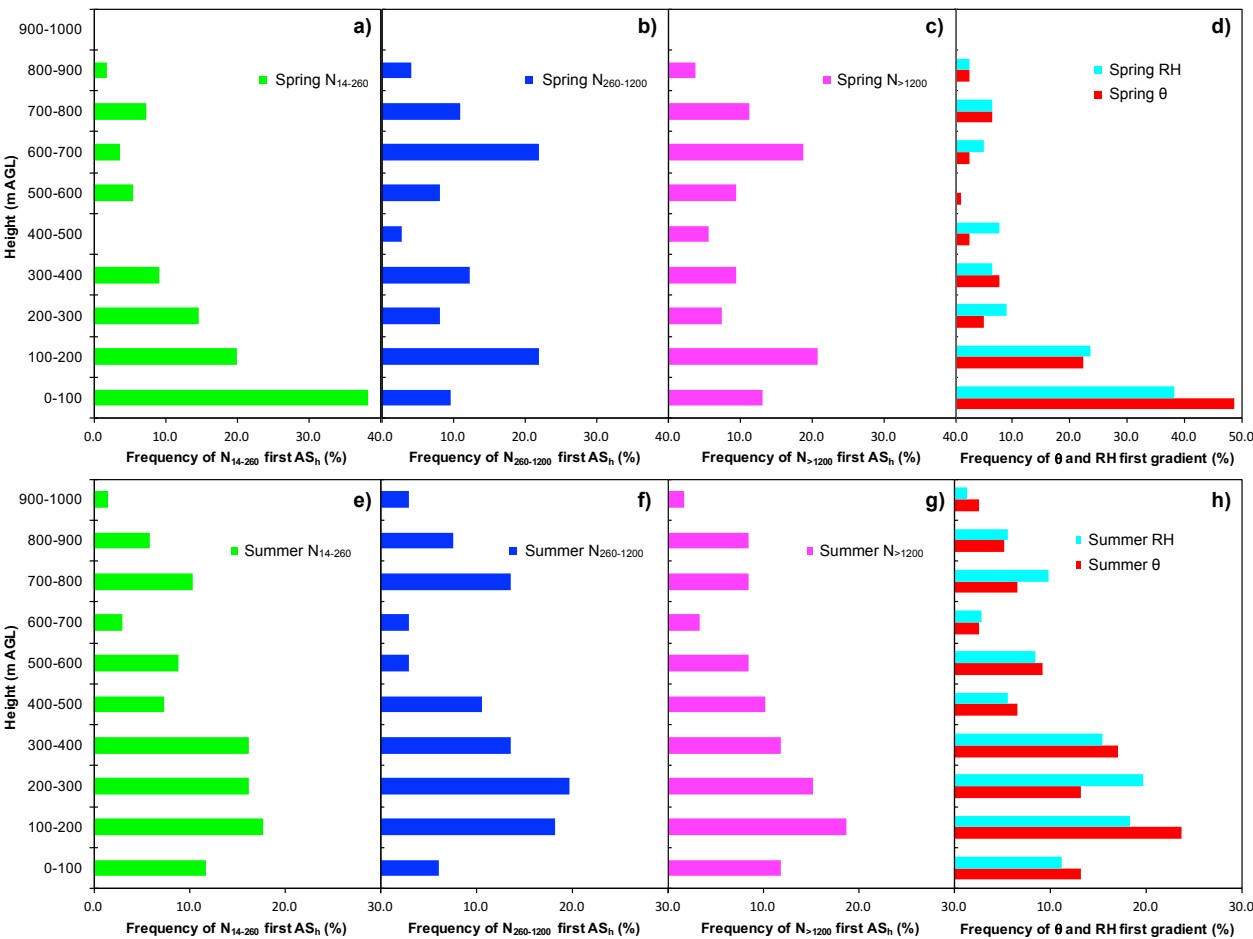

**Figure 3. Vertical frequency distribution of the AS$_h$ for N$_{14-260}$, N$_{260-1200}$, N$_{>1200}$, $\theta$ and RH during spring in panels from a) to d) and in summer in panels from e) to h).**

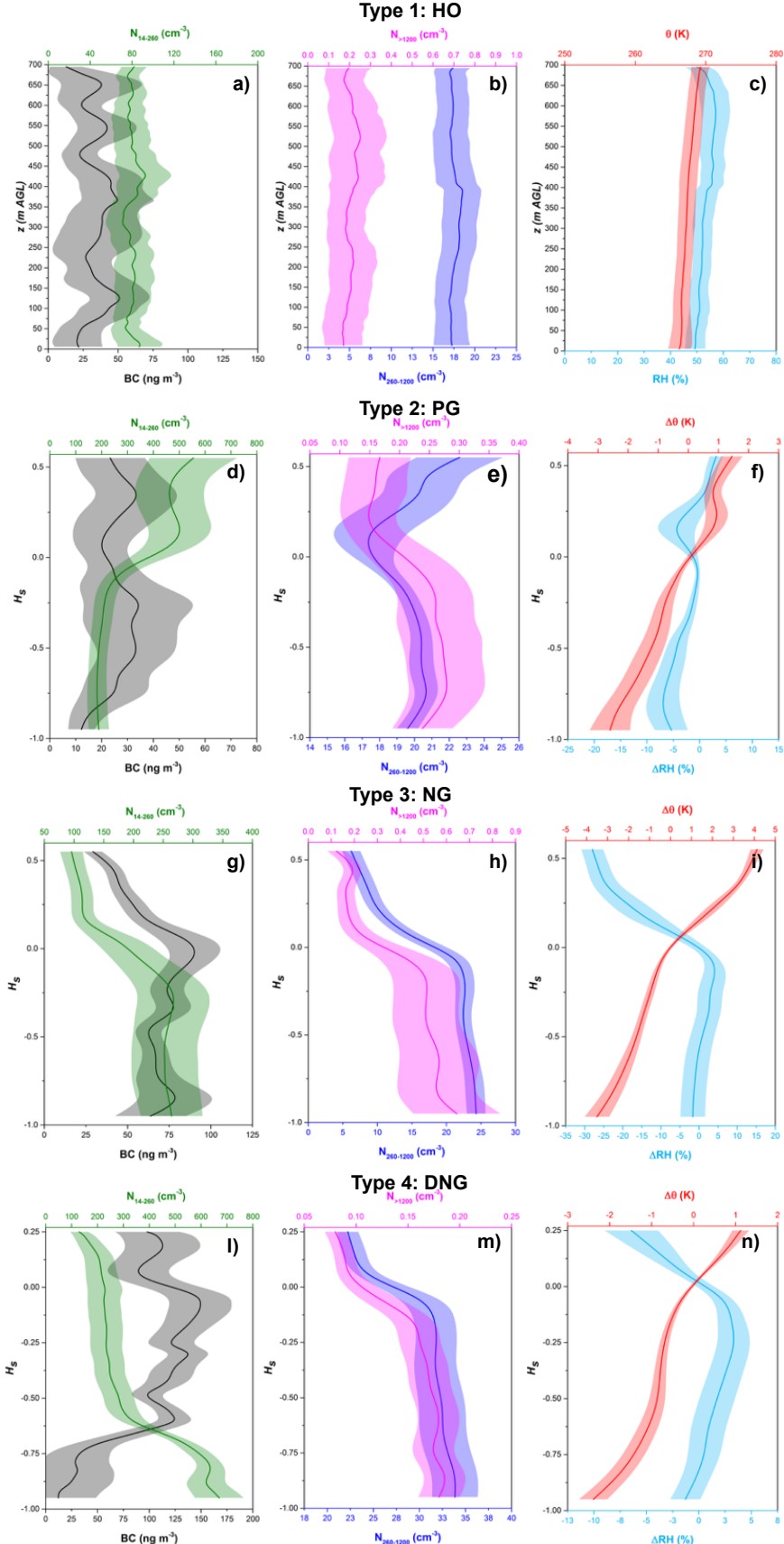

**Figure 4. Springtime statistical mean profiles of N$_{14-260}$ (green line), BC (black line), N$_{260-1200}$ (blue line), N$_{>1200}$ (magenta line), θ (red line) and RH (light blue line) along height over Ny-Ålesund for the four typologies of vertical profiles: a-c) homogeneous profiles (HO); d-f) positive gradient profiles (PG); g-i) negative gradient profiles (NG); l-n) decoupled negative gradient profiles (DNG).**

**Spring**

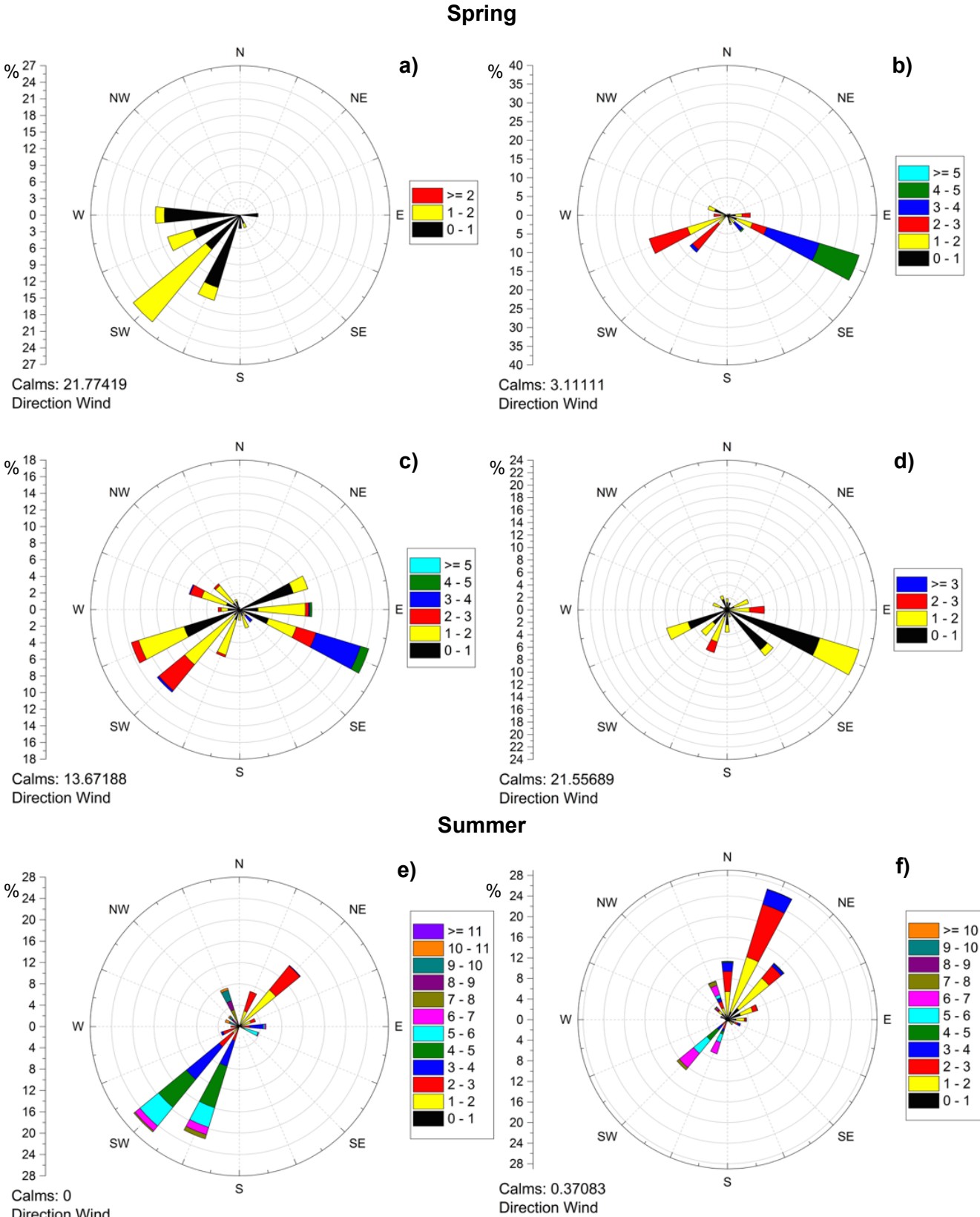

**Figure 5. Wind rose obtained from the measured wind speed and direction at the CCT (33 m). Springtime wind rose timely coincident with: a) homogeneous profiles (HO); b) positive gradient profiles (PG); c) negative gradient profiles (NG); d) decoupled negative gradient profiles (DNG). Summertime wind rose timely coincident with: e) homogeneous profiles (HO); f) profiles impacted by shipping emissions (SP).**

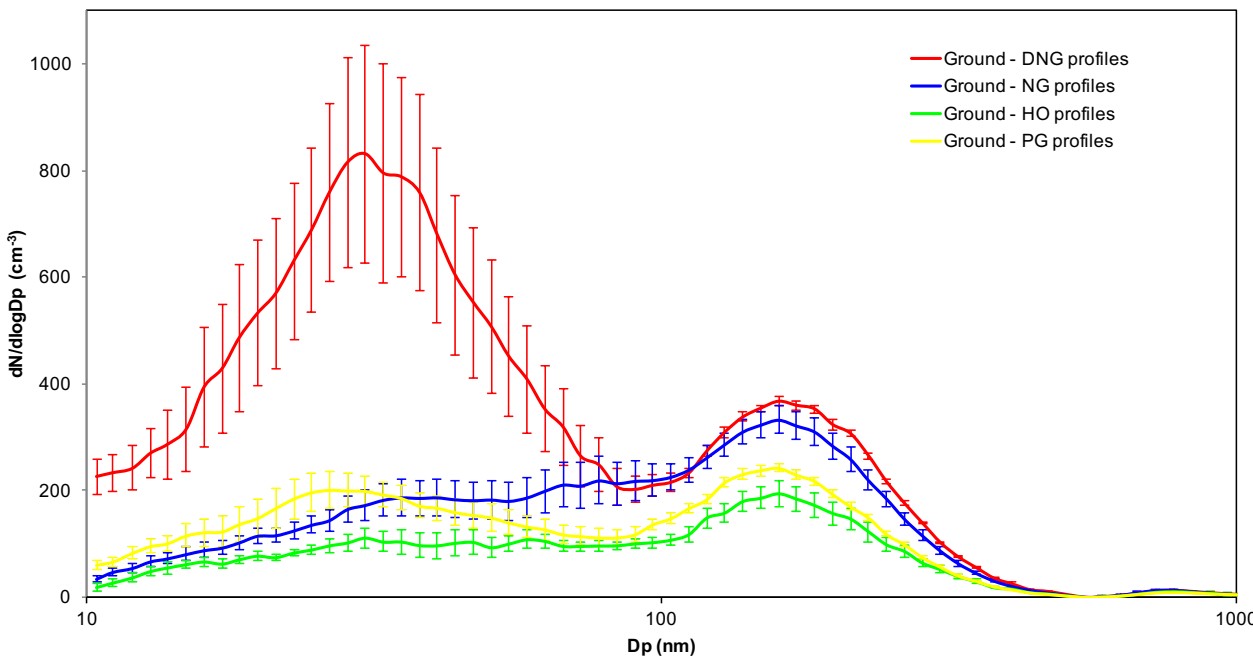

**Figure 6.** Springtime aerosol number size distribution measured at ground and timely coincident with: homogeneous profiles (HO), positive gradient profiles (PG), negative gradient profiles (NG) and decoupled negative gradient profiles (DNG).

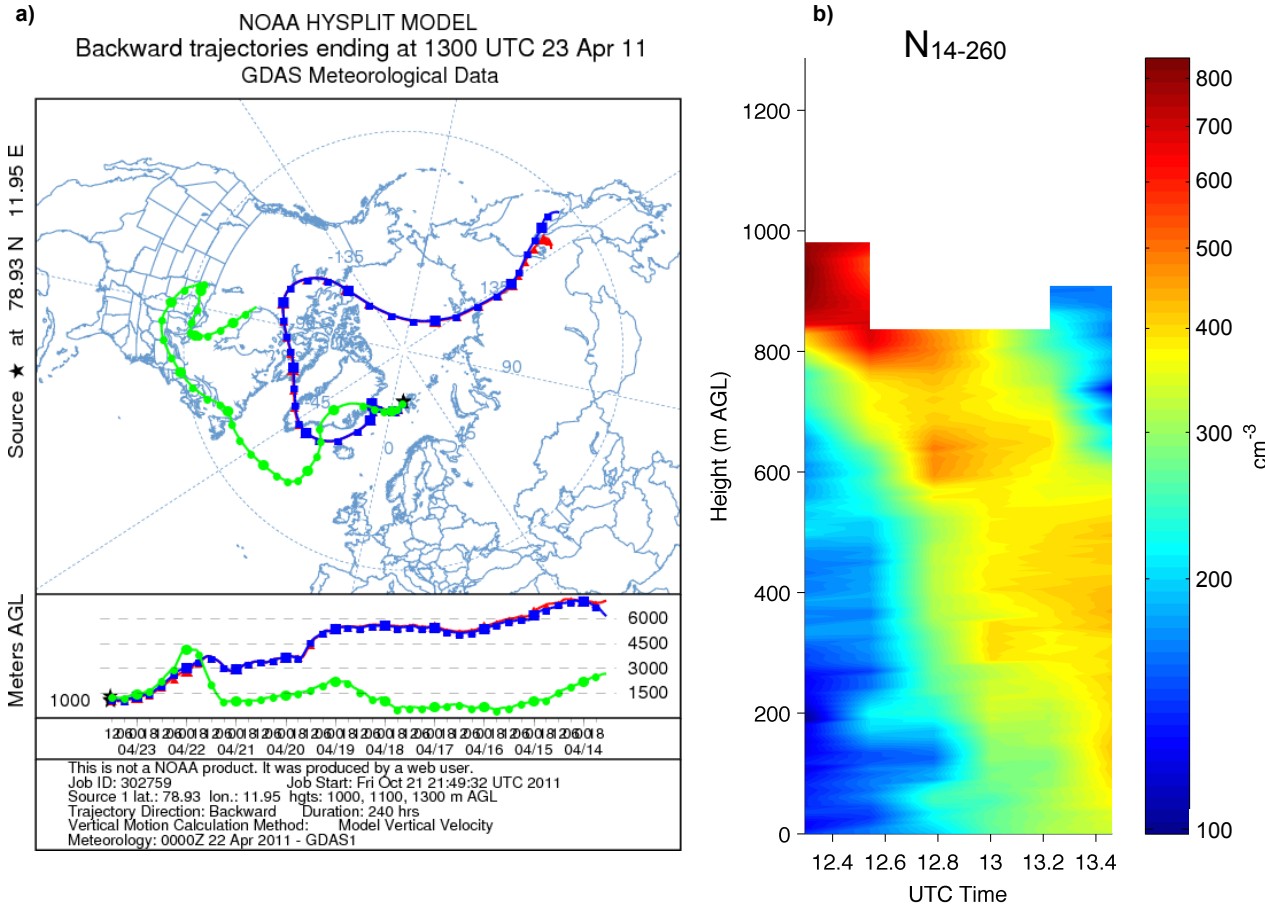

**Figure 7. Case study of 23th April 2011 (1200-1330 UTC): a) air mass back-trajectories; b) time evolution of $N_{14-260}$ obtained through the interpolation of 6 vertical profiles, each of which lasted ~15 min and was distanced ~1 min from the following profile.**

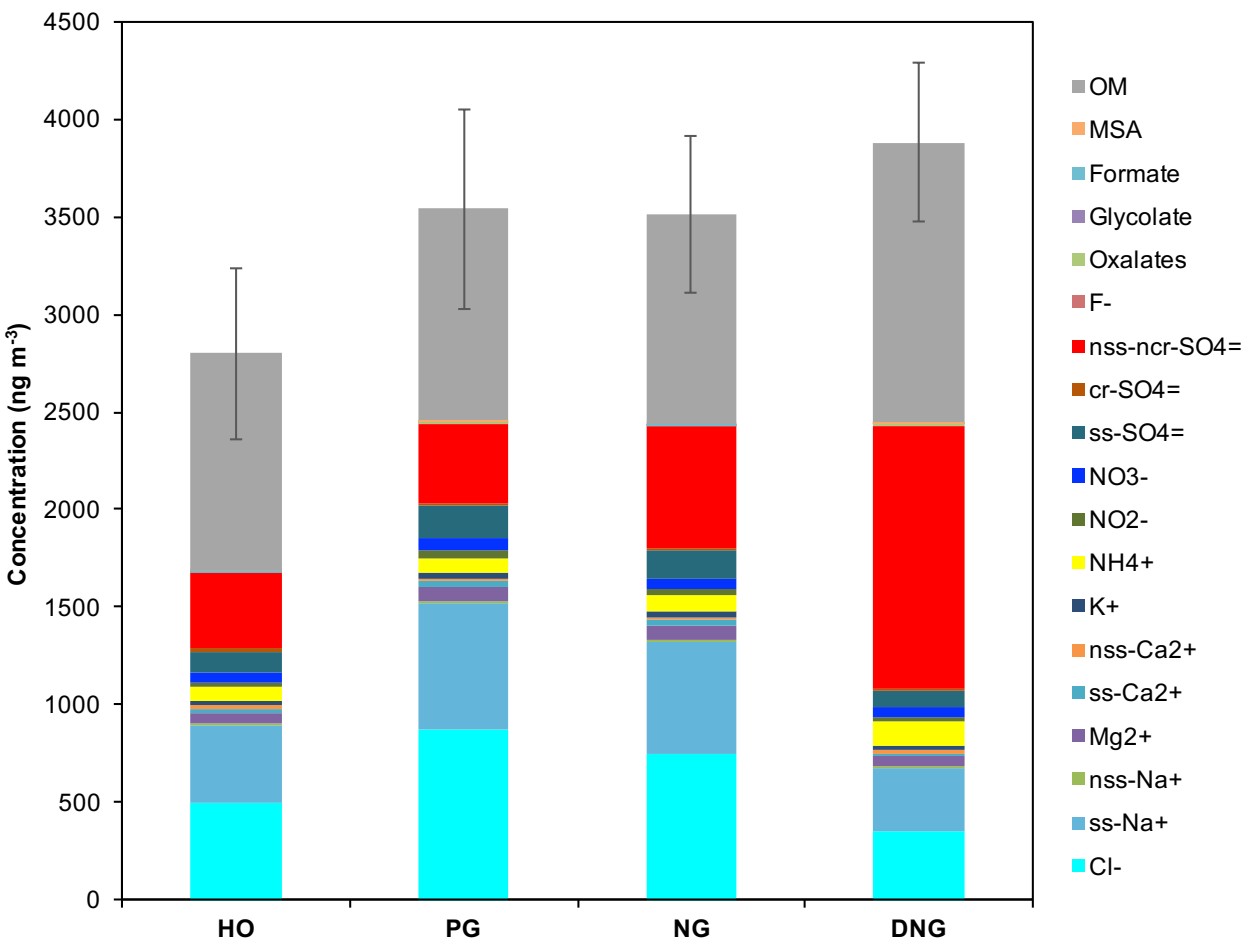

**Figure 8. Springtime aerosol chemical composition determined at ground during: a) homogeneous profiles (HO); b) positive gradient profiles (PG); c) negative gradient profiles (NG); d) decoupled negative gradient profiles (DNG). Data shown are the ambient concentrations of each individual aerosol species.**

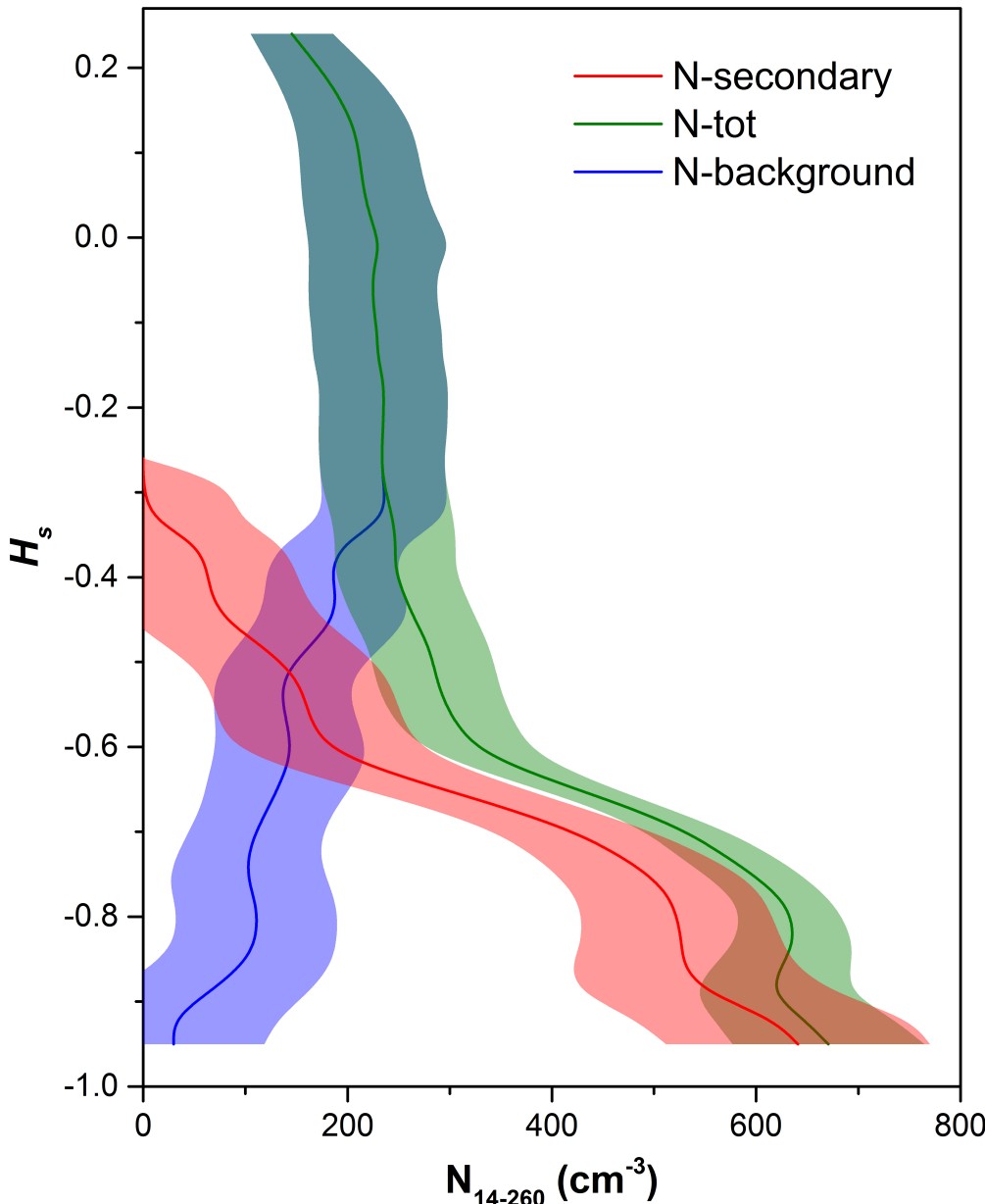

**Figure 9. Springtime statistical mean profiles of N$_{14-260}$ (green line) apportioned along height for the contribution of background N$_{14-260}$ aerosol (blue line) and of secondary locally formed N$_{14-260}$ aerosol (red line) for the decoupled negative gradient profiles (DNG) category.**

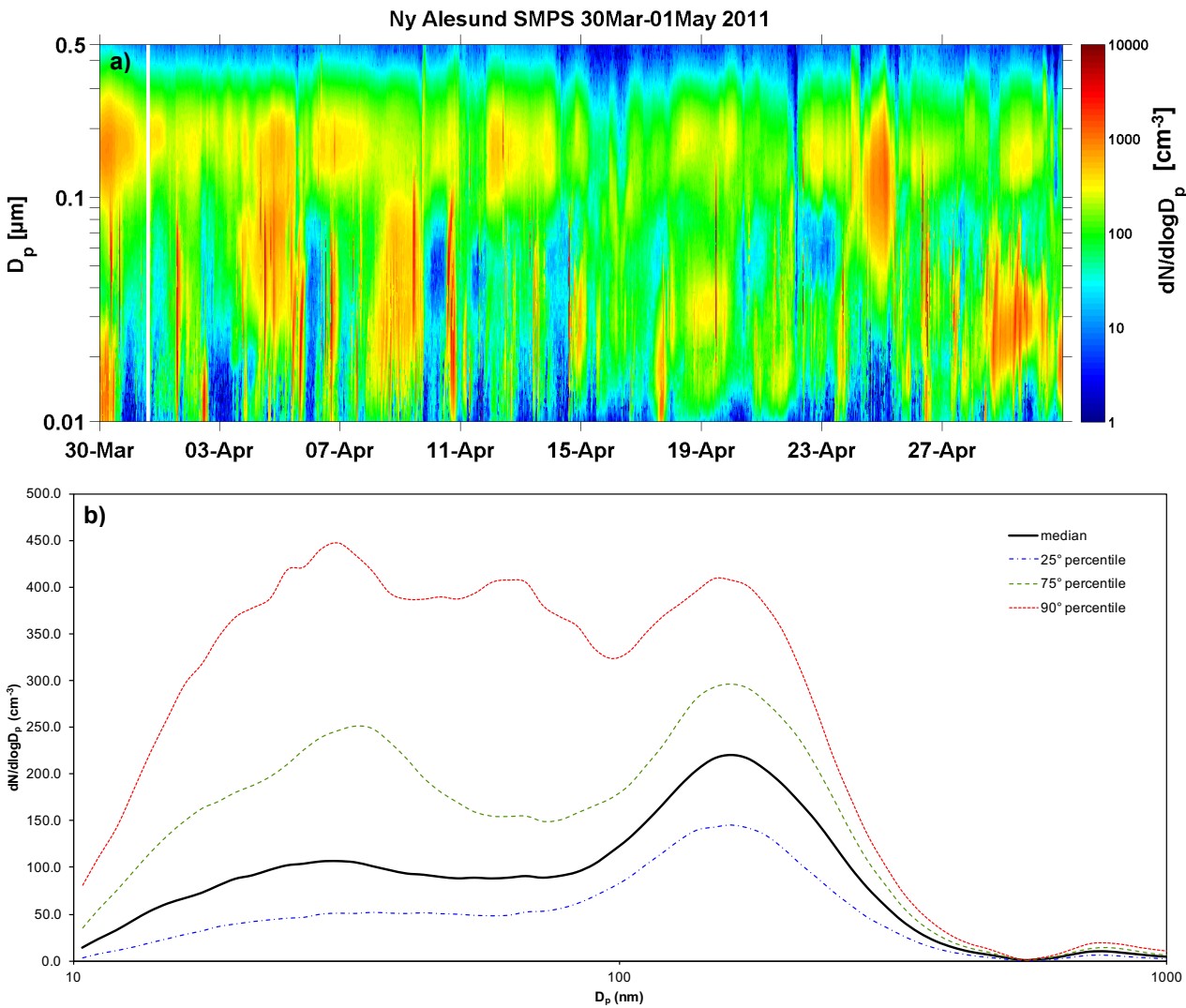

**Figure 10. a) aerosol number size distribution measured at ground during April 2011; b) 25°, 50°, 75° and 90° percentiles of the measured number size distribution.**

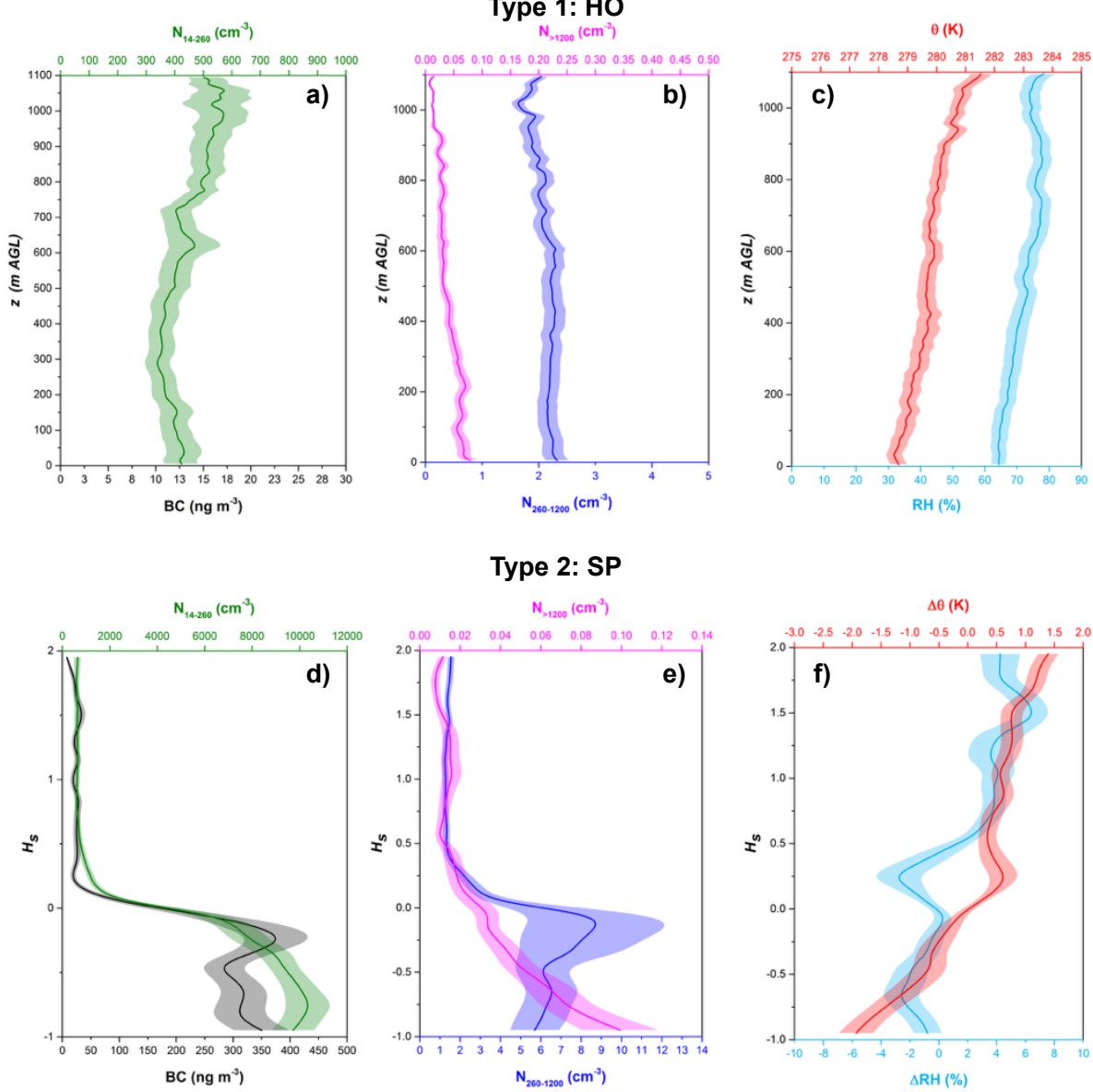

**Figure 11. Summertime statistical mean profiles of N_{14-260} (green line), BC (black line), N_{260-1200} (blue line), N_{>1200} (magenta line), θ (red line) and RH (light blue line) along height over Ny-Ålesund for the two typologies of vertical profiles: a-c) homogeneous profiles (HO); d-f) profiles impacted by shipping emissions (SP).**

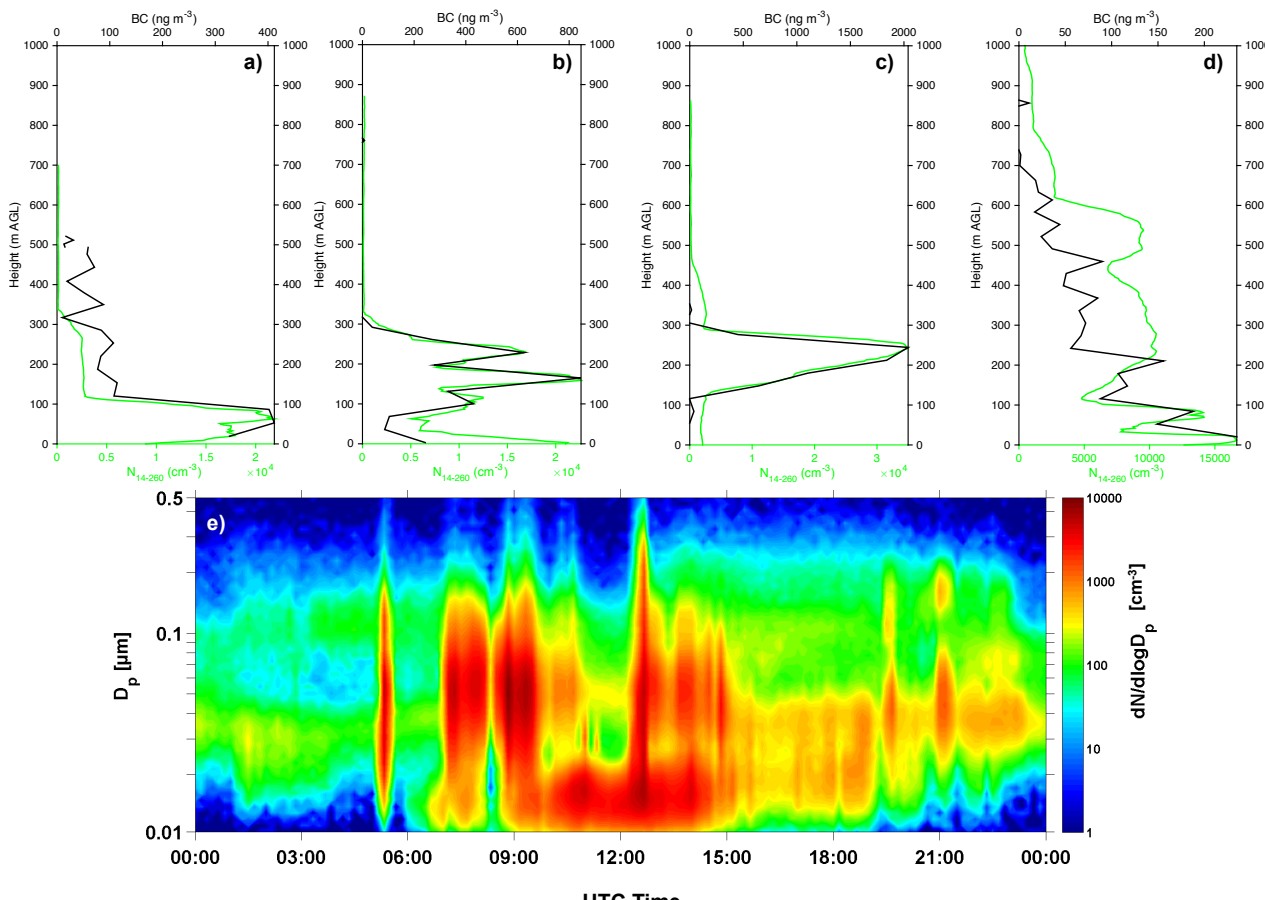

**Figure 12. Case study of 6th July 2011 when four ships anchored (not simultaneously) in the harbor of Ny-Ålesund. Vertical profiles of $N_{14-260}$ (green line) and BC (black line) (0740 UTC, 0901 UTC, 0932 UTC and 1340 UTC) are reported in panels from a) to d) together with ground SMPS data collected at Gruvebadet (e). Note the change in BC scale to progressively increasing BC values during the peak of the ship activity at mid-day.**