# Peer review of "Vertical profiles of aerosol and black carbon in the Arctic: a seasonal phenomenology along two years (2011-2012) of field campaigns"

_Atmospheric Chemistry and Physics, 2016_

## Referee Comment (RC1) · Anonymous Referee #1 · 12 May 2016

This study reports vertical profiles of aerosol number size distribution and black carbon (BC) concentrations from balloon measurements during a field campaign in Ny-Ålesund, Svalbard spring and summer 2011-2012. The authors divide the number size distribution into 3 modes and classify the vertical profiles in four shapes during spring. The authors also discuss secondary aerosol formation and emissions from shipping during summer.

This study is important because 1) the vertical distribution of aerosols affects its radiative forcing and 2) measurements of the vertical distribution of aerosols in the Arctic are particularly sparse. Since such measurements are highly needed and valuable, I think this study is relevant and within the scope of ACP.

The Method section is clearly outlined and the different instruments used are sufficiently and well explained. The figures are nice and easy to follow. However, I think the overall presentation of the results should be improved before it can be published. The manuscript needs more work in terms of language and structure. If this can be achieved, I recommend the manuscript for publication.

General comments:

1. The quality of the English language in this manuscript is variable (some parts are good, but others less good), and I think it would benefit by a thoroughly review of the language (and a spell check!). I have added a few examples under minor comments.

2. The Results section would be easier to read if it was shortened a bit. Description of the methodology should always be under Methods, not Results. I have a few specific suggestions below.

3. I miss a broader implication of this study. Why did you separate the profiles into the four shapes? Comprehensive measurement studies like this can provide physical understanding for evaluation/improvement of the modeling of aerosol processes. Do you have any suggestions? I understand that you cannot add any modeling, but I would like to know more what we can learn from this study.

4. Measurements of vertical profiles in the Arctic are sparse, but there are a few, e.g.:

- two ARCTAS campaigns in the North American Arctic (Jacob et al. 2010) in April and June/July 2008

- the ARCPAC campaign conducted together with ARCTAS in spring 2008 (Brock et al. 2011)

-the PAMARCMIP campaign in April 2009 (Stone et al. 2010) -the HIPPO campaign (Schwarz et al. 2010, 2013; Wofsy 2011) January and October 2009 + winter and autumn 2009

- the ARCTAS/ARCPAC campaign in spring 2008,

- the ARCTAS campaign in summer 2008

- the PAMARCMIP campaign in spring 2009.

On a general basis; How are those compared to your study? I suggest you also include more of these studies in the introduction.

Jacob, D.J., J.H. Crawford, H. Maring, A.D. Clarke, J.E. Dibb, L.K. Emmons,R.A. Ferrare, C.A. Hostetler, P.B. Russell, H.B. Singh, A.M. Thompson,G.E. Shaw, E. McCauley, J.R. Pederson and J.A. Fisher, 2010. The Arctic Research of the Composition of the Troposphere from Aircraft and Satellites (ARCTAS) mission: design, execution, and first results. Atmospheric Chemistry and Physics, 10:5191-5212.

Brock, C.A., J. Cozic, R. Bahreini, K.D. Froyd, A.M. Middlebrook, A.McComiskey, J. Brioude, O.R. Cooper, A. Stohl, K.C. Aikin, J.A. De Gouw,D.W. Fahey, R.A. Ferrare, R.-S. Gao, W. Gore, J. Holloway, G. Hubler, A.Jefferson, D.A. Lack, S. Lance, R.H. Moore, D.M. Murphy, A. Nenes,P.C. Novelli, J.B. Nowak, J.A. Ogren, J. Peischl, R.B. Pierce, P. Pilewskie,P.K. Quinn, T.B. Ryerson, K.S. Schmidt, J.P. Schwarz, H. Sodemann, J.R.Spackman, H. Stark, D.S. Thomson, T. Thornberry, P. Veres, L.A. Watts, C.Warneke and A.G. Wollny, 2011. Characteristics, sources, and transport of aerosols measured in spring 2008 during the aerosol, radiation, and cloud processes affecting Arctic climate (ARCPAC) project. Atmospheric Chemistry and Physics, 11:2423-2453.

Wofsy, S.C., 2011. HIAPER Pole-to-Pole Observations (HIPPO): finegrained,global-scale measurements of climatically important atmospheric gases and aerosols. Philosophical Transactions of the Royal Society, 369:2073-2086.

Stone, R.S., A. Herber, V. Vitale, M. Mazzola, A. Lupi, R.C. Schnell, E.G.Dutton, P.S.K. Liu, S.M. Li, K. Dethloff, A. Lampert, C. Ritter, M. Stock,R. Neuber and M. Maturilli, 2010. A three-dimensional characterization of Arctic aerosols from airborne Sun photometer observations: PAMARCMIP,April 2009. Journal of Geophysical Research: Atmospheres,115:doi 10.1029/2009jd013605.

Schwarz, J.P., J.R. Spackman, R.S. Gao, L. Watts, P. Stier, M. Schulz,S.M. Davis, S.C. Wofsy and D.W. Fahey, 2010. Global-scale black carbon profiles observed in the remote atmosphere and compared to model. Geophysical Research Letters, 37:L18812,doi:10.1029/2010gl044372.

Schwarz, J.P., B.H. Samset, A.E. Perring, J.R. Spackman, R.S. Gao, P. Stier,M.G. Schultz, F.L. Moore, E.A. Ray and D.W. Fahey, 2013b. Global-scale seasonally re-solved black carbon vertical profiles over the Pacific.Geophysical Research Letters, 40:5542-5547.

Specific comments:

Abstract: You should mention in the abstract that these were balloon measurements up to 1200 meters height.

Page 2, L 17: 'to influence with semi-direct effects the atmospheric properties'. Could this be rewritten and explained further, maybe by 1 or 2 examples?

Page 2, L 34: You mention Arctic Haze here without explaining it. Since this is an important part of your study, I think you should briefly explain the phenomena with a few references (e.g. Stohl 2006)

Stohl, A. (2006), Characteristics of atmospheric transport into the Arctic troposphere, J. Geophys. Res., 111, D11306, doi:10.1029/2005JD006888.

Page 3, L 25: 'These reports may well highlight opposing forms of behavior '. I am not quite sure what this means?

Page 3, L 25: One reason for this difference between the observations could be the strong influence of biomass burning during spring 2008 (Warneke et al. 2010).

Warneke, C., K.D. Froyd, J. Brioude, R. Bahreini, C.A. Brock, J. Cozic, J.A.de Gouw, D.W. Fahey, R. Ferrare, J.S. Holloway, A.M. Middlebrook, L.Miller, S. Montzka, J.P.

Schwarz, H. Sodemann, J.R. Spackman and A.Stohl, 2010. An important contribution to springtime Arctic aerosol from biomass burning in Russia. Geophysical Research Letters, 37:L01801, doi:10.1029/2009GL041816.

Page 3, L 30: Drop 'should', as this is written it seems like you tell the emissions to do so? In bracelets: we do not know for sure if these emissions will warm the surface and be deposited, but as you write above; studies show that there are higher probability for this to happen when the concentrations are located close to the surface.

Page 4, L 23: Could you add just one sentence summarizing this table? E.g. 25 measurement days, balloons measured 2-14 profiles each day, altitude range?

Page 5, L 5: Is there a reference for this instrument and the calculated uncertainty in mass concentrations?

Page 6; L 18: Could you briefly here explain what you mean when the atmosphere is 'stable' and does not encourage vertical mixing? (in terms of potential temperature)

Page 8, L 28: In this paragraph you define the 3 modes of particles, 'Aitken', 'Accumulation' and 'course' and say that you will also use these names for the rest of the discussion, but most of the time you use N14-260, N260-1200, and N>1200 anyway. I suggest you use the names Aitken etc. throughout the text once you have defined them, as this is easier to read.

Page 8, L 36: Since there are many figures in this paper; I suggest removing fig 2 (or move to the supplementary).

Page 9: There are various methods to measure BC concentrations, and they can disagree by a factor of seven or more (Petzold et al 2013). Since the (common) filter-based method like you have used is not a direct measurement of BC, it is recommended to report the resulting BC concentration (eBC) together with the assumed MAC value. Maybe you should change 'BC' to 'eBC' to make sure that we know that this is equivalent BC? I also think you should add a brief discussion on how your measurements

depend on the assumed MAC number (you use 12.5 m2/g?) (or at least make a note about this).

Petzold, A., J.A. Ogren, M. Fiebig, P. Laj, S.M. Li, U. Baltensperger, T. Holzer-Popp, S. Kinne, G. Pappalardo, N. Sugimoto, C. Wehrli, A. Wiedensohler and X.Y. Zhang, 2013. Recommendations for reporting "black carbon" measurements. Atmospheric Chemistry and Physics, 13:8365-8379.

Page 9: Filter-based methods are sensitive to absorbing and non-absorbing non-BC particles. Could you please add a few sentences about the uncertainties in your method as well?

Page 11, L 15: 'Figure S1a shows a larger interannual springtime variability.' Of what?

Page 11, L 13 - 20: Since you are referring to figures in the supplement, maybe you could rewrite this paragraph so this is easier to follow without the figures? Not 'Fig Sxx shows ..' but instead just briefly state that the spring season had surface temperatures close to the climatology, summer season had .. etc. and then mention that figures are in Supplementary (–OR- move the sup. figures to the paper, but then you already have many figures there).

P11: 'Particularly, the maximum wind speeds registered at ground during balloon flights in spring 2011 and in summer 2011 2012 were 4.9 m s and 10.7 m s lower than the absolute wind speeds registered during the same periods: 27.9 m s and 16.3 m s.' I'm not sure if I understood this. The absolute winds measured by ..? With movement? How do you conclude that the measurement periods are representative for days with low winds?

Page 11, line 32 - page 13, line 14: I think you spend too much time explaining figure 3. Parts of this can be moved to Methods, e.g. what type of information you can retrieve from the measurements. You can also move parts to Introduction as a way of motivating the study. When I read the 'Results'-chapter I want to read about the results

right away. Could you also try to merge some of this information when you present the other results? I would skip everything between L31, p11 to L21,p12 and start on 'An example ..'. Is fig 3 needed at all? Why cannot the measured potential temperature and the RH for each group be plotted in fig 5 instead?

On the other hand, figure 7 is hardly mentioned. Can the wind roses be put in better context with the profiles described in 3.3.1-3.3.4? Also, this text is a bit hard to read, because of all the numbers listed. Do you need to list them all? Maybe put them in a table?

Page 17, L3: does this text and forward belong to 3.3.4 or should it have a separate heading?

Page 18: anything that has to do with methodology should be under Methods, not Results.

Page 19, L16: what is meant by a 'meaningful' impact of ship emissions?

Page 20: It is interesting to see the impact of ship emissions. Could you remind us here how far the measurements were from the ships? This also relates to your final conclusion on page 21 (where you suggest that increased shipping could significantly increase BC concentrations during summer and enhance climate change in the Arctic). Currently, BC emissions from shipping in the Arctic comprise a small fraction of within-Arctic BC. Browse et al. 2012 found that even under a high-projection of shipping, by 2050 BC emissions from shipping would still contribute less than 1 % of total Arctic deposition. Do you suggest that current emission inventories are too low and that future emission projections should also be higher?

Browse, J., K.S. Carslaw, S. Arnold, K.J. Pringle and O. Boucher, 2012. The scavenging processes controlling the seasonal cycle in Arctic sulphate and black carbon aerosol. Atmospheric Chemistry and Physics,12:6775-6798.

Page 20, L22: 'forbidden' – by who/what? What is meant by: '. And the locally formed

aerosol becomes in summer' ?

Do you find any (systematic) correlation between the different vertical profiles and the measurements at the ground? E.g. for special ground conditions, one can assume (with some certainty) a particular profile? Or that when using ground measurements (which are more abundant) when comparing to models, it is not such a bad assumption?

Minor comments/technical corrections (language etc):

Line 28 page 2: write the Q as a full sentence, e.g. How does the aerosol (. . ..) vary by season?

Page 2, line 31: 'Very pronounced' → drop 'very'

Page 3, line 1: know → known

There are several long sentences in this paper, which makes it a bit hard to read. E.g. Page 3. Line 2-7 is one sentence over 6 lines. Could this be split in 2? Also in this sentence: 'leads' → 'could lead'.

Page 4, line 28: form → from

Page 6, line 5 double ..

Page 6, line 13: operates since 2009 → 'have operated'

Page 6, line 21: 'during snow covered or not periods' Please rewrite.

Page 8, line : closets → closest

Page 11, line 2: 'Aerosol and BC and vertical profile (. . .)' Please rewrite. By vertical profile do you mean the meteorological fields? And aerosol are the size distributions?

Page 11, Line 10: 'Before to introduce' .. please change

P 11, L 24: 'Moreover, quite all measurements were conducted' quite all? You mean

'all'?

P11, L31: drop 'now'

P11, L34: 'Several information can be derived' please rewrite

P20, L24: reasing –> rising?

---

## Referee Comment (RC2) · Anonymous Referee #2 · 19 May 2016

This manuscript describes vertical profiles of aerosol number density over Arctic during spring and summer and presents authentic and original scientific material that has relevant implications for atmospheric science (aerosol, clouds, CCN, and others). This study is based on very important aerosol data over Ny-Ålesund, Svalbard, although the tethered-balloon-borne aerosol measurements are restricted to the good weather (i.e., clear sky, calm winds etc.). On the whole, the topic of the manuscript is relevant and suitable for the scope of the "ACP". However, there are several points which require some careful revision and corrections before publication.

General comments: 1. Quality of English I found many typo, miss-spell, and grammatical errors (e.g., location of ",").

2. Comparison with previous airborne aerosol measurements Several airborne aerosol measurements have carried out in Arctic area since 2000, for instances, ASTAR (2000, 2004, and 2007), ARCTAS (2008), ARCPAC (2008), and PAMARCMiP (2009 and 2011). Particularly, ASTAR campaigns were made around Svalbard. I suggest strongly that your data are compared to these previous results, and that these campaigns should be added into description of introduction.

3. Classification of aerosol type In this study, authors classified aerosol profiles into four groups. I agree with the classification of aerosol profiles. Unfortunately, typical weather/meteorological conditions and air mass origins in each type were not mentioned in the text. These information is very important to characterize vertical features of aerosols in Arctic region, and to be compared to aerosol data taken in the other project.

4. Relation between aerosol vertical profiles and structure of boundary layer Vertical features of aerosols in the lower troposphere are associated with the structure of boundary layer (i.e., surface inversion and height of boundary layer). What is typical height of top of boundary layer in each type? Aerosol data should be compared to vertical structure of boundary layer (surface inversion and top of the layer).

5.

Specific comments 1. Abstract: Height range of aerosol measurements are added in the text of abstract.

2. Page 5 Line 11, Unit of conductivity: M $\Omega$ cm-1 (not M $\Omega$ cm)

3. Page 9 – 10, Sensitivity (detection limit) of BC measurement In general, high flow rate is required for BC measurements in regions with lower BC concentration. Flow rate for BC measurement was 2.5 X 10-6 m3 s-1 (0.15 L min-1) in this study. I understand that authors chose the largest flow rate of the micro-aethalometer. However, this flow rate might be not enough in lower BC concentrations. Although BC concentration

increases during winter –spring in the Arctic regions, the BC level is lower than that in the mid-latitudes. What is the sensitivity (detection limit) of BC measurement in your measurement setting and analytical procedures?

4. Page 10, Aerosol stratification height (ASh) Procedure for ASh estimation should be mentioned in the text.

5. Sections 3.1 - 3.2 In the sections of 3.1 – 3.2, typical examples of vertical profiles of aerosols and meteorological parameters were mentioned. The description is slightly redundant. Some sentences can be simplified. In addition, general vertical structure of the boundary layer over Ny-Ålesund (i.e., thickness of surface inversion layer, and height of top of boundary layer) should be mentioned to understand characteristics of the vertical profiles. The vertical structure is associated closely with vertical features of aerosols.

6. Figure 5 In addition to four groups, general vertical profiles (all data) should be shown in the Figure. The general profiles can be useful, when authors want to know general (average) vertical profiles. Because the vertical profiles of aerosols related to profiles of meteorological parameter (potential temperature and relative humidity), mean profiles of meteorological parameter (or normalized meteorological parameter) should be shown together with those of aerosols.

7. Page 15 Line 8 Before statement about Figure 6, Figure 7 and explanation appeared here. Check or arrange figure number or description in the text.

8. Page 15 Line 30 – 34 It is true that wet removal processes have an impact on aerosol number density, but dry deposition make an important contribution to the aerosol number density, especially in coarse particles.

9. Page 15 Line 35-39 Air mass origins was shown and explained in only in "positive gradient profiles". However, transport pathway should be discussed together with aerosol source areas to show "plume" transport. In addition, general pattern and characteristics of transport processes in each type should be shown and discussed to understand relation between vertical profiles of aerosols and air mass origins.

10. Page 15 Line 38- Some of particulate organics can be derived from secondary formation. However, there are matters to be discussed whether organics can play an important role in "new particle formation" or not. Actually, organics are condensable vapors to grow aerosol particles in ultrafine mode. So, authors should distinguish between new particle formation and secondary formation and mention them in the text.

11. Section 3.4 Different condition of solar radiation between spring and summer can engender change of height of top of the boundary layer. This change is very important to vertical profiles of aerosols and meteorological parameters. Other comments about section 3.4 is similar to previous comments about section 3.3.

---

## Referee Comment (RC3) · Anonymous Referee #3 · 6 Jun 2016

This manuscript is based on vertical profiles of aerosol number density, eBC and ground measurements of the above at the Ny Alesund Arctic research station. This study is providing very useful data for the vertical structure of the aerosol column at a well studied area, where this type of data are still missing.

As a general outcome, the topic of the manuscript is relevant and suitable for the scope of "ACP". However, there are several points where the manuscript is failing to follow and deliver the methods and data quality needed for this study.

General comment: The description of the vertical structure of the atmosphere is well documented and useful and the classification of the different structures useful to relate to known aerosol properties based on the aerosol number size distributions from OPCs.

[Figure]

There is also a good documentation of aerosol contamination events from harbour traffic of large boats

Major problems can be identified as follows:

1. The classification and discussion of results is not based on the understunding we can derive for the aerosol microphysics based the origin of aerosol during the study.

2. There is no attempt to compare with data obtained by numerous studies in the area using aircraft or lidar techniques. Although several studies are mentioned no quantitative comparsion is given at least for the ground measuremnts or data published.

3. The use of micro aethalometers in this area can be only used for obtaining EBC concentrations at minimum concentrations, which the authors have yet to derive. They show in figure 2c) a good correlation between the two micro-aethalometers used. This also shows an uncertainty at a 100% level for concentrations below 30 ug/m3 The other serious flaw in the processing of these data is the calculation of the absorption coefficient using a well established methodology and an unrealistic "C" factor. They quote a study in Milan where the "C" factor was derived for urban concentration levels and mixture of urban aerosol species. The authors must remove all absorption coefficients calculated in this manner and reported in this manuscript.

4. The chemical composition reported in figure 8 is given only in % of the total mass. How is the total mass derived and what are the actual mass concentrations of the different species reported in otherwise incredible detail where the non sea salt and non crustal fractions are calculated? These data do not appear realistic. For example in most cases the EC is found to 0.1 % of the aerosol mass. If one assumes that in the worst case eBC and EC mass concentrations can differ by a factor of 2 (+/- 100%) in the HO case where the eBC is found on average at 25 ng/m3 the aerosol mass concentration levels would range between1 to 50 ug/m3. This upper limit is totally unrealistic and even the 25 ug/m3 is extremely high. The other cases would produce even more grossly biased results.

This comparison puts in doubt the whole dataset of eBC and chemical data leaving the OPC and ground SMPS measurements as the only dataset worth considering for this manuscript.

---

## Author Comment (AC1) · 21 Jul 2016

**Response to Reviewer#1**

*We thank the reviewer for his or her helpful comments and insight. We respond to the general and to the specific points below. All the comments are addressed in the revised manuscript. As requested, the whole text was proofread and edited, to eliminate the typos and to improve the language.*

General Comment 1: This study reports vertical profiles of aerosol number size distribution and black carbon (BC) concentrations from balloon measurements during a field campaign in Ny-Ålesund, Svalbard spring and summer 2011-2012. The authors divide the number size distribution into 3 modes and classify the vertical profiles in four shapes during spring. The authors also discuss secondary aerosol formation and emissions from shipping during summer.

This study is important because 1) the vertical distribution of aerosols affects its radiative forcing and 2) measurements of the vertical distribution of aerosols in the Arctic are particularly sparse. Since such measurements are highly needed and valuable, I think this study is relevant and within the scope of ACP. The Method section is clearly outlined and the different instruments used are sufficiently and well explained. The figures are nice and easy to follow. However, I think the overall presentation of the results should be improved before it can be published. The manuscript needs more work in terms of language and structure. If this can be achieved, I recommend the manuscript for publication.

*Answer to the General Comment 1 (AGC)1: Thank you very much for your comment which underline the experimental efforts and the high relevance of the results presented in our paper. We agree with you that an improved organization of the manuscript and a better presentation of the results is necessary. For this reason, we managed the paper accordingly to your suggestions (here below answered). The whole text was also proofread and edited, to eliminate the typos and to improve the language.*

General comment 2. The quality of the English language in this manuscript is variable (some parts are good, but others less good), and I think it would benefit by a thoroughly review of the language (and a spell check!). I have added a few examples under minor comments.

*AGC2: The manuscript was proofread and edited, to eliminate the typos and to improve the language as required. A particular attention was given to shorten the several long sentences present in the paper, as also required in your minor comment 4. Thank you for the suggestion.*

General comment 3. The Results section would be easier to read if it was shortened a bit. Description of the methodology should always be under Methods, not Results. I have a few specific suggestions below.

*AGC3: Thank you for this comment. We agree with you. The description of the methodology (presently in the result section) was moved to the method section. Moreover, sections 3.1 and 3.2 were shortened and merged together in the revised version of the paper.*

General comment 4. I miss a broader implication of this study. Why did you separate the profiles into the four shapes? Comprehensive measurement studies like this can provide physical understanding for evaluation/improvement of the modeling of aerosol processes. Do you have any suggestions? I understand that you cannot add any modeling, but I would like to know more what we can learn from this study.

*AGC4: Thank you for this question. Out answer is also related to that reported below for your specific comment SC22 (see our answer ASC22).*

*We separated the profiles in the four shapes because each shape is the result of an interplay of several processes: 1) transport events, 2) the planetary boundary layer dynamics and 3) the local formation of aerosol. The different combinations of these factors result in a specific profile class.*

*Figure 6 represents a good example in which the transport of polluted air masses from mid-latitudes generated initially PG profiles that naturally evolved (due to the entrance into the PBL) into a NG profile.*

*Even though a modelling simulation is beyond the scope of the present paper, some indication can be obtained. One of this is related to your question about the validity of ground-based measurements with respect to the vertical aerosol distribution in modelling comparison (SC21). HO profile showed that ground measurements are fully representative of the vertical column (up until ~1 km, our vertical limit) while during NG and PG profiles the ground based measurements are representative for the column up to the PBL. DNG profiles show that ground-based measurements differ from the measurements performed within the column. However, the last case is influenced by secondary aerosol formation that can be easily detected by an SMPS.*

*Thus ground-based measurements (coupled with a proper PBL determination) are fundamental for model validation.*

General comment 5. Measurements of vertical profiles in the Arctic are sparse, but there are a few, e.g.:

- two ARCTAS campaigns in the North American Arctic (Jacob et al. 2010) in April and June/July 2008
- the ARCPAC campaign conducted together with ARCTAS in spring 2008 (Brock et al. 2011)
-the PAMARCMIP campaign in April 2009 (Stone et al. 2010) -the HIPPO campaign (Schwarz et al. 2010, 2013; Wofsy 2011) January and October 2009 + winter and autumn 2009
- the ARCTAS/ARCPAC campaign in spring 2008,
- the ARCTAS campaign in summer 2008
- the PAMARCMIP campaign in spring 2009.

On a general basis; How are those compared to your study? I suggest you also include more of these studies in the introduction.

Jacob, D.J., J.H. Crawford, H. Maring, A.D. Clarke, J.E. Dibb, L.K. Emmons,R.A. Fer- rare, C.A. Hostetler, P.B. Russell, H.B. Singh, A.M. Thompson,G.E. Shaw, E. McCauley, J.R. Pederson and J.A. Fisher, 2010. The Arctic Research of the Composition of the Troposphere from Aircraft and Satellites (ARCTAS) mission: design, execution, and first results. Atmospheric Chemistry and Physics, 10:5191-5212.

Brock, C.A., J. Cozic, R. Bahreini, K.D. Froyd, A.M. Middlebrook, A.McComiskey, J. Brioude, O.R. Cooper, A. Stohl, K.C. Aikin, J.A. De Gouw,D.W. Fahey, R.A. Ferrare, R.-S. Gao, W. Gore, J. Holloway, G. Hubler, A.Jefferson, D.A. Lack, S. Lance, R.H. Moore, D.M. Murphy, A. Nenes,P.C. Novelli, J.B. Nowak, J.A. Ogren, J. Peischl, R.B. Pierce, P. Pilewskie,P.K. Quinn, T.B. Ryerson, K.S. Schmidt, J.P. Schwarz, H. Sode- mann, J.R.Spackman, H. Stark, D.S. Thomson, T. Thornberry, P. Veres, L.A. Watts, C.Warneke and A.G. Wollny, 2011. Characteristics, sources, and transport of aerosols measured in spring 2008 during the aerosol, radiation, and cloud processes affecting Arctic climate (ARCPAC) project. Atmospheric Chemistry and Physics, 11:2423-2453.

Wofsy, S.C., 2011. HIAPER Pole-to-Pole Observations (HIPPO): finegrained,global- scale measurements of climatically important atmospheric gases and aerosols. Philo- sophical Transactions of the Royal Society, 369:2073-2086.

Stone, R.S., A. Herber, V. Vitale, M. Mazzola, A. Lupi, R.C. Schnell, E.G.Dutton, P.S.K. Liu, S.M. Li, K. Dethloff, A. Lampert, C. Ritter, M. Stock,R. Neuber and M. Maturilli, 2010. A three-dimensional characterization of Arctic aerosols from airborne Sun pho- tometer observations:

PAMARCMIP,April 2009. Journal of Geophysical Research: Atmospheres,115:doi 10.1029/2009jd013605.

Schwarz, J.P., J.R. Spackman, R.S. Gao, L. Watts, P. Stier, M. Schulz,S.M. Davis, S.C. Wofsy and D.W. Fahey, 2010. Global-scale black carbon profiles observed in the remote atmosphere and compared to model. Geophysical Research Letters, 37:L18812,doi:10.1029/2010gl044372.

Schwarz, J.P., B.H. Samset, A.E. Perring, J.R. Spackman, R.S. Gao, P. Stier,M.G. Schultz, F.L. Moore, E.A. Ray and D.W. Fahey, 2013b. Global-scale seasonally re- solved black carbon vertical profiles over the Pacific.Geophysical Research Letters, 40:5542-5547.

*AGC5: Thanks for this comment.*

*An important comparison was reported in the manuscript, page 15 (lines 10-14), where "the columnar averages of both total aerosol number and BC concentrations [236.1±23.9 cm$^{-3}$ ($N_{14-260}$), 21.1±1.3 cm$^{-3}$ ($N_{260-1200}$), 0.2±4\*10$^{-2}$ cm$^{-3}$ ($N_{>1200}$) and 52±8 ng m$^{-3}$ (BC)]" were successfully compared with long-term data series collected over Ny-Ålesund at the Zeppelin observatory (Eleftheriadis et al., 2009; Tunved et al., 2013) during Spring.*

*This comparison underlined the accuracy of the collected data and, most importantly, suggested that all the investigated vertical profile classes may influence the background Arctic aerosol measured by Arctic observatories within GAW and EMEP observation programmes.*

*Moreover, at page 3 (lines 23-25) we cited some of the Arctic campaigns (i.e. Kupiszewski et al., 2013; Schwarz et al., 2010).*

*However, we fully agree with you that a better contextualization of the measuring campaign is required.*

*Thus, we modified the introduction section, adding and discussing the suggested campaigns (and related references). Moreover, we discussed the obtained results with respect to the same references.*

Specific Comment 1 (SC1): Abstract: You should mention in the abstract that these were balloon measurements up to 1200 meters height.

Answer to the Specific Comment 1 (ASC)1: *Thank you very much, we modified the abstract accordingly to your suggestion.*

SC2: Page 2, L 17: 'to influence with semi-direct effects the atmospheric properties'. Could this be rewritten and explained further, maybe by 1 or 2 examples?

*ASC2: The sentence was rewritten as follows: "Many of these effects can be altered by the aerosol due to its ability to absorb and scatter solar radiation (direct effect) or to seed and modify the cloud properties (indirect effects). In addition, light absorption by BC can alter the atmospheric thermal structure within, below, or above clouds, consequently affecting cloud distributions (IPCC, 2013; Bond et al., 2013; Ramanathan and Feng, 2009; Koren et al. 2008; Koren et al., 2004; Kaufman et al., 2002)".*

SC3: Page 2, L 34: You mention Arctic Haze here without explaining it. Since this is an important part of your study, I think you should briefly explain the phenomena with a few references (e.g. Stohl 2006). Stohl, A. (2006), Characteristics of atmospheric transport into the Arctic troposphere, J. Geophys. Res., 111, D11306, doi:10.1029/2005JD006888.

*ASC3: Thanks for raising this point, we agree with you that a better description of the Arctic Haze could help the reader. For this reason, we added to the introduction the following description: "The Arctic Haze is an effect, where an inflow of pollution (aerosol and gases) from northern mid-latitudes during winter-spring, result in a reduction in visibility (Jacob et al., 2010; Sthol et al.,*

*2006; Radke et al., 1984; Barrie and Hoff, 1985; Brock et al., 1989; Shaw, 1995). The Arctic Haze manifests itself during special meteorological conditions (thermally stable stratifications with frequent and persistent occurrences of surface-based inversions) during which the air pollution can be transported into the Arctic at low-level (followed by ascent in the Arctic or low-level alone) or with an uplift outside the Arctic, followed by descent in the Arctic itself (Sthol, 2006)."*

SC4: Page 3, L 25: 'These reports may well highlight opposing forms of behavior '. I am not quite sure what this means?

*ASC4: The intention was to underline the differences in the vertical aerosol behavior found during the reported field campaigns. However, to avoid any confusion, we delated this sentence and replaced it with your suggestion in the following specific comment.*

SC5: Page 3, L 25: One reason for this difference between the observations could be the strong influence of biomass burning during spring 2008 (Warneke et al. 2010).
Warneke, C., K.D. Froyd, J. Brioude, R. Bahreini, C.A. Brock, J. Cozic, J.A.de Gouw, D.W. Fahey, R. Ferrare, J.S. Holloway, A.M. Middlebrook, L.Miller, S. Montzka, J.P. Schwarz, H. Sodemann, J.R. Spackman and A.Stohl, 2010. An important contribution to springtime Arctic aerosol from biomass burning in Russia. Geophysical Research Letters, 37:L01801, doi:10.1029/2009GL041816.

*ASC5: Thank you very much for the suggestion, we added the reference to the introduction section.*

SC6: Page 3, L 30: Drop 'should', as this is written it seems like you tell the emissions to do so? In bracelets: we do not know for sure if these emissions will warm the surface and be deposited, but as you write above; studies show that there are higher probability for this to happen when the concentrations are located close to the surface.

*ASC6: Thank you. We modified the sentence.*

SC7: Page 4, L 23: Could you add just one sentence summarizing this table? E.g. 25 measurement days, balloons measured 2-14 profiles each day, altitude range?

*ASC7: Thank you for the suggestion. The new sentence is as follows: "Table 1 lists the dates of the campaign (25 measurement days), the number of flights (197 measured profiles), the maximum altitudes (~700-1300 m) and the cloud base height (clouds present for 48% of the campaign)."*

SC8: Page 5, L 5: Is there a reference for this instrument and the calculated uncertainty in mass concentrations?

*ASC8: As stated at page 5, line 5, we used the 5-digit Sartorius ME235P microbalance. In this respect, 5% is the uncertainty related to the weighing procedure experimentally evaluated. The text is changed as follows: reproducibility error on Filter weighing was lower than 5% (experimentally evaluated).*

SC9: Page 6; L 18: Could you briefly here explain what you mean when the atmosphere is 'stable' and does not encourage vertical mixing? (in terms of potential temperature)

*ASC9: We modified the sentence as follows: "the atmosphere is stable for about 50% of the time along the year. The term stability refers to the propensity of air masses to move vertically: stable air resists to vertical motion, while unstable air masses are prone to vertical movements. A parcel of air results to be stable/unstable if the temperature lapse rate is lower/higher than the adiabatic*

*one, i.e. if the potential temperature is increasing/decreasing with height. Air within a stable layer is not turbulent and these conditions will cause pollutants to become trapped near ground level, as the vertical mixing of the aerosols is not encouraged".*

SC10: Page 8, L 28: In this paragraph you define the 3 modes of particles, 'Aitken', 'Accumulation' and 'coarse' and say that you will also use these names for the rest of the discussion, but most of the time you use N14-260, N260-1200, and N>1200 anyway. I suggest you use the names Aitken etc. throughout the text once you have defined them, as this is easier to read.

*ASC10: Thank you for addressing it. With the sentences reported at page 8, line 28 we wanted to explain the meaning of each investigated broadsize range. We consider your comment while preparing the revised version of the manuscript.*

SC11: Page 8, L 36: Since there are many figures in this paper; I suggest removing fig 2 (or move to the supplementary).

*ASC11: Figure 2 was moved to the supplemental material.*

SC12: Page 9: There are various methods to measure BC concentrations, and they can disagree by a factor of seven or more (Petzold et al 2013). Since the (common) filter-based method like you have used is not a direct measurement of BC, it is recommended to report the resulting BC concentration (eBC) together with the assumed MAC value. Maybe you should change 'BC' to 'eBC' to make sure that we know that this is equivalent BC? I also think you should add a brief discussion on how your measurements depend on the assumed MAC number (you use 12.5 m2/g?) (or at least make a note about this).
Petzold, A., J.A. Ogren, M. Fiebig, P. Laj, S.M. Li, U. Baltensperger, T. Holzer-Popp, S. Kinne, G. Pappalardo, N. Sugimoto, C. Wehrli, A. Wiedensohler and X.Y. Zhang, 2013. Recommendations for reporting "black carbon" measurements. Atmospheric Chemistry and Physics, 13:8365-8379.

*ASC12: Petzold et al. (2013), Andreae and Gelencser (2006) and other authors (Gilardoni et al., 2010; Sthol et al., 2013; Eckhardt et al., 2013), suggest reporting BC concentrations with the term eBC, arising from the need to report the method to determine BC and the parameters used in the method. When optical methods are used to measure light transmission through the filter loaded with BC, the mass equivalent concentration is determined using the mass attenuation cross-section ($\sigma_{ATN}$). For the case of our measurements, we report the $\sigma_{ATN}$ value 12.5 $m^2$ $g^{-1}$ We note this value (Ferrero et al., 2011) in the method section, as the one used in the micro-Aethalometer AE51, page 9, line 23. This approach was also used by Eleftheriadis et al. (2009) when reported ten years of BC measurements in Ny-Ålesund.*
*Moreover, the $\sigma_{ATN}$ value (12.5 $m^2$ $g^{-1}$) "was obtained by comparing the BC values measured with the microAeth® Model AE51, with an AE31 Aethalometer (880 nm wavelength) operating in a test chamber with different BC concentrations at low attenuation values. The comparison was then repeated using ambient air" (as reported in Ferrero et al., 2011).*
*This value is not far from the $\sigma_{ATN}$ values of 15.2 $m^2$ $g^{-1}$ and 15.9 $m^2$ $g^{-1}$ reported in Eleftheriadis et al. (2009) and applied to the attenuation coefficient measured at the Zeppelin station (Ny-Ålesund) with the Aethalometers AE9 and AE31, respectively. The difference between these values results from the use of different filter materials to collect the sample in the different Aethalometers, which was quantified in Ferrero et al. (2011) and Drinovec et al (2015).*
*We compare our results with those previously measured at Ny Ålesund in the manuscript, page 15, lines 4-8, where we state: "the columnar average of BC concentrations obtained by averaging the profile classes was 52±8 ng $m^{-3}$ (eBC)". This value "perfectly agrees with long-term data series collected over Ny-Ålesund at the Zeppelin observatory (Eleftheriadis et al., 2009) during Spring."*

*Finally, considering the relationship between the attenuation and absorption coefficients (page 9, equation 9), the apparent mass attenuation cross-section of 12.5 $m^2 g^{-1}$ corresponds to the mass absorption cross-section (MAC) of 6.1 $m^2 g^{-1}$ (using C=2.05±0.03). As suggested by Petzold et al. (2013), we report the wavelengths, the mass attenuation cross-section and the mass absorption cross-section used in the determination of the absorption coefficients, using the methodology reported in Weingartner et al. (2003). The MAC of 6.1 $m^2 g^{-1}$ is in agreement with the previously published range of values (see for example Petzold et al. (2013) and references therein).*

SC13: Page 9: Filter-based methods are sensitive to absorbing and non-absorbing non-BC particles. Could you please add a few sentences about the uncertainties in your method as well?

*ASC13: Considering the absorbing non-BC particles, different types of aerosol may in principle contribute to the signal in Aethalometers (i.e. Brown Carbon, dust). However, BrC is characterized by negligible absorption in the infrared (Andreae and Gelencsér, 2006), the wavelength range of the eBC measurements (micro-Aeth AE51 uses 880 nm). In this respect, Massabò et al. (2013) show the potential contribution of BrC to the determination of eBC to be below 10%.*
*To estimate the possible influence of BrC on eBC measurements carried out during the spring 2011 campaign, the few data collected with the micro-Aeth prototype AE5x at 370 and 880 nm were considered. Particularly, the Aethalometer model (Sandradewi et al., 2008) was applied to the apportionment of absorption due to both BC and BrC as reported in Massabò et al (2013) and in Shamjad et al. (2015) as follows:*

$$\frac{b_{abs}(370\,nm)_{BC}}{b_{abs}(880\,nm)_{BC}} = \left(\frac{370}{880}\right)^{-\alpha_{BC}} \tag{1}$$

$$\frac{b_{abs}(370\,nm)_{BrC}}{b_{abs}(880\,nm)_{BrC}} = \left(\frac{370}{880}\right)^{-\alpha_{BrC}} \tag{2}$$

$$b_{abs}(\lambda) = b_{abs}(\lambda)_{BC} + b_{abs}(\lambda)_{BrC} \tag{3}$$

*where $\alpha_{BC}$ and $\alpha_{BrC}$ represent the Absoprtion Angstrom Exponents of BC and BrC, respectively. The value for $\alpha_{BC}$ was taken to be 1 as suggested by Massabò et al (2013) and Sandradewi et al. (2008), while $\alpha_{BrC}$ was set at 3.5 (Yang et al., 2009), 3.95 (Massabò et al., 2013), 6.6 (Shamjad et al., 2015) and 9.0 (Bikkina and Sarin, 2013), respectively. With this inputs, the percentage of absorption at 880 nm due to BrC instead that BC was 8.5%, 5.8%, 0.5% and 0.1%, respectively. Thus, it is possible to estimate that the BrC positive artifact on eBC measurements was less than 10% during the campaign.*
*The effect of non-absorbing particles on the filter photometer measurements was also quantified for the Aethalometer AE33 Drinovec et al (2015), which uses the same filter material as the AE51, and was shown to be below 2.5%. To reduce the effect of the non-absorbing particles sampled onto the filter, the experimental protocol during the campaign followed that reported in Ferrero et al. (2011): "all vertical BC profiles were conducted by changing the filter ticket after each profile. As a result, ATN never achieved values higher than 20 during all profiles. The average ATN measured along vertical profiles was 5±1". This means that the total amount of aerosol collected on each filter during the vertical profile was very low resulting in a negligible effect of non-absorbing particles.*
*We added the aforementioned considerations to the supplemental material and referenced them in the method section.*

SC14: Page 11, L 15: 'Figure S1a shows a larger interannual springtime variability.' Of what?

*ASC14: Thanks for addressing this point, with this sentence we meant that Figure S1a shows a larger interannual springtime variability of temperature measured close to the ground. Due also to your question here below (SC14) we rewrite the manuscript text at page 11, lines 13-20 with the following sentence: "The temperature measured in spring 2011 was within the standard deviation range of the long-term observations, while a 10-day period at the end of April 2011 was slightly warmer than the climatological mean (Figure S1a). The temperatures during the summer seasons 2011 and 2012 were mostly within the range of the long-term observations (Figure S1b). Neither of the campaign periods was conducted under exceptional meteorological conditions, so the vertical profile measurements can be considered to have been obtained under typical meteorological conditions representative for the Ny-Ålesund environment".*

SC15: Page 11, L 13 - 20: Since you are referring to figures in the supplement, maybe you could rewrite this paragraph so this is easier to follow without the figures? Not 'Fig Sxx shows ..' but instead just briefly state that the spring season had surface temperatures close to the climatology, summer season had .. etc. and then mention that figures are in Supplementary (–OR- move the sup. figures to the paper, but then you already have many figures there).

*ASC15: We modified the paper accordingly to your observation.*

SC16: P11: 'Particularly, the maximum wind speeds registered at ground during balloon flights in spring 2011 and in summer 2011 2012 were 4.9 m s and 10.7 m s lower than the absolute wind speeds registered during the same periods: 27.9 m s and 16.3 m s.' I'm not sure if I understood this. The absolute winds measured by ..? With movement? How do you conclude that the measurement periods are representative for days with low winds?

*ASC16: The wind speed was measured at the Amundsen-Nobile Climate Change Tower as described in section 2.1.2 (page 6, lines 13-16). The wind speed (average, max value) measured during balloon flights was lower than that during the whole period (April 2011, June and July 2011-2012) of the campaign. Thus, we reported that balloon flights have limitations with respect to its launch conditions, in particular they favor low wind conditions as it is very difficult to launch the balloons during high winds. We understand from your question that the sentence was not clear. Thus, we rephrased it as follows: "It should be reminded, however, that the tethered balloon has limitations with respect to its launch conditions (section 2.2). Particularly, balloon profiles are measured in low wind conditions, as it is very difficult to launch the balloons during high winds – this introduces a bias in respect to average meteorological conditions above the launch site. The maximum wind speed measured at the Amundsen-Nobile Climate Change Tower (section 2.1.2) during balloon flights was lower than that during the whole period of the campaign (April 2011, June and July 2011-2012): 4.9 m s$^{-1}$ and 10.7 m s$^{-1}$ (springtime and summertime balloon profiles) compared to 27.9 m s$^{-1}$ and 16.3 m s$^{-1}$ (full spring 2011 and summer 2011-2012)."*

SC17: Page 11, line 32 - page 13, line 14: I think you spend too much time explaining figure 3. Parts of this can be moved to Methods, e.g. what type of information you can retrieve from the measurements. You can also move parts to Introduction as a way of motivating the study. When I read the 'Results'-chapter I want to read about the results right away. Could you also try to merge some of this information when you present the other results? I would skip everything between L31, p11 to L21,p12 and start on 'An example ..'. Is fig 3 needed at all? Why cannot the measured potential temperature and the RH for each group be plotted in fig 5 instead?
On the other hand, figure 7 is hardly mentioned. Can the wind roses be put in better context with the profiles described in 3.3.1-3.3.4? Also, this text is a bit hard to read, because of all the numbers listed. Do you need to list them all? Maybe put them in a table?

ASC17: _We agree with your observation. Particularly, as we reply in AGC3 to your general comment, all the description of the methodology presently in the result section was moved under the method section. Sections 3.1 and 3.2 were shortened and merged together in the revised version of the paper. We considered your suggestion for Figure 3 and we added averaged meteorological parameters to figure 5._
_Figure 7 was put in evidence in the revised version of the paper and a table resuming data of the campaign was prepared._

SC18: Page 17, L3: does this text and forward belong to 3.3.4 or should it have a separate heading?

ASC18: _Page 17, line 3 and the following lines belong to section 3.3.4. Section 3.3.4 is quite long, but we shortened it following your suggestion (see AGC3) to move the methodology present here in the method section._

SC19: Page 18: anything that has to do with methodology should be under Methods, not Results.

ASC19: _We agree with you and we modified the paper accordingly._

SC20: Page 19, L16: what is meant by a 'meaningful' impact of ship emissions?

ASC20: _The intention was to point out that profiles were affected by a local, high, plume emitted from the ships. We agree that the sentence, as stated, was not clear and thus we rephrased it as follows: "Type 2, profiles characterized by the presence of shipping emissions (hereinafter addressed as SP), Figure 12c-d"._

SC21: Page 20: It is interesting to see the impact of ship emissions. Could you remind us here how far the measurements were from the ships? This also relates to your final conclusion on page 21 (where you suggest that increased shipping could significantly increase BC concentrations during summer and enhance climate change in the Arctic). Currently, BC emissions from shipping in the Arctic comprise a small fraction of within- Arctic BC. Browse et al. 2012 found that even under a high-projection of shipping, by 2050 BC emissions from shipping would still contribute less than 1 % of total Arctic deposition. Do you suggest that current emission inventories are too low and that future emission projections should also be higher?
Browse, J., K.S. Carslaw, S. Arnold, K.J. Pringle and O. Boucher, 2012. The scavenging processes controlling the seasonal cycle in Arctic sulphate and black carbon aerosol. Atmospheric Chemistry and Physics,12:6775-6798.

ASC21: _Vertical profile measurements in summer were carried out from the German-French AWIPEV research base (78°55'24" N 11°55'15"E), 600 m far from the harbor (we added this information to section 2). Thus, the reported results (section 3.4.2, Figure 12c-d and Figure 13) refer to the local impact of ships._
_The obtained results are in agreement with those reported in Eckhardt et al. (2013), who showed an enhancement of 72 and 45 % (up to 81 and 72 % in stagnant conditions) of eBC, when ships cruised in the Kongsfjord, compared to values when ships were not present._
_From these results, yet Eckhardt et al. (2013) concluded that the eBC increase due to shipping emission can "be taken as a warning signal of future pan-Arctic conditions if Arctic shipping becomes more frequent and emission regulations are not strict enough"._
_On the other hand, as you addressed, shipping contribution was less than 1% of BC emissions in the Arctic as reported in in the study of Stohl et al. (2013) where, conversely, gas flaring was estimated to contribute 42% to the annual mean BC surface concentrations in the Arctic dominating the estimated BC emissions north of 66° N._

*However, even also Corbett et al. (2010) reported that "the magnitude of emissions from shipping on a mass basis may be modest compared to other anthropogenic sources, the proximity of activity to the Arctic may help explain regional effects important for global and regional climate change". The possible implications of shipping emissions were also addressed in the work of Sand et al. (2013), who underlined that the BC aerosols would be emitted by ships directly into the Arctic planetary boundary layer with a stronger interaction with the surface (both by deposition of BC on snow and ice and by radiative and sensible heat fluxes down to the surface).*

*Given the aforementioned considerations and the experimental conditions of vertical profile measurements, we rephrased the conclusion referring to the impact of shipping emission at a local and regional scale.*

*The emission inventories are beyond the scope of the present work. We report the increased number of ships (page 19, lines 37-40) and passengers (a proxy of ship dimensions) in summer 2012 (78 days with 138 ships) relative to summer 2011 (57 days with a total of 103 ships). The obtained results can be thus considered an important phenomenon which should remain under observation in the future due to the sensitivity of the Arctic environment.*

SC22: Page 20, L22: 'forbidden' – by who/what? What is meant by: '. And the locally formed aerosol becomes in summer' ?

Do you find any (systematic) correlation between the different vertical profiles and the measurements at the ground? E.g. for special ground conditions, one can assume (with some certainty) a particular profile? Or that when using ground measurements (which are more abundant) when comparing to models, it is not such a bad assumption?

*ASC22: Thank for addressing this point, the sentence was incomplete due to an erroneous application of tracking changes in the word processor. We rewrote the sentence as follows: "It is important to note that SP profiles were observed for the majority of cases in summer. In summer, the long-range transport of aerosol from mid-latitudes is minor (Browse et al., 2012; Quinn et al., 2008; Stohl et al., 2006) and the locally formed aerosol becomes dominant (Giardi et al., 2015; Tunved et al., 2013; Ström et al., 2009 and 2003)".*

*Figure 5 clearly shows that for HO profiles ground measurements are fully representative of the vertical column (up until ~1 km, our vertical limit); during NG and PG profiles the ground based measurements are representative for the column up to the planetary boundary layer. DNG profiles show that ground-based measurements differ from the measurements aloft. However, the last case is influenced by secondary aerosol formation that can be easily detected by an SMPS (or similar experimental devices).*

*Given the above considerations, in our opinion, ground-based measurements (coupled with a proper PBL determination) are fundamental and very useful for model validation.*

*We added these considerations in the conclusion section.*

Minor Comment 1 (MC1): Line 28 page 2: write the Q as a full sentence, e.g. How does the aerosol (. . ..) vary by season?

Answer to the Minor Comment 1 (AMC)1: *Done*

MC2: Page 2, line 31: 'Very pronounced' → drop 'very'

*AMC2: Done*

MC3: Page 3, line 1: know → known

*AMC3: Thanks, corrected.*

MC4: There are several long sentences in this paper, which makes it a bit hard to read. E.g. Page 3. Line 2-7 is one sentence over 6 lines. Could this be split in 2? Also in this sentence: 'leads' → 'could lead'.

*AMC4: We agree with you. We modified the sentence at page 3 and we shortened the long sentences present in the paper. Thanks for the suggestion.*

MC5: Page 4, line 28: form → from

*AMC5: Thanks, corrected.*

MC6: Page 6, line 5 double ..

*AMC6: Thanks, corrected.*

MC7: Page 6, line 13: operates since 2009 → 'have operated'

*AMC7: Thanks, corrected.*

MC8: Page 6, line 21: 'during snow covered or not periods' Please rewrite.

*AMC8: We rephrased the sentence as: "This is clearly related to a different heating of the ground related to seasonal changes of the snow covering".*

MC9: Page 8, line: closets → closest

*AMC9: Thanks, corrected.*

MC10: Page 11, line 2: 'Aerosol and BC and vertical profile (. . .)' Please rewrite. By vertical profile do you mean the meteorological fields? And aerosol are the size distributions?

*AMC10: We rephrased the sentence as: "Vertical profiles of aerosol number size distribution and BC concentrations were measured to assess changes in aerosol properties within the vertical column in the Arctic region". Here the intention was to draw the reader's towards the topic of this paper: the determination of the vertical behavior of the aerosol properties (number size distribution and BC concentration) in the Arctic.*

MC11: Page 11, Line 10: 'Before to introduce' .. please change

*AMC11: We rephrased the sentence as: "Here below, the ambient conditions under which the vertical profiles were measured are briefly described".*

MC12: P 11, L 24: 'Moreover, quite all measurements were conducted' quite all? You mean 'all'?

*AMC12: Table 1 summarizes the conditions during the measuring campaign. It can be observed that, during the majority of vertical profile measurements, clear sky conditions were present. Due to your question we rephrased the sentence to clarify this point.*

MC13: P11, L31: drop 'now'

*AMC13: Done*

MC14: P11, L34: 'Several information can be derived' please rewrite

*AMC14: We rephrased the sentence as: Figure 3a-d (and the whole ensemble of collected data), accurately describe the vertical distribution of the aerosol and its properties in the first kilometer above Ny-Ålesund.*

MC15: P11, L34: P20, L24: reasing –> rising?

*AMC15: Yes, rising. Thanks, corrected.*

[revised manuscript text omitted]

---

## Author Comment (AC2) · 21 Jul 2016

**Response to Reviewer#3**

*We thank the reviewer for his or her helpful comments and insight. We respond to the general and specific points below. All the comments are addressed in the revised manuscript. As requested, the whole text was proofread and edited, to eliminate the typos and to improve the language.*

General Comment 1: This manuscript is based on vertical profiles of aerosol number density, eBC and ground measurements of the above at the Ny Alesund Arctic research station. This study is providing very useful data for the vertical structure of the aerosol column at a well studied area, where this type of data are still missing.

As a general outcome, the topic of the manuscript is relevant and suitable for the scope of "ACP". However, there are several points where the manuscript is failing to follow and deliver the methods and data quality needed for this study.

The description of the vertical structure of the atmosphere is well documented and useful and the classification of the different structures useful to relate to known aerosol properties based on the aerosol number size distributions from OPCs. There is also a good documentation of aerosol contamination events from harbour traffic of large boats.

*Answer to the General Comment 1 (AGC)1: Thank you very much for your comment which underline the importance of the collected experimental data as well as the high relevance of the results presented in our paper. Here below we answer to the raised points.*

General comment 2. The classification and discussion of results is not based on the understunding we can derive for the aerosol microphysics based the origin of aerosol during the study.

*AGC2: The aim of the paper is to determine the seasonal phenomenology of the aerosol behavior along vertical profiles; this goal was chieved classifying the collected experimental data, according to their shape, both during spring and summer. Thus, attention was paid to the description of the vertical structure of the atmosphere and its relationship to the aerosol number size distribution.*

*We underline that, within the context of vertical profile classification, the aerosol microphysics and aerosol/air mass origin were investigated, when necessary, with the aim to deepen the understanding on aerosol vertical behavior for a specific vertical profile class.*

*In this respect, Figure 6 and related results reported in section 3.3.2 describe the influence of transport from mid-latitudes and the formation of PG profiles. Another example is related to the DNG profiles, which are related to the secondary aerosol formed close to the ground (section 3.3.4 and Figures 9, 10, 11).*

General comment 3. There is no attempt to compare with data obtained by numerous studies in the area using aircraft or lidar techniques. Although several studies are mentioned no quantitative comparsion is given at least for the ground measuremnts or data published.

*AGC3: Thank you for this comment as a comparison with previous studies is very important.*

*An important comparison was reported in the manuscript, page 15 (lines 10-14), where "the columnar averages of both total aerosol number and BC concentrations [236.1±23.9 cm$^{-3}$ ($N_{14-260}$), 21.1±1.3 cm$^{-3}$ ($N_{260-1200}$), 0.2±4*10$^{-2}$ cm$^{-3}$ ($N_{>1200}$) and 52±8 ng m$^{-3}$ (BC)]" were successfully compared with long-term data series collected over Ny-Ålesund at the Zeppelin observatory (Eleftheriadis et al., 2009; Tunved et al., 2013) during Spring.*

*This comparison underlined the accuracy of the collected data and, most importantly, suggested that all the investigated vertical profile classes may influence the background Arctic aerosol measured by Arctic observatories within GAW and EMEP observation programmes.*

*However, we fully agree with you that a better contextualization of the measuring campaign with respect to the international is required. Thus, we modified the introduction section referencing to previous airborne aerosol measurements carried out in Arctic area. Here below a brief resume.*

*The ARCTAS mission (Jacob et al., 2010 and reference therein) evidenced in spring a highly-layered air pollution transport from North America and East Asia characterized by anthropogenic aerosol below 2 km and by biomass burning above (2–4 km). Always in spring, the ARCPAC campign (Brock et al. 2011) allowed to describe and group the aerosol affecting the Arctic in four categories: a background troposphere (relatively diffuse, sulfate-rich aerosol), a region of depleted aerosol within the surface inversion layer over sea-ice, a layer of organic-rich biomass burning aerosol (above the top of the inversion layer) and a layer dominated by fossil fuel combustion.*

*During the trasnition period from spring to summer in Svalbard, the ASTAR campaign (Engvall et al., 2008) found aitken and accumulation mode particles more concentrated in the free troposphere compared to boundary layer while, Kupiszewski et al. (2013), resumed results from the summer ASCOS campaign during which new particle formation events in the near-surface layer, possibly related to biological processes, were found.*

*For what concern the BC, the springtime results reported in the PAM-ARCMIP (Stone et al., 2010) and HIPPO (Schwarz et al., 2010) campaigns evidenced high BC concentrations close to the ground, below the thermal inversion but also dense pollution and BC at high altitudes over the Arctic (Wofsy et al., 2011). Interesting, the PAM-ARCMIP results evidenced a decrease of BC compared with respect to past measurements (i.e. AGASP, Hansen and Novakov, 1989). In addition, the HIPPO campaign revealed that in the lower troposphere the BC vertical gradient can change seasonally from positive to negative (Schwarz et al., 2013). In this respect, Spackman et al. (2010) and Koch et al. (2009) reported BC located mainly in the Arctic free troposphere with a positive gradient in the lower troposphere.*

*All the aforementioned findings appear very important from a long-term monitoring point of view as underline the need for continuous vertical profile campaign in the Arctic to improve the description of a seasonally resolved aerosol and BC vertical behaviour along time.*

*Thus, in order to discuss the results in the context of the aforementioned campaigns, we modified also the result sections including them.*

General comment 4. The use of micro aethalometers in this area can be only used for obtaining EBC concentrations at minimum concentrations, which the authors have yet to derive. They show in figure 2c) a good correlation between the two micro-aethalometers used. This also shows an uncertainty at a 100% level for concentrations below 30 ug/m3 The other serious flaw in the processing of these data is the calculation of the absorption coefficient using a well established methodology and an unrealistic "C" factor. They quote a study in Milan where the "C" factor was derived for urban concentration levels and mixture of urban aerosol species. The authors must remove all absorption coefficients calculated in this manner and reported in this manuscript.

*AGC4: Your comment is divided in three parts. Here below we answer to each one:*
1. *Terminology for BC: we agree with you with the need to report the BC concentrations with the term eBC as suggested by Petzold et al. (2013) and by Andreae and Gelencser (2006) and as widely reported in literature (Gilardoni et al., 2010; Sthol et al., 2013; Eckhardt et al., 2013). We specified it in the method section.*
2. *Figure 2c: data reported in Figure 2c showed a good correlation. This result is first important because the AE5x prototype operated at 265 ml/min ($4.42 \cdot 10^{-6}$ $m^3$ $sec^{-1}$) and the AE51 at 150 ml/min (AE51 commercial version). However, we agree with you that a single point can be affected by a high level of uncertainty and that a deeper analysis of this comparison is required. Moreover, your question allows us to better explain our approach. In this respect, considering the average of the two EBC measurements (AE51 and AE5x) as the target value (as reported at Page 10, line 5) the absolute error (in percentage) of each*

*eBC data was calculated. Figure AGC4.1 reports the average (± standard deviation) and the 90° percentile of the Absolute value of the error in percentage of the measured eBC in function of eBC concentration using intervals of 5 ng m$^{-3}$ each. As it possible to observe at low concentration the error can reach values of 90% and more. This error decrease with increasing eBC concentration and reach a reasonable value (less than 20% for both the average and the 90° Percentile) above 20 ng m$^{-3}$. However, we have to remind that the aim of this paper is to determine the seasonal phenomenology of the aerosol behaviour along vertical profiles classifying the collected experimental data, according to their shape, and averaging them for each season. This is very important as, even the error in percentage of each data point can reach high values (especially at low concentrations), the average of the data stabilize the instrumental fluctuations. This effect is demonstrated by Figure AGC4.2 which reports the correlation between the eBC concentrations (AE51 and AE5x) averaged on the same intervals of 5 ng m$^{-3}$ used in Figure AGC4.1 (R$^2$=0.986; slope=1.017). The results demonstrate the reliability of the seasonal phenomenology of the aerosol vertical profiles reported in Figure 5 and 12 and sections 3.3 and 3.4 along the manuscript for what concern eBC concentrations. We added the aforementioned analysis to the revised version of the manuscript.*

3. *"C" factor: the reviewer's statement does not consider the experimental conditions of the C determination reported in Ferrero et al. (2011). Ferrero et al. (2011) (page 2832) state that: "The experimental design of vertical profiles does not require any estimation of the aerosol loading factor R(ATN): all vertical BC profiles were conducted by changing the filter ticket after each profile. Every Aethalometer measurement cycle (ascent and descent of the balloon) took less than 40–50 min. As a result, ATN never achieved values higher than 20 during all profiles, meaning that the b$_{ATN}$ measurements were not affected by the "shadowing" effect due to filter loading. The average ATN measured along vertical profiles was 5±1". The experimental conditions for the study in Milano (Ferrero et al., 2011) were intentionally selected not to be influenced by the accumulation of the sample on the filter. This means that the total amount of aerosol collected on each filter during the determination of the parameter C was negligible, making the determined C values a function of the "filter material and the instrument specification" as again stated at page 2832. It is also possible to estimate the total amount of aerosol collected on each filter (ascent and descent of the balloon) during the C determination using the data reported in the paper. Considering the average BC concentrations below and above the mixing layer; the AE51 flowrate, the sampling time and the percentage of BC in PM (both below and above the mixing layer), it is possible to determine that on each filter (changed for each balloon launch) the total PM collected was less than 400 ng. The influence of the type of PM on the C values is therefore negligible, and the C reflects the instrumental properties, which are dominated by the highly scattering filter material. Finally, it is also necessary to observe that the C value was determined using data collected not only below the mixing layer but also above it, in a cleaner atmosphere, along the vertical profiles. The reliability of the obtained C (2.05±0.03) was also demonstrated further in Ferrero et al. (2014), both below the mixing layer and in the free troposphere. For the all the aforementioned reasons, considering that also the detailed explanation of the measurement protocol using the Aethalometer AE51 on page 9 of the manuscript, which ends with: "In this study… the filter tickets were changed to always keep ATN lower than 20 as recommended by Weingartner et al. (2003)", we maintain the estimation of the absorption coefficient. However, as the value of the absorption coefficient is very important in the Arctic, due to your question we decided to add the reference of Ran et al. (2016, ACPD) to the paper. In fact, until now, the C (2.05±0.03) reported in Ferrero et al. (2011 and 2014) was the only one for the AE51. Recently, on ACPD Ran et al. (2016) proposed 2.52 as C determined at ground-level in China. This value of the parameter C was*

*determined in a completely different environment and, unfortunately the authors have not (yet) reported its uncertainty.*

[Figure]

Figure AGC4.1. Absolute value of the error in percentage of the measured eBC in function of its concentration.

[Figure]

Figure AGC4.2. Absolute value of the error in percentage of the measured eBC in function of its concentration.

General comment 5. The chemical composition reported in figure 8 is given only in % of the total mass. How is the total mass derived and what are the actual mass concentrations of the different species reported in otherwise incredible detail where the non sea salt and non crustal fractions are calculated? These data do not appear realistic. For example in most cases the EC is found to 0.1 % of the aerosol mass. If one assumes that in the worst case eBC and EC mass concentrations can differ by a factor of 2 (+/- 100%) in the HO case where the eBC is found on average at 25 ng/m3 the aerosol mass concentration levels would range between1 to 50 ug/m3. This upper limit is totally unrealistic and even the 25 ug/m3 is extremely high. The other cases would produce even more grossly biased results.

This comparison puts in doubt the whole dataset of eBC and chemical data leaving the OPC and ground SMPS measurements as the only dataset worth considering for this manuscript.

*AGC5: Thank you for this question which allowed us both to deepen the chemical characterization of the aerosol and to better describe the approach reported in the manuscript. Here below we answer to your questions following their order.*

1- *The chemical composition reported in Figure 8 was given in percentage. This choice was done to put in evidence (section 3.3.4, page 16, lines 21-29) that "the total sulphate fraction in DNG profiles was double than that observed in the other profile classes" and that also "the nss-nc-SO$_4^{2-}$fraction (determined as reported in section 2.1.1) in DNG profiles was 2.4±0.4 times higher than that observed in the other profile classes while both the ss-SO$_4^{2-}$ and the cr-SO$_4^{2-}$ fractions remained quite constant". These observations, coupled with the lowering of eBC fraction in proximity of the ground (Figure 5g) and the SMPS data reported in Figure 9 show that the ground-based N$_{14-260}$ concentration peak was due to secondary origin. Thus, the percentage of aerosol chemical composition was used in support of the other datasets to describe the origin of the aerosol found at ground in DNG profiles (see also AA2).*

*However, do to your question we are going to change figure 8 to report the absolute concentration values (ng m$^{-3}$) of each aerosol chemical component. We moved the previous figure 8, reporting relative contributions, to the supplemental material.*

2- *The total mass was measured by weighing the filters before and after the sampling, using a 5-digit microbalance (Sartorius ME235P) after they were conditioned for 48 hours (25°C and 50% relative humidity). The reproducibility error on filter weighing was lower than 5% (experimentally evaluated) as described in section 2.1 (page 5, lines 5-7).*

*The absolute concentration values (ng m$^{-3}$) of each aerosol chemical component were added in the new figure 8. Moreover the same concentration values (ng m$^{-3}$), together with the corresponding detection limit (ng m$^{-3}$) of each aerosol chemical component were reported in a new Table in the supplemental material. This table (Table AGC5.1) is attached here below. The detection limits refer to the sampling conditions and the analytical procedures. Particularly, as reported in section 2.1.1, for the ion chromatography, only half filter was used (extraction in 12 ml of ultrapure water), 55 m$^3$ of air were filtered for each sample (24 h sampling). For the EC/OC analysis, a 1.5 cm$^2$ punch was used and the volume of air filtered for each sample (96 h sampling) was 220 m$^3$.*

*Table AGC5.1 (please see below) shows that all the analyzed chemical components were characterized by ambient concentrations largely above the detection limit.*

*The non-sea salt and crustal fractions of sodium, calcium and sulphate were determined as well documented in section 2.1.1. This approach produce reliable results as documented in previous works (Udisti et al., 2016; Giardi et al., 2016; Becagli et al., 2012; Udisti et al., 2012) already referenced in the paper text.*

*When comparing the eBC and EC several points has to be considered. First of all they are not the same quantity and, as reported by Petzold et al. (2013), they can differ by a factor of 7 and depend on the assumed MAC (for optical measurements) and the thermal protocol for EC/OC analysis. As EC is the only component, which is at the limit of detection, we decided to not report this component in the new figure 8 (EC is also not important in the discussion reported in section 3.3.4 in relation to DNG profiles).*

*Most important, it has to be considered that vertical profile data are "quasi instantaneous" while the reported chemical composition was determined with the time resolution of 24 h for ionic species and 96 h for EC/OC. In this respect, the chemical speciation and in particular the sulphate content was only used along the paper to support the secondary origin of the ground-based concentration peak of N$_{14-260}$ in DNG profiles. For this purpose,*

*the chemical composition was coupled with information coming from the vertical profiles, the SMPS data and the meteorological parameters.*

*These observations underline the reliability of the whole dataset used in the paper, and therefore we absolutely reject the statement that these data appear non-realistic.*

| Conc (ng m$^{-3}$) | | Na$^+$ | NH$_4^+$ | K$^+$ | Mg$^{2+}$ | Ca$^{2+}$ | Cl$^-$ | NO$_2^-$ | NO$_3^-$ | SO$_4^{2-}$ | Oxalates | F$^-$ | Glycolate | Formate | MSA | EC | OC |
|---|---|---|---|---|---|---|---|---|---|---|---|---|---|---|---|---|---|
| HO | mean | 410.85 | 66.33 | 21.73 | 54.21 | 39.45 | 495.82 | 22.51 | 59.92 | 504.71 | 4.79 | 0.18 | 1.16 | 2.15 | 2.28 | <DL | 534.36 |
| | σ$_m$ | 252.44 | 13.85 | 8.43 | 27.28 | 7.67 | 332.03 | 10.00 | 12.96 | 93.15 | 0.98 | 0.08 | 0.16 | 0.70 | 0.58 | <DL | 39.52 |
| PG | mean | 655.24 | 74.61 | 28.81 | 79.34 | 43.28 | 871.20 | 36.00 | 68.66 | 584.69 | 5.46 | 0.21 | 1.29 | 2.92 | 4.47 | <DL | 522.87 |
| | σ$_m$ | 293.67 | 12.22 | 9.50 | 31.70 | 7.07 | 392.54 | 15.66 | 12.22 | 72.69 | 0.54 | 0.13 | 0.29 | 0.68 | 1.07 | <DL | 52.69 |
| NG | mean | 590.94 | 85.04 | 27.34 | 73.35 | 38.88 | 745.67 | 31.43 | 51.64 | 779.09 | 4.99 | 0.07 | 1.05 | 2.99 | 3.57 | <DL | 517.13 |
| | σ$_m$ | 192.18 | 15.89 | 5.98 | 20.50 | 4.71 | 267.28 | 11.98 | 9.59 | 204.18 | 0.50 | 0.09 | 0.14 | 0.47 | 0.81 | <DL | 43.96 |
| DNG | mean | 325.71 | 128.31 | 23.94 | 51.21 | 33.45 | 357.20 | 20.06 | 49.37 | 1441.91 | 6.97 | <DL | 1.27 | 2.78 | 1.84 | 31.85 | 689.94 |
| | σ$_m$ | 84.23 | 29.21 | 2.40 | 13.16 | 3.17 | 173.05 | 2.33 | 15.90 | 354.09 | 1.12 | <DL | 0.19 | 0.38 | 0.36 | 0.20 | 19.59 |
| Detection Limit | | 0.04 | 0.4 | 0.04 | 0.04 | 0.04 | 0.04 | 0.04 | 0.04 | 0.04 | 0.4 | 0.004 | 0.4 | 0.4 | 0.04 | 11 | 120 |

Table AGC5.1. Ambient concentrations (ng m$^{-3}$) of the aerosol chemical components (mean±mean standard deviation) and their analytical detection limits (DL).

*References:*

[revised manuscript text omitted]

---

## Author Response (AR1)

Dear Editor, Dr. Kostas Tsigaridis,

Thank you and thank to the staff of ACP for your work. We are glad to notice that all the referees appreciated the experimental effort and the high relevance of the results presented in our paper. Furthermore, all the referees asked for elucidation of a number of technical points. All the raised criticisms and relative answers have been addressed in the revised manuscript. Finally, we added the new figures, required to help the reader understand the vertical behavior of aerosol properties. The whole text was proofread, and edited to improve the language. The manuscript was shortened when possible, as new parts were introduced as required by the referees. We are pleased that this discussion based on the constructive criticisms of the referees has helped us to improve the scientific quality of the work done.

With our best regards,

Yours sincerely,

Dr. Luca Ferrero

**Response to Reviewer#1**

We thank the reviewer for his or her helpful comments and insight. We respond to the general and to the specific points below. All the comments are addressed in the revised manuscript. As requested, the whole text was proofread and edited, to eliminate the typos and to improve the language.

General Comment 1: This study reports vertical profiles of aerosol number size distribution and black carbon (BC) concentrations from balloon measurements during a field campaign in Ny-Ålesund, Svalbard spring and summer 2011-2012. The authors divide the number size distribution into 3 modes and classify the vertical profiles in four shapes during spring. The authors also discuss secondary aerosol formation and emissions from shipping during summer.

This study is important because 1) the vertical distribution of aerosols affects its radiative forcing and 2) measurements of the vertical distribution of aerosols in the Arctic are particularly sparse. Since such measurements are highly needed and valuable, I think this study is relevant and within the scope of ACP. The Method section is clearly outlined and the different instruments used are sufficiently and well explained. The figures are nice and easy to follow. However, I think the overall presentation of the results should be improved before it can be published. The manuscript needs more work in terms of language and structure. If this can be achieved, I recommend the manuscript for publication.

Answer to the General Comment 1 (AGC)1: Thank you very much for your comment which underline the experimental efforts and the high relevance of the results presented in our paper. We agree with you that an improved organization of the manuscript and a better presentation of the results is necessary. For this reason, we managed the paper accordingly to your suggestions (here below answered). The whole text was also proofread and edited, to eliminate the typos and to improve the language.

General comment 2. The quality of the English language in this manuscript is variable (some parts are good, but others less good), and I think it would benefit by a thoroughly review of the language (and a spell check!). I have added a few examples under minor comments.

AGC2: The manuscript was proofread and edited, to eliminate the typos and to improve the language as required. A particular attention was given to shorten the several long sentences present in the paper, as also required in your minor comment 4. Thank you for the suggestion.

General comment 3. The Results section would be easier to read if it was shortened a bit. Description of the methodology should always be under Methods, not Results. I have a few specific suggestions below.

AGC3: Thank you for this comment. We agree with you. The description of the methodology was moved to the method section (section 2.2.1, page 9, lines 17-36; section 2.2.3, page 13, lines 1-13). Moreover, sections 3.1 and 3.2 were shortened and merged together in the revised version of the paper (new section 3.1, pages 13-15).

General comment 4. I miss a broader implication of this study. Why did you separate the profiles into the four shapes? Comprehensive measurement studies like this can provide physical understanding for evaluation/improvement of the modeling of aerosol processes. Do you have any suggestions? I understand that you cannot add any modeling, but I would like to know more what we can learn from this study.

AGC4: Thank you for this question. Out answer is also related to that reported below for your specific comment SC22 (see our answer ASC22).

We separated the profiles in the four shapes because each shape is the result of an interplay of several processes: 1) transport events, 2) the planetary boundary layer dynamics and 3) the local formation of aerosol. The different combinations of these factors result in a specific profile class.

Figure 7 represents a good example in which the transport of polluted air masses from midlatitudes generated initially PG profiles that naturally evolved (due to the entrance into the PBL) into a NG profile.

Even though a modelling simulation is beyond the scope of the present paper, some indication can be obtained. One of this is related to your question about the validity of ground-based measurements with respect to the vertical aerosol distribution in modelling comparison (SC21). HO profile showed that ground measurements are fully representative of the vertical column (up until ~1 km, our vertical limit) while during NG and PG profiles the ground based measurements are representative for the column up to the PBL. DNG profiles show that ground-based measurements differ from the measurements performed within the column. However, the last case is influenced by secondary aerosol formation that can be easily detected by an SMPS.

Thus ground-based measurements (coupled with a proper PBL determination) are fundamental for model validation. Due to your question we added the aforementioned consideration in the conclusions (section 4, pages 23-24, lines 36-40 and lines 1-4).

General comment 5. Measurements of vertical profiles in the Arctic are sparse, but there are a few, e.g.:

- two ARCTAS campaigns in the North American Arctic (Jacob et al. 2010) in April and June/July 2008

- the ARCPAC campaign conducted together with ARCTAS in spring 2008 (Brock et al. 2011) -the PAMARCMIP campaign in April 2009 (Stone et al. 2010) -the HIPPO campaign (Schwarz et

al. 2010, 2013; Wofsy 2011) January and October 2009 + winter and autumn 2009

- the ARCTAS/ARCPAC campaign in spring 2008,

- the ARCTAS campaign in summer 2008

- the PAMARCMIP campaign in spring 2009.

On a general basis; How are those compared to your study? I suggest you also include more of these studies in the introduction.

Jacob, D.J., J.H. Crawford, H. Maring, A.D. Clarke, J.E. Dibb, L.K. Emmons, R.A. Fer- rare, C.A. Hostetler, P.B. Russell, H.B. Singh, A.M. Thompson, G.E. Shaw, E. McCauley, J.R. Pederson and J.A. Fisher, 2010. The Arctic Research of the Composition of the Troposphere from Aircraft and Satellites (ARCTAS) mission: design, execution, and first results. Atmospheric Chemistry and Physics, 10:5191-5212.

Brock, C.A., J. Cozic, R. Bahreini, K.D. Froyd, A.M. Middlebrook, A.McComiskey, J. Brioude, O.R. Cooper, A. Stohl, K.C. Aikin, J.A. De Gouw, D.W. Fahey, R.A. Ferrare, R.-S. Gao, W. Gore, J. Holloway, G. Hubler, A.Jefferson, D.A. Lack, S. Lance, R.H. Moore, D.M. Murphy, A. Nenes, P.C. Novelli, J.B. Nowak, J.A. Ogren, J. Peischl, R.B. Pierce, P. Pilewskie, P.K. Quinn, T.B. Ryerson, K.S. Schmidt, J.P. Schwarz, H. Sode- mann, J.R.Spackman, H. Stark, D.S. Thomson, T. Thornberry, P. Veres, L.A. Watts, C.Warneke and A.G. Wollny, 2011. Characteristics, sources, and transport of aerosols measured in spring 2008 during the aerosol, radiation, and cloud processes affecting Arctic climate (ARCPAC) project. Atmospheric Chemistry and Physics, 11:2423-2453.

Wofsy, S.C., 2011. HIAPER Pole-to-Pole Observations (HIPPO): finegrained, global- scale measurements of climatically important atmospheric gases and aerosols. Philo- sophical Transactions of the Royal Society, 369:2073-2086.

Stone, R.S., A. Herber, V. Vitale, M. Mazzola, A. Lupi, R.C. Schnell, E.G.Dutton, P.S.K. Liu, S.M. Li, K. Dethloff, A. Lampert, C. Ritter, M. Stock, R. Neuber and M. Maturilli, 2010. A threedimensional characterization of Arctic aerosols from airborne Sun pho- tometer observations: PAMARCMIP,April 2009. Journal of Geophysical Research: Atmospheres,115:doi 10.1029/2009jd013605.

Schwarz, J.P., J.R. Spackman, R.S. Gao, L. Watts, P. Stier, M. Schulz, S.M. Davis, S.C. Wofsy and D.W. Fahey, 2010. Global-scale black carbon profiles observed in the remote atmosphere and compared to model. Geophysical Research Letters, 37:L18812,doi:10.1029/2010gl044372.

Schwarz, J.P., B.H. Samset, A.E. Perring, J.R. Spackman, R.S. Gao, P. Stier, M.G. Schultz, F.L. Moore, E.A. Ray and D.W. Fahey, 2013b. Global-scale seasonally re- solved black carbon vertical profiles over the Pacific.Geophysical Research Letters, 40:5542-5547.

**AGC5: Thanks for this comment.**

An important comparison was reported in the manuscript (section 3.2, page 16, lines 21-25), where "the columnar averages of both total aerosol number and eBC concentrations [236.1±23.9 cm-3 ( $N_{14-260}$ ), 21.1±1.3 cm-3 ( $N_{260-1200}$ ), 0.2±4\*10-2 cm-3 ( $N_{>1200}$ ) and 52±8 ng m-3 (eBC)]" were successfully compared with long-term data series collected over Ny-Ålesund at the Zeppelin observatory (Eleftheriadis et al., 2009; Tunved et al., 2013) during Spring.

This comparison underlined the accuracy of the collected data and, most importantly, suggested that all the investigated vertical profile classes may influence the background Arctic aerosol measured by Arctic observatories within GAW and EMEP observation programs.

Moreover, at page 3 lines 23-25 (original version of the manusript) we cited some of the Arctic campaigns (i.e. Kupiszewski et al., 2013; Schwarz et al., 2010).

However, we fully agree with you that a better contextualization of the measuring campaign is required.

Thus, we modified the introduction section, adding and discussing the suggested campaigns (and related references). Moreover, we discussed the obtained results with respect to the same references.

Particularly we modified the introduction section at page 3, lines 26-40 and at page 4, lines 1-10. The result section was modified at the following points: section 3.2 (page 16, lines 24-29), section 3.2.1 (page 17, lines 1-2 and lines 17-24), section 3.2.2 (page 18, lines 4-6 and lines 12-17), section 3.2.3 (page 19, lines 4-10), section 3.2.4 (page 20, lines 12-14, page 21, lines 13-25).

Specific Comment 1 (SC1): Abstract: You should mention in the abstract that these were balloon measurements up to 1200 meters height.

Answer to the Specific Comment 1 (ASC)1: Thank you very much, we modified the abstract accordingly to your suggestion at page 1, lines 26-27.

SC2: Page 2, L 17: 'to influence with semi-direct effects the atmospheric properties'. Could this be rewritten and explained further, maybe by 1 or 2 examples?

ASC2: The sentence was rewritten at page 2, lines 15-20, as follows: "Many of these processes depend on aerosol absorption and scattering of the solar radiation (direct effect). Additionally, indirect effects play an important role as the aerosols seed and modify the cloud properties. Lastly, light absorption by BC can alter the atmospheric thermal structure within, below, or above clouds consequently affecting cloud distributions (IPCC, 2013; Bond et al., 2013; Ramanathan and Feng, 2009; Koren et al. 2008; Koren et al., 2004; Kaufman et al., 2002)".

SC3: Page 2, L 34: You mention Arctic Haze here without explaining it. Since this is an important part of your study, I think you should briefly explain the phenomena with a few references (e.g. Stohl 2006). Stohl, A. (2006), Characteristics of atmospheric transport into the Arctic troposphere, J. Geophys. Res., 111, D11306, doi:10.1029/2005JD006888.

ASC3: Thanks for raising this point, we agree with you that a better description of the Arctic Haze could help the reader. For this reason, we added to the introduction section (pages 2-3, lines 34-39 and line 1) the following description: "During the Arctic Haze, an inflow of pollution (aerosol and gases) from northern mid-latitudes (during winter-spring) results in a reduction in visibility (Jacob et al., 2010; Sthol et al., 2006; Radke et al., 1984; Barrie and Hoff, 1985; Brock et al., 1989; Shaw, 1995). The Arctic Haze occurs under meteorological conditions with stable stratifications and the frequent and persistent occurrences of surface-based inversions. According to Stohl et al. (2006), within these conditions, the air pollution can be transported into the Arctic at low-level (followed by ascent in the Arctic or low-level alone) or with an uplift outside the Arctic, followed by descent in the Arctic itself".

SC4: Page 3, L 25: 'These reports may well highlight opposing forms of behavior '. I am not quite sure what this means?

ASC4: The intention was to underline the differences in the vertical aerosol behavior found during the reported field campaigns. However, to avoid any confusion, we delated this sentence.

SC5: Page 3, L 25: One reason for this difference between the observations could be the strong influence of biomass burning during spring 2008 (Warneke et al. 2010).

Warneke, C., K.D. Froyd, J. Brioude, R. Bahreini, C.A. Brock, J. Cozic, J.A.de Gouw, D.W. Fahey, R. Ferrare, J.S. Holloway, A.M. Middlebrook, L.Miller, S. Montzka, J.P. Schwarz, H. Sodemann, J.R. Spackman and A.Stohl, 2010. An important contribution to springtime Arctic aerosol from biomass burning in Russia. Geophysical Research Letters, 37:L01801, doi:10.1029/2009GL041816.

ASC5: Thank you very much for the suggestion, we added the reference to the introduction section (page 3, line 33).

SC6: Page 3, L 30: Drop 'should', as this is written it seems like you tell the emissions to do so? In bracelets: we do not know for sure if these emissions will warm the surface and be deposited, but as you write above; studies show that there are higher probability for this to happen when the concentrations are located close to the surface.

ASC6: Thank you. We modified the sentence.

SC7: Page 4, L 23: Could you add just one sentence summarizing this table? E.g. 25 measurement days, balloons measured 2-14 profiles each day, altitude range?

ASC7: Thank you for the suggestion. The new sentence (page 5, lines 11-13) is as follows: "Table 1 lists the dates of the campaign (25 measurement days), the number of flights (197 measured profiles), the maximum altitudes (~700-1300 m) and the cloud base height (clouds present for 48% of the campaign)."

SC8: Page 5, L 5: Is there a reference for this instrument and the calculated uncertainty in mass concentrations?

ASC8: As stated at page 5, line 32, we used the 5-digit Sartorius ME235P microbalance. In this respect, 5% is the uncertainty related to the weighing procedure experimentally evaluated. The text is changed as follows (page 5, lines 32-33): "The reproducibility error for filter weighing was lower than 5% (experimentally evaluated)".

SC9: Page 6; L 18: Could you briefly here explain what you mean when the atmosphere is 'stable' and does not encourage vertical mixing? (in terms of potential temperature)

ASC9: We modified the sentence as follows (page 7, lines 1-6): "The term stability refers to the propensity of air masses to move vertically: stable air resists any vertical motion, while unstable air masses are prone to vertical movements. A parcel of air results to be stable/unstable if the temperature lapse rate is lower/higher than the adiabatic one, i.e. if the potential temperature is increasing/decreasing with height, respectively. In stable stratification, turbulence and vertical mixing is suppressed, leading to trapping of pollutants near ground level".

SC10: Page 8, L 28: In this paragraph you define the 3 modes of particles, 'Aitken', 'Accumulation' and 'coarse' and say that you will also use these names for the rest of the discussion, but most of the time you use N14-260, N260-1200, and N>1200 anyway. I suggest you use the names Aitken etc. throughout the text once you have defined them, as this is easier to read.

ASC10: Thank you for addressing it. With these sentences we wanted to explain the meaning of each investigated broadsize range. We carefully considered your comment. However, " $N_{14-260}$  includes a small fraction of the Nucleation mode (from 14 to 20 nm), the totality of the Aitken mode (20-100 nm) and a fraction of the Accumulation mode (from 100 to 260 nm)" (page 9, lines 4-5); in addition, "the mode  $N_{260-1200}$  includes most of the Accumulation mode particles" (page 9, lines 5-6). Thus, as both  $N_{14-260}$  and  $N_{260-1200}$  are note "pure/whole" Aitken and Accumulation modes we decided to maintain the manuscript in the present form. A similar approach was also used in Kupiszewski et al. (2013).

SC11: Page 8, L 36: Since there are many figures in this paper; I suggest removing fig 2 (or move to the supplementary).

ASC11: Figure 2 was moved to the supplemental material (now Figure S1).

SC12: Page 9: There are various methods to measure BC concentrations, and they can disagree by a factor of seven or more (Petzold et al 2013). Since the (common) filter-based method like you have used is not a direct measurement of BC, it is recommended to report the resulting BC concentration (eBC) together with the assumed MAC value. Maybe you should change 'BC' to 'eBC' to make sure that we know that this is equivalent BC? I also think you should add a brief discussion on how your measurements depend on the assumed MAC number (you use 12.5 m2/g?) (or at least make a note about this).

Petzold, A., J.A. Ogren, M. Fiebig, P. Laj, S.M. Li, U. Baltensperger, T. Holzer-Popp, S. Kinne, G. Pappalardo, N. Sugimoto, C. Wehrli, A. Wiedensohler and X.Y. Zhang, 2013. Recommendations for reporting "black carbon" measurements. Atmospheric Chemistry and Physics, 13:8365-8379.

ASC12: Petzold et al. (2013), Andreae and Gelencser (2006) and other authors (Gilardoni et al., 2010; Sthol et al., 2013; Eckhardt et al., 2013) suggest reporting BC concentrations with the term equivalent Black Carbon (eBC), arising from the need to report the method to determine BC and the parameters used in the method. We specified it in section 2.2.2 (page 9, line 40 and page 10, lines 1-4). Moreover, we used the term eBC throughout the whole paper.

When optical methods are used to measure light transmission through the filter loaded with BC, the mass equivalent concentration is determined using the mass attenuation cross-section ( $\sigma_{ATN}$ ). For the case of our measurements, we report the  $\sigma_{ATN}$  value 12.5 m2 g-1 We note this value (Ferrero et al., 2011) in the method section, as the one used in the micro-Aethalometer AE51 (page 10, line 20). This approach was also used by Eleftheriadis et al. (2009) when reported ten years of BC measurements in Ny-Ålesund.

Moreover, the  $\sigma_{ATN}$  value (12.5 m2 g-1) "was obtained by comparing the BC values measured with the microAeth® Model AE51, with an AE31 Aethalometer (880 nm wavelength) operating in a test chamber with different BC concentrations at low attenuation values. The comparison was then repeated using ambient air" (as reported in Ferrero et al., 2011).

This value is not far from the  $\sigma_{ATN}$  values of 15.2 m2 g-1 and 15.9 m2 g-1 reported in Eleftheriadis et al. (2009) and applied to the attenuation coefficient measured at the Zeppelin station (Ny-Ålesund) with the Aethalometers AE9 and AE31, respectively. The difference between these values results from the use of different filter materials to collect the sample in the different Aethalometers, which was quantified in Ferrero et al. (2011) and Drinovec et al. (2015).

Due to your question we added the aforementioned specifications to section 2.2.2 (page 10, lines 21-28).

We compare our results with those previously measured at Ny Ålesund in the manuscript (page 16, lines 21-24), where we state: "the columnar average of BC concentrations obtained by averaging the profile classes was  $52\pm8$  ng m-3 (eBC)". This value "perfectly agrees with long-term data series collected over Ny-Ålesund at the Zeppelin observatory (Eleftheriadis et al., 2009) during Spring."

Finally, considering the relationship between the attenuation and absorption coefficients (page 11, equation 11), the apparent mass attenuation cross-section of 12.5 m2 g-1 corresponds to the mass absorption cross-section (MAC) of 6.1 m2 g-1 (using C=2.05±0.03). As suggested by Petzold et al. (2013), we report the wavelengths, the mass attenuation cross-section and the mass absorption cross-section used in the determination of the absorption coefficients, using the methodology reported in Weingartner et al. (2003). The MAC of 6.1 m2 g-1 is in agreement with the previously published range of values (see for example Petzold et al. (2013) and references therein).

SC13: Page 9: Filter-based methods are sensitive to absorbing and non-absorbing non-BC particles. Could you please add a few sentences about the uncertainties in your method as well?

ASC13: Considering the absorbing non-BC particles, different types of aerosol may in principle contribute to the signal in Aethalometers (i.e. Brown Carbon, dust). However, Brown Carbon (BrC) is characterized by negligible absorption in the infrared (Andreae and Gelencsér, 2006), the wavelength range of the eBC measurements (micro-Aeth AE51 uses 880 nm). In this respect, Massabò et al. (2013) show the potential contribution of BrC to the determination of eBC to be below 10%.

To estimate the possible influence of BrC on eBC measurements carried out during the spring 2011 campaign, the few data collected with the micro-Aeth prototype at 370 and 880 nm were considered. Particularly, the Aethalometer model (Sandradewi et al., 2008) was applied to the apportionment of absorption due to both BC and BrC as reported in Massabò et al (2013) and in Shamjad et al. (2015) as follows:

$$\frac{b_{abs}(370 \ nm)_{BC}}{b_{abs}(880 \ nm)_{BC}} = \left(\frac{370}{880}\right)^{-\alpha_{BC}} \tag{1}$$

$$\frac{b_{abs}(370 nm)_{BrC}}{b_{abs}(880 nm)_{BrC}} = \left(\frac{370}{880}\right)^{-\alpha_{BrC}}$$
(2)

$$b_{abs}(\lambda) = b_{abs}(\lambda)_{BC} + b_{abs}(\lambda)_{BrC}$$
(3)

where  $\alpha_{BC}$  and  $\alpha_{BrC}$  represent the Absoprtion Angstrom Exponents of BC and BrC, respectively. The value for  $\alpha_{BC}$  was taken to be 1 as suggested by Massabò et al (2013) and Sandradewi et al. (2008), while  $\alpha_{BrC}$  was set at 3.5 (Yang et al., 2009), 3.95 (Massabò et al., 2013), 6.6 (Shamjad et al., 2015) and 9.0 (Bikkina and Sarin, 2013), respectively. With this inputs, the percentage of absorption at 880 nm due to BrC instead that BC was 8.5%, 5.8%, 0.5% and 0.1%, respectively. Thus, it is

possible to estimate that the BrC positive artifact on eBC measurements was less than 10% during the campaign.

The effect of non-absorbing particles on the filter photometer measurements was also quantified for the Aethalometer AE33. Drinovec et al. (2015), which uses the same filter material as the AE51, and was shown to be below 2.5%. To reduce the effect of the non-absorbing particles sampled onto the filter, the experimental protocol during the campaign followed that reported in Ferrero et al. (2011): all vertical BC profiles were conducted by changing the filter ticket regularly. As a result, ATN never achieved values higher than 20 during all profiles. This means that the total amount of aerosol collected on each filter during the vertical profile was very low resulting in a negligible effect of non-absorbing particles.

We added the aforementioned considerations to the supplemental material and referenced them in section 2.2.2 (page 12, lines 3-9).

SC14: Page 11, L 15: 'Figure S1a shows a larger interannual springtime variability.' Of what?

ASC14: Thanks for addressing this point, with this sentence we meant that Figure S1a (now Figure S3a) shows a larger interannual springtime variability of temperature measured close to the ground. Due also to your question here below (SC15) we rewrite the manuscript text (page 13, lines 27-33) with the following sentence: "First of all, the observational periods (spring 2011, summer 2011 and summer 2012) were addressed in a climatological context. In this respect, the temperature measured in spring 2011 was within the standard deviation range of the long-term observations, while a 10-day period at the end of April 2011 was slightly warmer than the climatological mean (Figure S3a). The temperatures during the summer seasons 2011 and 2012 were mostly within the range of the long-term observations, so the vertical profile measurements can be considered to have been obtained under typical meteorological conditions representative for the Ny-Ålesund environment."

SC15: Page 11, L 13 - 20: Since you are referring to figures in the supplement, maybe you could rewrite this paragraph so this is easier to follow without the figures? Not 'Fig Sxx shows ...' but instead just briefly state that the spring season had surface temperatures close to the climatology, summer season had .. etc. and then mention that figures are in Supplementary (–OR- move the sup. figures to the paper, but then you already have many figures there).

ASC15: We modified the paper accordingly to your observation (see ASC14 and pages 13, lines 27-33).

SC16: P11: 'Particularly, the maximum wind speeds registered at ground during balloon flights in spring 2011 and in summer 2011 2012 were 4.9 m s and 10.7 m s lower than the absolute wind speeds registered during the same periods: 27.9 m s and 16.3 m s.' I'm not sure if I understood this. The absolute winds measured by ..? With movement? How do you conclude that the measurement periods are representative for days with low winds?

ASC16: The wind speed was measured at the Amundsen-Nobile Climate Change Tower as described in section 2.1.2 (page 6, lines 36-38). The wind speed (average, max value) measured during balloon flights was lower than that during the whole period (April 2011, June and July 2011-2012) of the campaign. Thus, we reported that balloon flights have limitations with respect to its launch conditions, in particular they favor low wind conditions as it is very difficult to launch the balloons during high winds. We understand from your question that the sentence was not clear. Thus, we rephrased it as follows (pages 13-14, lines 34-40 and lines 1-2): "We note, however, that the tethered balloon measurements have limitations with respect to its launch conditions (section

2.2). Particularly, balloon profiles were measured in low wind conditions, as it is very difficult to launch the balloons during high winds. This introduces a bias in respect to average meteorological conditions above the launch site. The maximum wind speed measured at the Amundsen-Nobile Climate Change Tower (section 2.1.2) during balloon flights was lower than that during the whole period of the campaign (April 2011, June and July 2011-2012): 4.9 m s-1 and 10.7 m s-1 (springtime and summertime balloon profiles) compared to 27.9 m s-1 and 16.3 m s-1 (full spring 2011 and summer 2011-2012). Table 1 resumes the conditions for all the measured profiles. The majority of vertical profile measurements was conducted under clear sky conditions (no clouds) or with clouds with base height above the balloon payload."

SC17: Page 11, line 32 - page 13, line 14: I think you spend too much time explaining figure 3. Parts of this can be moved to Methods, e.g. what type of information you can retrieve from the measurements. You can also move parts to Introduction as a way of motivating the study. When I read the 'Results'-chapter I want to read about the results right away. Could you also try to merge some of this information when you present the other results? I would skip everything between L31, p11 to L21,p12 and start on 'An example ...'. Is fig 3 needed at all? Why cannot the measured potential temperature and the RH for each group be plotted in fig 5 instead?

On the other hand, figure 7 is hardly mentioned. Can the wind roses be put in better context with the profiles described in 3.3.1-3.3.4? Also, this text is a bit hard to read, because of all the numbers listed. Do you need to list them all? Maybe put them in a table?

ASC17: We agree with your observations. Particularly, as we reply in AGC3 to your general comment, all the description of the methodology presently in the result section was moved under the method section. Sections 3.1 and 3.2 were shortened and merged together in the revised version of the paper. Figure 3 (now Figure 2) was maintained in the revised version to introduce the set of vertical profile measurements. We added averaged meteorological parameters to figure 5 (now Figure 4 in the revised version of the manuscript).

Figure 7 (now Figure 5) was put in evidence in the revised version of the paper when discussing each profile class (section 3.2.1, page 17, lines 2-5; section 3.2.2, page 17, lines 29-32; section 3.2.3, page 18, lines 21-22; section 3.2.4, page 19, lines 15-17; section 3.3.1, page 22, lines 5-8; section 3.3.2, page 22, lines 31-32. Finally, a table (Table 3) resuming data of the campaign was added to the revised version of the manuscript.

SC18: Page 17, L3: does this text and forward belong to 3.3.4 or should it have a separate heading?

ASC18: Page 17, line 3 and the following lines (original version) belong to section 3.3.4 (now section 3.2.4). Section 3.2.4 was quite long, but we shortened it following your suggestion (see AGC3) moving the methodology present here in the method section.

SC19: Page 18: anything that has to do with methodology should be under Methods, not Results.

ASC19: We agree with you and we modified the paper accordingly (see AGC3).

SC20: Page 19, L16: what is meant by a 'meaningful' impact of ship emissions?

ASC20: The intention was to point out that profiles were affected by a local, high, plume emitted from the ships. We agree that the sentence, as stated, was not clear and thus we rephrased it as follows (page 21, lines 34-35): "Type 2, profiles characterized by the presence of shipping emissions (hereinafter addressed as SP), Figure 11d-f."

SC21: Page 20: It is interesting to see the impact of ship emissions. Could you remind us here how far the measurements were from the ships? This also relates to your final conclusion on page 21 (where you suggest that increased shipping could significantly increase BC concentrations during summer and enhance climate change in the Arctic). Currently, BC emissions from shipping in the Arctic comprise a small fraction of within- Arctic BC. Browse et al. 2012 found that even under a high-projection of shipping, by 2050 BC emissions from shipping would still contribute less than 1 % of total Arctic deposition. Do you suggest that current emission inventories are too low and that future emission projections should also be higher?

Browse, J., K.S. Carslaw, S. Arnold, K.J. Pringle and O. Boucher, 2012. The scavenging processes controlling the seasonal cycle in Arctic sulphate and black carbon aerosol. Atmospheric Chemistry and Physics, 12:6775-6798.

ASC21: Vertical profile measurements in summer were carried out from the German-French AWIPEV research base (78°55'24" N 11°55'15"E), 600 m far from the harbor (we added this information to section 2, page 5, line 10). Thus, the reported results (section 3.3.2, Figure 11d-f and Figure 12) refer to the local impact of ships.

The obtained results are in agreement with those reported in Eckhardt et al. (2013), who showed an enhancement of 72 and 45 % (up to 81 and 72 % in stagnant conditions) of eBC, when ships cruised in the Kongsfjord, compared to values when ships were not present.

From these results, yet Eckhardt et al. (2013) concluded that the eBC increase due to shipping emission can "be taken as a warning signal of future pan-Arctic conditions if Arctic shipping becomes more frequent and emission regulations are not strict enough".

On the other hand, as you addressed, shipping contribution was less than 1% of BC emissions in the Arctic as reported in in the study of Stohl et al. (2013) where, conversely, gas flaring was estimated to contribute 42% to the annual mean BC surface concentrations in the Arctic dominating the estimated BC emissions north of  $66^{\circ}$  N.

However, even also Corbett et al. (2010) reported that "the magnitude of emissions from shipping on a mass basis may be modest compared to other anthropogenic sources, the proximity of activity to the Arctic may help explain regional effects important for global and regional climate change". The possible implications of shipping emissions were also addressed in the work of Sand et al. (2013), who underlined that the BC aerosols would be emitted by ships directly into the Arctic planetary boundary layer with a stronger interaction with the surface (both by deposition of BC on snow and ice and by radiative and sensible heat fluxes down to the surface).

Given the aforementioned considerations and the experimental conditions of vertical profile measurements, we rephrased the conclusion referring to the impact of shipping emission at a local scale (page 23, line 10).

The emission inventories are beyond the scope of the present work. We report the increased number of ships (page 22, lines 19-23) and passengers (a proxy of ship dimensions) in summer 2012 (78 days with 138 ships) relative to summer 2011 (57 days with a total of 103 ships). The obtained results can be thus considered an important phenomenon which should remain under observation in the future due to the sensitivity of the Arctic environment.

SC22: Page 20, L22: 'forbidden' – by who/what? What is meant by: '. And the locally formed aerosol becomes in summer' ?

Do you find any (systematic) correlation between the different vertical profiles and the measurements at the ground? E.g. for special ground conditions, one can assume (with some certainty) a particular profile? Or that when using ground measurements (which are more abundant) when comparing to models, it is not such a bad assumption?

ASC22: Thank for addressing this point, the sentence was incomplete due to an erroneous application of tracking changes in the word processor. We rewrote the sentence as follows (page

23, lines 5-8): "It is important to note that SP profiles were observed in summer. In summer the long-range transport of aerosol from mid-latitudes is minor (Browse et al., 2012; Quinn et al., 2008; Stohl et al., 2006) and the locally formed aerosol becomes dominant (Giardi et al., 2015; Tunved et al., 2013; Ström et al., 2009 and 2003)".

Figure 4 clearly shows that for HO profiles ground measurements are fully representative of the vertical column (up until  $\sim 1$  km, our vertical limit); during NG and PG profiles the ground based measurements are representative for the column up to the planetary boundary layer. DNG profiles show that ground-based measurements differ from the measurements aloft. However, the last case is influenced by secondary aerosol formation that can be easily detected by an SMPS (or similar experimental devices).

Given the above considerations, in our opinion, ground-based measurements (coupled with a proper PBL determination) are fundamental and very useful for model validation. We added these considerations in the conclusion section (pages 23-24, lines 36-40 and lines 1-4).

Minor Comment 1 (MC1): Line 28 page 2: write the Q as a full sentence, e.g. How does the aerosol (...) vary by season?

Answer to the Minor Comment 1 (AMC)1: The sentence was changed into: "To adopt the right mitigation strategies, key scientific issues in the study of Arctic aerosols has to be solved. They include the identification of the relative importance of long-range advection with respect to local emissions (Flanner, 2013; Sand et al., 2013; Shindell and Faluvegi, 2009).

Most important, the seasonal characterization of the aerosol vertical structure, a very poorly determined piece of information, is required" (page 2, lines 25-29).

MC2: Page 2, line 31: 'Very pronounced'  $\rightarrow$  drop 'very'

AMC2: Done

MC3: Page 3, line 1: know  $\rightarrow$  known

AMC3: Thanks, corrected.

MC4: There are several long sentences in this paper, which makes it a bit hard to read. E.g. Page 3. Line 2-7 is one sentence over 6 lines. Could this be split in 2? Also in this sentence: 'leads'  $\rightarrow$  'could lead'.

AMC4: We agree with you. We modified the sentence at page 3 and we shortened the long sentences present in the paper. Thanks for the suggestion.

MC5: Page 4, line 28: form  $\rightarrow$  from

AMC5: Thanks, corrected.

MC6: Page 6, line 5 double ..

AMC6: Thanks, corrected.

MC7: Page 6, line 13: operates since  $2009 \rightarrow$  'have operated'

AMC7: Thanks, corrected.

MC8: Page 6, line 21: 'during snow covered or not periods' Please rewrite.

AMC8: We rephrased and move the sentence at page 14, lines 11-16 (section 3.1).

MC9: Page 8, line: closets  $\rightarrow$  closest

AMC9: Thanks, corrected.

MC10: Page 11, line 2: 'Aerosol and BC and vertical profile (. . .)' Please rewrite. By vertical profile do you mean the meteorological fields? And aerosol are the size distributions?

AMC10: We rephrased the sentence as: "Vertical profiles of aerosol number size distribution and eBC concentrations were measured to assess changes in aerosol properties within the vertical column in the Arctic region" (page 13, lines 16-17). Here the intention was to draw the reader's attention towards the topic of this paper: the determination of the vertical behavior of the aerosol properties (number size distribution and eBC concentration) in the Arctic.

MC11: Page 11, Line 10: 'Before to introduce' .. please change

AMC11: We rephrased the sentence as: "Here below, the ambient conditions under which the vertical profiles were measured are briefly described" (page 13, lines 25-26).

MC12: P 11, L 24: 'Moreover, quite all measurements were conducted' quite all? You mean 'all'?

AMC12: Table 1 summarizes the conditions during the measuring campaign. It can be observed that, during the majority of vertical profile measurements, clear sky conditions were present. Due to your question we rephrased the sentence to clarify this point (page 14, lines 1-2).

MC13: P11, L31: drop 'now'

AMC13: Done

MC14: P11, L34: 'Several information can be derived' please rewrite

AMC14: We rephrased the sentence as (page 13, lines 1-3): "Examples of  $AS_h$ , accompanied with the corresponding potential temperature ( $\theta$ ) and RH profiles, are presented in Figure 2a-d. The presented data, accurately describe the vertical distribution of the aerosol and its properties in the first kilometer above Ny-Ålesund".

MC15: P11, L34: P20, L24: reasing -> rising?

AMC15: Yes, rising. Thanks, corrected.

**Response to Reviewer#2**

We thank the reviewer for his or her helpful comments and insight. We respond to the general and specific points below. All the comments are addressed in the revised manuscript. As requested, the whole text was proofread and edited, eliminate the typos and to improve the language.

General Comment 1: This manuscript describes vertical profiles of aerosol number density over Arctic during spring and summer and presents authentic and original scientific material that has relevant implications for atmospheric science (aerosol, clouds, CCN, and others). This study is based on very important aerosol data over Ny-Ålesund, Svalbard, although the tethered-balloonborne aerosol measurements are restricted to the good weather (i.e., clear sky, calm winds etc.). On the whole, the topic of the manuscript is relevant and suitable for the scope of the "ACP". However, there are several points which require some careful revision and corrections before publication.

Answer to the General Comment 1 (AGC)1: Thank you very much for your comment which underline the originality and the high relevance of the results presented in our paper. Concerning your request of revision and correction we managed the paper accordingly to your suggestions (here below answered). The whole text was also proofread and edited, to eliminate the typos and to improve the language.

General comment 2. Quality of English I found many typo, miss-spell, and grammatical errors (e.g., location of ",").

AGC2: The manuscript was proofread and edited, to eliminate the typos and to improve the language as required.

General comment 3. Comparison with previous airborne aerosol measurements Several airborne aerosol measurements have carried out in Arctic area since 2000, for instances, ASTAR (2000, 2004, and 2007), ARCTAS (2008), ARCPAC (2008), and PAMARCMiP (2009 and 2011). Particularly, ASTAR campaigns were made around Svalbard. I suggest strongly that your data are compared to these previous results, and that these campaigns should be added into description of introduction.

AGC3: Thank you for this comment. An important comparison was reported in the manuscript (section 3.2, page 16, lines 21-25), where "the columnar averages of both total aerosol number and BC concentrations [236.1±23.9 cm-3 ( $N_{14-260}$ ), 21.1±1.3 cm-3 ( $N_{260-1200}$ ), 0.2±4\*10-2 cm-3 ( $N_{>1200}$ ) and 52±8 ng m-3 (BC)]" were successfully compared with long-term data series collected over Ny-Ålesund at the Zeppelin observatory (Eleftheriadis et al., 2009; Tunved et al., 2013) during Spring. Moreover, at page 3 (lines 23-25, original version of the manusript) we cited some of the Arctic campaigns (i.e. Kupiszewski et al., 2013; Schwarz et al., 2010).

However, we fully agree with you that a better contextualization of the measuring campaign with respect to the international is required. Thus, as also required by reviewer#1, we modified the introduction section adding and discussing the suggested campaigns. Moreover, as you suggested, we discussed the obtained results with respect to the same references.

Particularly we modified the introduction section at page 3, lines 26-40 and at page 4, lines 1-10. The result section was modified at the following points: section 3.2 (page 16, lines 24-29), section 3.2.1 (page 17, lines 1-2 and lines 17-24), section 3.2.2 (page 18, lines 4-6 and lines 12-17), section 3.2.3 (page 19, lines 4-10), section 3.2.4 (page 20, lines 12-14, page 21, lines 13-25).

General comment 4. Classification of aerosol type In this study, authors classified aerosol profiles into four groups. I agree with the classification of aerosol profiles. Unfortunately, typical weather/meteorological conditions and air mass origins in each type were not mentioned in the text. These information is very important to characterize vertical features of aerosols in Arctic region, and to be compared to aerosol data taken in the other project.

**AGC4: Thank you for your comment, which supports the classification of aerosol profiles.**

The weather and meteorological conditions for each profile class were addressed in Table 2, Figure 7 (now Figure 5) and discussed along section 3.2 and 3.3. However, they were mainly addressed to illustrate the differences between DNG profiles and the other profile classes. We modified the paper introducing the meteorological context for each profiles class.

In this respect, we modified the result section at the following points: section 3.2 (page 16, lines 30-33), section 3.2.1 (page 17, lines 3-11), section 3.2.2 (page 17, lines 29-34), section 3.2.3 (page 18, lines 21-25), section 3.2.4 (page 19, lines 15-19) section 3.3.1 (page 22, lines 5-9), section 3.3.2 (page 22, lines 31-35).

The air mass origins are addressed below and we refer to the following explanation also for the answer to your specific comment 9.

We agree with you that air mass origin was shown and explained (in the original version of the manuscript) only for the case study reported in figure 6 (now Figure 7) and discussed in sections 3.2.2 and 3.2.3 for both PG and NG profiles.

However, before trying to find a relationship between the type of the vertical profile and air mass origin, it is necessary to consider that each profile shape is the result of an interplay among several processes: 1) transport events, 2) the planetary boundary layer dynamic and 3) the local formation of aerosol. Among this, only the transport event process is strongly related to the air mass origin (some precursors transported may also affect secondary aerosol formation), while the final profile shape is the result of the specific combination of the aforementioned processes.

Figure 7 represents a good example in which the same air mass origin that transported polluted air masses from mid-latitudes generated initially PG profiles that naturally evolved (due to the entrance into the PBL) into NG profiles.

Thus, the same air mass origin could be related to different profile classes.

Due to your question, even with the aforementioned limitations, we performed a cluster analysis of back-trajectories corresponding to each profile class obtained at using the Hysplit 4 (rev. 513). The result is attached here below (Figure AGC4.1).

From Figure AGC4.1 it is possible to observe first that back-trajectories were close in the Arctic area during HO profiles. HO profiles were then representative for background conditions in the Arctic. Both PG and NG can be affected by transport from mid-latitudes. The same happened for DNG profiles, which are a special type of NG profiles.

We added this figure and the aforementioned discussion to the supplemental material (Figure S6) and we address them in the revised version of the manuscript (page 16, lines 35-36).